# Online-Within-Online Meta-Learning

**Giulia Denevi**[1,2]**, Dimitris Stamos**[3]**, Carlo Ciliberto**[3,4] **and Massimiliano Pontil**[1,3]

[1]Istituto Italiano di Tecnologia (Italy), [2]University of Genoa (Italy),
[3]University College of London (UK),[4]Imperial College of London (UK),
*giulia.denevi@iit.it, c.ciliberto@imperial.ac.uk, {d.stamos.12,m.pontil}@ucl.ac.uk*

## Abstract

We study the problem of learning a series of tasks in a fully online Meta-Learning setting. The goal is to exploit similarities among the tasks to incrementally adapt an inner online algorithm in order to incur a low averaged cumulative error over the tasks. We focus on a family of inner algorithms based on a parametrized variant of online Mirror Descent. The inner algorithm is incrementally adapted by an online Mirror Descent meta-algorithm using the corresponding within-task minimum regularized empirical risk as the meta-loss. In order to keep the process fully online, we approximate the meta-subgradients by the online inner algorithm. An upper bound on the approximation error allows us to derive a cumulative error bound for the proposed method. Our analysis can also be converted to the statistical setting by online-to-batch arguments. We instantiate two examples of the framework in which the meta-parameter is either a common bias vector or feature map. Finally, preliminary numerical experiments confirm our theoretical findings.

## 1 Introduction

Humans can quickly adapt knowledge gained when learning past tasks, in order to solve new tasks from just a handful of examples. In contrast, learning systems are still rather limited when it comes to transfer knowledge over a sequence of learning problems. Overcoming this limitation can have a broad impact in artificial intelligence, as it can save the expensive preparation of large training samples, often humanly annotated, needed by current machine learning methods. As a result, Meta-Learning is receiving increasing attention, both from applied [15, 32] and theoretical perspective [5, 40, 17].

Until very recently, Meta-Learning was mainly studied in the batch statistical setting, where data are assumed to be independently sampled from some distribution and they are processed in one batch, see [6, 23, 24, 25, 26, 29]. Only recently, a lot of interest raised in investigating more efficient methods, combining ideas from Online Learning and Meta-Learning, see [1, 12, 13, 30, 3, 21, 16, 8, 11, 30]. In this setting, which is sometimes referred to as *Lifelong Learning*, the tasks are observed sequentially – via corresponding sets of training examples – and the broad goal is to exploit similarities across the tasks to incrementally adapt an inner (within-task) algorithm to such a sequence. There are different ways to deal with Meta-Learning in an online framework: the so-called *Online-Within-Batch* (OWB) framework, where the tasks are processed online but the data within each task are processed in one batch, see [1, 12, 13, 16, 8, 3, 21], or the so-called *Online-Within-Online* (OWO) framework, where data are processed sequentially both within and across the tasks, see [1, 3, 21, 16, 11]. Previous work mainly analyzed specific settings, see the technical discussion in App. A. The main goal of this work is to propose an OWO Meta-Learning approach that can be adapted to a broad family of algorithms.

We consider a general class of inner algorithms based on primal-dual Online Learning [37, 33, 38, 36, 35]. In particular, we discuss in detail the case of online Mirror Descent on a regularized variant of the empirical risk. The regularizer belongs to a general family of strongly convex functions parametrized by a meta-parameter. The inner algorithm is adapted by a meta-algorithm, which also consists in applying online Mirror Descent on a meta-objective given by the within-task minimum regularized

empirical risk. The interplay between the meta-algorithm and the inner algorithm plays a key role in our analysis. The latter is used to compute a good approximation of the meta-subgradient which is supplied to the former. A key novelty of our analysis is to show that, exploiting a closed form expression of the error on the meta-subgradients, we can automatically derive a cumulative error bound for the entire procedure. Our analysis holds also for more aggressive primal-dual online updates and it can be adapted to the statistical setting by online-to-batch arguments.

**Contributions.** Our contribution is threefold. First, we derive an efficient and theoretically grounded OWO Meta-Learning framework which is inspired by Multi-Task Learning (MTL). Our framework applies to a wide class of within-task algorithms and tasks' relationships. Second, we establish how our analysis can be converted to the statistical setting. Finally, we show how our general analysis can be directly applied to two important families of inner algorithms in which the meta-parameter is either a bias vector or a feature map shared across the tasks.

**Paper organization.** We start by introducing in Sec. 2 our OWO Meta-Learning setting. In Sec. 3 we recall some background material from primal-dual Online Learning. In Sec. 4 we outline the proposed method and we give a cumulative error bound for it. In Sec. 5 we show how the above analysis can be used to derive guarantees for our method in the statistical setting. In Sec. 6 we specify our framework to two important examples in which the tasks share a common bias vector or feature map. Finally, in Sec. 7 we report preliminary experiments with our method and in Sec. 8 we draw conclusions. Technical proofs are postponed to the appendix.

## 2  Setting

In this section we introduce the OWO Meta-Learning problem. We consider that the learner is facing a sequence of online tasks. Corresponding to each task, there is an input space $\mathcal{X}$, an output space $\mathcal{Y}$ and a dataset $Z = (z_i)_{i=1}^n = (x_i, y_i)_{i=1}^n \in (\mathcal{X} \times \mathcal{Y})^n$, which is observed *sequentially*. Online Learning aims to design an algorithm that makes predictions through time from past information. More precisely, at each step $i \in \{1, \ldots, n\}$: (a) a datapoint $z_i = (x_i, y_i)$ is observed, (b) the algorithm outputs a label $\hat{y}_i$, (c) the learner incurs the error $\ell_i(\hat{y}_i)$, where $\ell_i(\cdot) = \ell(\cdot, y_i)$ for a loss function $\ell$. To simplify our presentation, throughout we let $\mathcal{X} \subseteq \mathbb{R}^d$, $\mathcal{Y} \subseteq \mathbb{R}$ and we consider algorithms that perform linear predictions of the form $\hat{y}_i = \langle x_i, w_i \rangle$, where $(w_i)_{i=1}^n$ is a sequence of weight vectors updated by the algorithm and $\langle \cdot, \cdot \rangle$ denotes the standard inner product in $\mathbb{R}^d$. The goal is to bound the cumulative error of the algorithm, i.e. $\mathcal{E}_{\text{inner}}(Z) = \sum_{i=1}^n \ell_i(\langle x_i, w_i \rangle)$, with respect to (w.r.t.) the same quantity incurred by a vector $\hat{w} \in \mathbb{R}^d$ fixed in hindsight, i.e. $\sum_{i=1}^n \ell_i(\langle x_i, \hat{w} \rangle)$.

In the OWO Meta-Learning setting, we have a family of inner online algorithms identified by a meta-parameter $\theta$ belonging to a prescribed set $\Theta$ and the goal is to adapt $\theta$ to a sequence of learning tasks, in online fashion. Throughout this work, $\Theta$ will be a closed, convex, non-empty subset of an Euclidean space $\mathcal{M}$. The broad goal is to "transfer information" gained when learning previous tasks, in order to help learning future tasks. For this purpose, we propose a Meta-Learning procedure, acting across the tasks, which modifies the inner algorithm one task after another. More precisely, we let $T$ be the number of tasks and, for each task $t \in \{1, \ldots, T\}$ we let $Z_t = (x_{t,i}, y_{t,i})_{i=1}^n$ [1] be the corresponding data sequence. At each time $t$: (a) the meta-learner incrementally receives a task dataset $Z_t$, (b) it runs the inner online algorithm with meta-parameter $\theta_t$ on $Z_t$, returning the predictor vectors $(w_{\theta_t,i})_{i=1}^n$, (c) it incrementally incurs the errors $\ell_{t,i}(\langle x_{t,i}, w_{\theta_t,i} \rangle)$, where $\ell_{t,i}(\cdot) = \ell(\cdot, y_{t,i})$, (d) the meta-parameter (and consequently, the inner algorithm) is updated in $\theta_{t+1}$. Denoting by $\mathcal{E}_{\text{inner}}(Z_t, \theta_t)$ the cumulative error of the inner algorithm with meta-parameter $\theta_t$ on the dataset $Z_t$, the goal is to bound the error accumulated across the tasks, i.e.

$$\mathcal{E}_{\text{meta}}\big((Z_t)_{t=1}^T\big) = \sum_{t=1}^T \mathcal{E}_{\text{inner}}(Z_t, \theta_t) = \sum_{t=1}^T \sum_{i=1}^n \ell_{t,i}(\langle x_{t,i}, w_{\theta_t,i} \rangle), \tag{1}$$

w.r.t. the same quantity incurred by a sequence of tasks' vectors $(\hat{w}_t)_{t=1}^T$ fixed in hindsight, i.e. $\sum_{t=1}^T \sum_{i=1}^n \ell_{t,i}(\langle x_{t,i}, \hat{w}_t \rangle)$.

The setting we consider in the paper is inspired by previous work on Multi-Task Learning, such as [2, 10, 18]. To describe it, we use extended real-valued functions and, for any data sequence $Z$ and

meta-parameter $\theta \in \Theta$, we define the within-task minimum regularized empirical risk

$$\mathcal{L}_Z(\theta) = \min_{w \in \mathbb{R}^d} \mathcal{R}_Z(w) + \lambda f(w, \theta) \qquad \mathcal{R}_Z(w) = \frac{1}{n} \sum_{i=1}^{n} \ell_i(\langle x_i, w \rangle), \qquad (2)$$

where $\lambda > 0$ is a regularization parameter and $f$ is an appropriate complexity term ensuring the existence and the uniqueness of the above minimizer $\hat{w}_\theta$. Assuming the entire sequence $(Z_t)_{t=1}^T$ available in hindsight, introducing the notation $\mathcal{L}_t = \mathcal{L}_{Z_t}$, many MTL methods read as follows

$$\min_{\theta \in \mathcal{M}} \sum_{t=1}^{T} \mathcal{L}_t(\theta) + \eta F(\theta), \qquad (3)$$

where $\eta > 0$ is a meta-regularization parameter and $F$ is an appropriate meta-regularizer ensuring that the above minimum is attained. We stress that in our OWO Meta-Learning setting, the data are received sequentially, both within and across the tasks. The above formulation inspires us to take a within-task online algorithm that mimics well the (batch) objective in Eq. (2) and to define as meta-objectives for the online meta-algorithm the functions $(\mathcal{L}_t)_{t=1}^T$. Obviously, in this setting, the meta-objectives (and consequently their subgradients used by the meta-algorithm) are computed only up to an approximation error, depending on the specific properties of the inner algorithm we are using. We will show how to control and exploit this approximation error in the analysis.

In the sequel, for an Euclidean space $\mathcal{V}$, we let $\Gamma_0(\mathcal{V})$ to be the set of proper, closed and convex functions over $\mathcal{V}$ and, for any $f \in \Gamma_0(\mathcal{V})$, we denote by $\mathrm{Dom} f$ its domain (we refer to App. B and [31] for notions on convex analysis). In this work, we make the following standard assumptions in which we introduce two norms $\|\cdot\|_\theta$ and $\|\!|\cdot|\!\|$ that will be specified in two applications below.

**Assumption 1** (Loss and regularizer). *Let $\ell(\cdot, y)$ be a convex and closed real-valued function for any $y \in \mathcal{Y}$ and let $f \in \Gamma_0(\mathbb{R}^d \times \mathcal{M})$ be such that, for any $\theta \in \Theta$, $f(\cdot, \theta)$ is $1$-strongly convex w.r.t. a norm $\|\cdot\|_\theta$ over $\mathbb{R}^d$, $\inf_{w \in \mathbb{R}^d} f(w, \theta) = 0$ and, for any $\theta \notin \Theta$, $\mathrm{Dom} f(\cdot, \theta) = \emptyset$.*

**Assumption 2** (Meta-regularizer). *Let $F$ be a closed and $1$-strongly convex function w.r.t. a norm $\|\!|\cdot|\!\|$ over $\mathcal{M}$ such that $\inf_{\theta \in \mathcal{M}} F(\theta) = 0$ and $\mathrm{Dom} F = \Theta$.*

Notice that the norm w.r.t. which the function $f(\cdot, \theta)$ is assumed to be strongly convex may vary with $\theta$. Moreover, under Asm. 1, $\mathrm{Dom} \mathcal{L}_Z = \Theta$ and, since $\mathcal{L}_Z$ is defined as the partial minimum of a function in $\Gamma_0(\mathbb{R}^d \times \mathcal{M})$, $\mathcal{L}_Z \in \Gamma_0(\mathcal{M})$. This property supports the choice of this function as the meta-objective for our meta-algorithm. Finally, by Lemma 29 in App. B, Asm. 1 and Asm. 2 ensure the existence and the uniqueness of the minimizers in Eq. (2) and Eq. (3).

We conclude this section by giving two examples included in the framework above. The first one is inspired by the MTL variance regularizer in [14], while the second example, which can be easily extended to more general MTL regularizers such as in [2, 10, 27, 28], relates to the MTL trace norm regularizer. As we will see in the following, in the first example the tasks' predictors are encouraged to stay close to a common bias vector, in the second example they are encouraged to lie in the range of a low-rank feature map. In order to describe these examples we require some additional notation. We let $\|\cdot\|_2, \|\cdot\|_F, \|\cdot\|_{\mathrm{Tr}}, \|\cdot\|_\infty$, be the Euclidean, Frobenius, trace, and operator norm, respectively. We also let "$\cdot^\dagger$" be the pseudo-inverse, $\mathrm{Tr}(\cdot)$ be the trace, $\mathrm{Ran}(\cdot)$ be the range and $\mathbb{S}^d$ (resp. $\mathbb{S}^d_+$) be the set of symmetric (resp. positive semi-definite) matrices in $\mathbb{R}^{d \times d}$. Finally, $\iota_\mathcal{S}$ denotes the indicator function of the set $\mathcal{S}$, taking value $0$ when the argument belongs to $\mathcal{S}$ and $+\infty$ otherwise.

**Example 1** (Bias). *We choose $\mathcal{M} = \Theta = \mathbb{R}^d$, $F(\cdot) = \frac{1}{2}\|\cdot\|_2^2$, satisfying Asm. 2 with $\|\!|\cdot|\!\| = \|\cdot\|_2$, and $f(\cdot, \theta) = \frac{1}{2}\|\cdot - \theta\|_2^2$, satisfying Asm. 1 with $\|\cdot\|_\theta = \|\cdot\|_2$ for every $\theta \in \mathbb{R}^d$.*

**Example 2** (Feature Map). *We choose $\mathcal{M} = \mathbb{S}^d$ and $\Theta = \mathcal{S}$, where $\mathcal{S} = \{\theta \in \mathbb{S}^d_+ : \mathrm{Tr}(\theta) \leq 1\}$. For a fixed $\theta_0 \in \mathcal{S}$, we set $F(\cdot) = \frac{1}{2}\|\cdot - \theta_0\|_F^2 + \iota_\mathcal{S}(\cdot)$, satisfying Asm. 2 with $\|\!|\cdot|\!\| = \|\cdot\|_F$, and $f(\cdot, \theta) = \frac{1}{2}\langle \cdot, \theta^\dagger \cdot \rangle + \iota_{\mathrm{Ran}(\theta)}(\cdot) + \iota_\mathcal{S}(\theta)$, satisfying Asm. 1 with $\|\cdot\|_\theta = \sqrt{\langle \cdot, \theta^\dagger \cdot \rangle}$ for any $\theta \in \mathcal{S}$.*

We will return to these examples in Sec. 6, specializing our method and our analysis to these settings.

## 3 Preliminaries: primal-dual Online Learning

Our OWO Meta-Learning method consists in the application of two nested primal-dual online algorithms, one operating within the tasks and another across the tasks. In particular, even though our

---

**Algorithm 1** Primal-dual online algorithm – online Mirror Descent

---

**Input** $(g_m)_{m=1}^M, (A_m)_{m=1}^M, (c_m)_{m=1}^M, (\epsilon_m)_{m=1}^M, r$ as described in the text

**Initialization** $\alpha_1 = (), v_1 = \nabla r^*(0) \in \text{Dom } r$

**For** $m = 1$ to $M$

  Receive $g_m, A_m, c_{m+1}, \epsilon_m$

  Suffer $g_m(A_m v_m)$ and compute $\alpha'_m \in \partial_{\epsilon_m} g_m(A_m v_m)$

  Update $\alpha_{m+1} = (\alpha_m, \alpha'_m)$

  Define $v_{m+1} = \nabla r^*\big(-1/c_{m+1} \sum_{j=1}^m A_j^* \alpha_{m+1,j}\big) \in \text{Dom } r$

**Return** $(\alpha_m)_{m=1}^{M+1}, (v_m)_{m=1}^{M+1}$

---

analysis holds also for more aggressive schemes, in this work, we consider online Mirror Descent algorithm. In this section we briefly recall some material from the primal-dual interpretation of this algorithm that will be used in our subsequent analysis. The material of this section is an adaptation from [37, 33, 38, 36, 35]; we refer to App. C for a more detailed presentation.

Online Mirror Descent algorithm on a (primal) problem can be derived from the following primal-dual framework in which we introduce an appropriate dual algorithm. Specifically, at each iteration $m \in \{1, \ldots, M\}$, we consider the following instantaneous primal optimization problem

$$\hat{P}_{m+1} = \inf_{v \in \mathcal{V}} P_{m+1}(v) \qquad P_{m+1}(v) = \sum_{j=1}^m g_j(A_j v) + c_m r(v) \tag{4}$$

where $\mathcal{V}$ is an Euclidean space, $c_m > 0$, $r \in \Gamma_0(\mathcal{V})$ is a 1-strongly convex function w.r.t. a norm $\|\cdot\|$ over $\mathcal{V}$ (with dual norm $\|\cdot\|_*$) such that $\inf_{v \in \mathcal{V}} r(v) = 0$, for any $j \in \{1, \ldots, M\}$, letting $\mathcal{V}_j$ an Euclidean space, $g_j \in \Gamma_0(\mathcal{V}_j)$ and $A_j : \mathcal{V} \to \mathcal{V}_j$ is a linear operator with adjoint $A_j^*$. As explained in App. C, the corresponding dual problem is given by

$$\hat{D}_{m+1} = \inf_{\alpha \in \mathcal{V}_1 \times \cdots \times \mathcal{V}_m} D_{m+1}(\alpha) \qquad D_{m+1}(\alpha) = \sum_{j=1}^m g_j^*(\alpha_j) + c_m r^*\Big(-\frac{1}{c_m} \sum_{j=1}^m A_j^* \alpha_j\Big), \tag{5}$$

where $g_j^*$ and $r^*$ are respectively the conjugate functions of $g_j$ and $r$. After this, we define the dual scheme in which the dual variable $\alpha_{m+1}$ is updated by a greedy coordinate descent approach on the dual, setting $\alpha_{m+1} = (\alpha_m, \alpha'_m)$, where $\alpha'_m \in \partial_{\epsilon_m} g_m(A_m v_m)$ is an $\epsilon_m$-subgradient of $g_m$ at $A_m v_m$ and $v_m$ is the current primal iteration. The primal variable is then updated from the dual one by a variant of the Karush–Kuhn–Tucker (KKT) conditions, providing its belonging to Dom $r$, see Alg. 1. In this paper, following [36], we refer to such a scheme as lazy online Mirror Descent. However, the term linearized Follow-The-Regularized-Leader is historically more accurate. We recall also that such a scheme includes many well-known algorithms, when one properly specifies the complexity term $r$. The behavior of Alg. 1 is analyzed in the next result which will be a key tool for our analysis.

**Theorem 1** (Dual optimality gap for Alg. 1). *Let $(v_m)_{m=1}^M$ be the primal iterates returned by Alg. 1 when applied to the generic problem in Eq. (4) and let $\Delta_{\text{Dual}} = D_{M+1}(\alpha_{M+1}) - \hat{D}_{M+1}$ be the corresponding (non-negative) dual optimality gap at the last dual iterate $\alpha_{M+1}$ of the algorithm.*

  *1. If, for any $m \in \{1, \ldots, M\}$, $c_{m+1} \geq c_m$, then,*

$$\Delta_{\text{Dual}} \leq -\sum_{m=1}^M g_m(A_m v_m) + \hat{P}_{M+1} + \frac{1}{2} \sum_{m=1}^M \frac{1}{c_m} \big\| A_m^* \alpha'_m \big\|_*^2 + \sum_{m=1}^M \epsilon_m.$$

  *2. If, for any $m \in \{1, \ldots, M\}$, $c_m = \sum_{j=1}^m \lambda_j$ for some $\lambda_j > 0$, then,*

$$\Delta_{\text{Dual}} \leq -\sum_{m=1}^M \Big\{ g_m(A_m v_m) + \lambda_m r(v_m) \Big\} + \hat{P}_{M+1} + \frac{1}{2} \sum_{m=1}^M \frac{1}{c_m} \big\| A_m^* \alpha'_m \big\|_*^2 + \sum_{m=1}^M \epsilon_m.$$

The first (resp. second) inequality in Thm. 1 links the dual optimality gap of the last dual iterate generated by Alg. 1, with the (resp. regularized) cumulative error of the corresponding primal iterates. Note that this result can be readily used to bound the cumulative error (resp. its regularized version) of Alg. 1 by the batch regularized comparative $\hat{P}_{M+1}$ and additional terms. In the following section, we will make use of the above theorem in order to analyze our OWO Meta-Learning method.

| **Algorithm 2** Within-task algorithm | **Algorithm 3** Meta-algorithm |
|---|---|
| **Input** $\lambda > 0, \theta \in \Theta, Z = (z_i)_{i=1}^n$ | **Input** $\eta > 0, (Z_t)_{t=1}^T$ |
| **Initialization** $s_{\theta,1} = (), w_{\theta,1} = \nabla f(\cdot, \theta)^*(0)$ | **Initialization** $\theta_1 = \nabla F^*(0)$ |
| **For** $i = 1$ to $n$ | **For** $t = 1$ to $T$ |
|   Receive the datapoint $z_i = (x_i, y_i)$ |   Receive incrementally the dataset $Z_t$ |
|   Compute $s'_{\theta,i} \in \partial \ell_i(\langle x_i, w_{\theta,i} \rangle) \subseteq \mathbb{R}$ |   Run Alg. 2 with $\theta_t$ over $Z_t$ |
|   Define $(s_{\theta,i+1})_i = s'_{\theta,i}, \gamma_i = \lambda(i+1)$ |   Compute $s_{\theta_t, n+1}$ |
|   Update $w_{\theta,i+1} = \nabla f(\cdot, \theta)^* \left( -1/\gamma_i \sum_{j=1}^i x_j s'_{\theta,j} \right)$ |   Compute $\nabla'_{\theta_t}$ as in Prop. 3 using $s_{\theta_t, n+1}$ |
| |   Update $\theta_{t+1} = \nabla F^* \left( -1/\eta \sum_{j=1}^t \nabla'_{\theta_j} \right)$ |
| **Return** $(w_{\theta,i})_{i=1}^{n+1}, \bar{w}_\theta = \frac{1}{n}\sum_{i=1}^n w_{\theta,i}, s_{\theta,n+1}$ | **Return** $(\theta_t)_{t=1}^{T+1}, \bar{\theta} = \frac{1}{T}\sum_{t=1}^T \theta_t$ |

## 4 Method

In this section we present the proposed OWO Meta-Learning method and we establish a (regularized) cumulative error bound for it. As anticipated in Sec. 2, the method consists in the application of Alg. 1 both to the (non-normalized) within-task problem in Eq. (2) and to the across-tasks problem in Eq. (3), corresponding, as we will show in the following, to Alg. 2 and Alg. 3, respectively. In order to analyze our method, we start from studying the behavior of the inner Alg. 2.

**Proposition 2** (Dual optimality gap for the inner Alg. 2). *Let Asm. 1 hold. Then, Alg. 2 coincides with Alg. 1 applied to the non-normalized within-task problem in Eq. (2). As a consequence, introducing the regularized cumulative error of the iterates generated by Alg. 2,*

$$\mathcal{E}_{\text{inner}}^{\text{reg}}(Z, \theta) = \sum_{i=1}^n \left\{ \ell_i(\langle x_i, w_{\theta,i} \rangle) + \lambda f(w_{\theta,i}, \theta) \right\}, \tag{6}$$

*where $w_{\theta,i} \in \text{Dom} f(\cdot, \theta)$ for any $i \in \{1, \dots, n\}$, the following upper bound for the associated dual optimality gap $\Delta_{\text{Dual}}$ introduced in Thm. 1 holds*

$$\Delta_{\text{Dual}} \leq \epsilon_\theta \qquad \epsilon_\theta = -\left( \mathcal{E}_{\text{inner}}^{\text{reg}}(Z, \theta) - n\mathcal{L}_Z(\theta) \right) + \frac{1}{2\lambda} \sum_{i=1}^n \frac{1}{i} \left\| x_i s'_{\theta,i} \right\|_{\theta,*}^2. \tag{7}$$

**Proof.** The inner Alg. 2 coincides with Alg. 1 applied to the non-normalized within-task problem in Eq. (2), once one makes the identifications $\alpha'_m \rightsquigarrow s'_{\theta,i}$ for the (exact) subgradients and realizes that the non-normalized within-task problem in Eq. (2) is of the form in Eq. (4) with

$m \rightsquigarrow M, \; j \rightsquigarrow i, \; M \rightsquigarrow n, \; v \rightsquigarrow w, \; \mathcal{V} \rightsquigarrow \mathbb{R}^d, \; g_j \rightsquigarrow \ell_i, \; A_j \rightsquigarrow x_i^\top, \; c_m \rightsquigarrow n\lambda, \; r(\cdot) \rightsquigarrow f(\cdot, \theta).$
Now, the bound in the statement directly derives from the second point of Thm. 1. ∎

Since $\Delta_{\text{Dual}} \geq 0$, by moving the terms and normalizing by the number of points $n$, the above result tells us that, when the terms $\|x_i s'_{\theta,i}\|_{\theta,*}^2$ are bounded, for an appropriate choice of $\lambda$, the inner algorithm attempts to mimic the function $\mathcal{L}_Z$ in Eq. (2), as the number of points $n$ increases. The method we propose in this work relies on the application of Alg. 1 also to the meta-problem in Eq. (3) as the tasks are sequentially observed, using the functions $(\mathcal{L}_t)_{t=1}^T$ as meta-objectives. A key difficulty here is that the meta-objective is defined via the inner batch problem in Eq. (2), hence it is not available exactly but it is only approximately approached by the within-task online algorithm. From a practical point of view, this means that in this case, differently from the inner algorithm, the resulting meta-algorithm has to deal with an error on the meta-subgradients at each iteration. Our next result describes how, leveraging on the dual optimality gap for the inner Alg. 2, we can compute an $\epsilon$-subgradient of the meta-objective, where $\epsilon$ is (up to normalization) the value stated in Prop. 2. This will allow us to develop an efficient method which is computationally appealing and fully online.

**Proposition 3** (Computation of an $\epsilon$-subgradient of $\mathcal{L}_Z$). *Let Asm. 1 hold and let $s_{\theta,n+1}$ be the output of Alg. 2 with $\theta \in \Theta$ over the dataset $Z$. Let $\nabla_\theta \in \partial\{-D_{n+1}(s_{\theta,n+1}, \cdot)\}(\theta)$, where*

$$D_{n+1}(s, \theta) = \sum_{i=1}^n \ell_i^*(s_i) + \lambda n f^*(\cdot, \theta)\left( -\frac{1}{\lambda n} \sum_{i=1}^n x_i s_i \right) \qquad s \in \mathbb{R}^n \tag{8}$$

is the dual of the non-normalized *Eq.* (2). *Then,* $\nabla'_\theta = \nabla_\theta / n \in \partial_{\epsilon_\theta/n} \mathcal{L}_Z(\theta)$, *with $\epsilon_\theta$ as in Prop. 2.*

The proof of the above statement is reported in App. D. It is based on rewriting the meta-objective as $\mathcal{L}_Z(\theta) = 1/n \max_{s \in \mathbb{R}^n} \{-D_{n+1}(s, \theta)\}$ (by strong duality, see Lemma 34 in App. D) and it essentially exploits Prop. 2, according to which, the last dual iteration $s_{\theta,n+1}$ returned by Alg. 2 is an $\epsilon_\theta$-maximizer of the dual objective $-D_{n+1}(\cdot, \theta)$. We remark that the procedure described above to compute an $\epsilon$-subgradient has been already used in our work [11] for the statistical setting in Ex. 1. Here, with a different proof technique, we show that it can be extended also to more general inner regularizers. Leveraging on the form of the error on the meta-subgradients in Prop. 3, we now show how we can automatically deduce a (regularized) cumulative error bound for the entire procedure.

**Theorem 4** (Cumulative error bound). *Let Asm. 1 and Asm. 2 hold. Then, Alg. 3 coincides with Alg. 1 applied to the outer-tasks problem in Eq.* (3)*. As a consequence, introducing the regularized cumulative error for the iterates generated by the combination of Alg. 2 and Alg. 3,*

$$\mathcal{E}_{\text{meta}}^{\text{reg}}\big((Z_t)_{t=1}^T\big) = \sum_{t=1}^T \mathcal{E}_{\text{inner}}^{\text{reg}}(Z_t, \theta_t) = \sum_{t=1}^T \sum_{i=1}^n \Big\{ \ell_{t,i}(\langle x_{t,i}, w_{\theta_t,i} \rangle) + \lambda f(w_{\theta_t,i}, \theta_t) \Big\}, \quad (9)$$

*where $\theta_t \in \Theta$ for any $t \in \{1, \ldots, T\}$, for any sequence of vectors $(\hat{w}_t)_{t=1}^T$ in $\mathbb{R}^d$ and any $\theta \in \Theta$ such that $f(\hat{w}_t, \theta) < +\infty$ for any $t \in \{1, \ldots, T\}$, the following upper bound holds*

$$\mathcal{E}_{\text{meta}}^{\text{reg}}\big((Z_t)_{t=1}^T\big) \leq nT \left( \frac{1}{T} \sum_{t=1}^T \mathcal{R}_{Z_t}(\hat{w}_t) + \frac{\lambda}{T} \sum_{t=1}^T f(\hat{w}_t, \theta) + \frac{1}{2\lambda nT} \sum_{t=1}^T \sum_{i=1}^n \frac{1}{i} \| x_{t,i} s'_{\theta_t, i} \|_{\theta_t,*}^2 \right.$$
$$\left. + \frac{\eta F(\theta)}{T} + \frac{1}{2\eta T} \sum_{t=1}^T \left\| \nabla'_{\theta_t} \right\|_*^2 \right).$$

**Proof.** The meta-algorithm in Alg. 3 coincides with Alg. 1 applied to the outer-tasks problem in Eq. (3), once one makes the identifications $\alpha'_m \rightsquigarrow \nabla'_{\theta_t}$ for the (approximated) subgradients and realizes that the outer-tasks problem in Eq. (3) is of the form in Eq. (4) with

$$m \rightsquigarrow M, \ j \rightsquigarrow t, \ M \rightsquigarrow T, \ v \rightsquigarrow \theta, \ \mathcal{V} \rightsquigarrow \Theta, \ g_j \rightsquigarrow \mathcal{L}_t, \ A_j \rightsquigarrow I, \ c_m \rightsquigarrow \eta, \ r \rightsquigarrow F.$$

As a consequence, denoting by $\Delta_{\text{Dual}}$ the associated dual optimality gap introduced in Thm. 1, specializing the first point of Thm. 1 to this setting and exploiting the fact $\Delta_{\text{Dual}} \geq 0$, we get

$$0 \leq - \sum_{t=1}^T \mathcal{L}_t(\theta_t) + \min_{\theta \in \Theta} \left\{ \sum_{t=1}^T \mathcal{L}_t(\theta) + \eta F(\theta) \right\} + \frac{1}{2\eta} \sum_{t=1}^T \left\| \nabla'_{\theta_t} \right\|_*^2 + \frac{1}{n} \sum_{t=1}^T \epsilon_{\theta_t}. \quad (10)$$

Substituting the closed form of $\epsilon_{\theta_t}$ in Prop. 2 (applied to the task $t$) into Eq. (10), one immediately observes that the term $\sum_{t=1}^T \mathcal{L}_t(\theta_t)$ erases. The desired statement then directly follows by rearranging the remaining terms, using the definition of $(\mathcal{L}_t)_{t=1}^T$ and multiplying by the number of points $n$. ∎

When the inputs are bounded and both the inner loss and meta-objective are Lipschitz w.r.t. the associated norms (as we will see for Ex. 1), the terms $\left\| \nabla'_{\theta_t} \right\|_*^2$ and $\| x_{t,i} s'_{\theta_t, i} \|_{\theta_t,*}^2$ can be upper bounded by a constant. In this case, for an appropriate choice of $\lambda$ and $\eta$, we recover a reasonable rate $\tilde{\mathcal{O}}(1/\sqrt{n}) + \mathcal{O}(1/\sqrt{T})$. However, when the bounds on $\left\| \nabla'_{\theta_t} \right\|_*^2$ hide a dependency w.r.t. $\lambda$ or $n$ (as we will see for Ex. 2), the bound must be accordingly analyzed.

## 5 Adaptation to the statistical setting

In this section we present guarantees for our method in the statistical setting. Following the framework outlined in [6, 23, 26] we assume that, for any $t \in \{1, \ldots, T\}$, the within-task dataset $Z_t$ is an independently identically distributed (i.i.d.) sample from a distribution (task) $\mu_t$, and in turn the tasks $(\mu_t)_{t=1}^T$ are an i.i.d. sample from a meta-distribution $\rho$. The estimator we consider here is $\bar{w}_{\bar{\theta}} = \frac{1}{n} \sum_{i=1}^n w_{\theta,i}$, the average of the iterates resulting from applying Alg. 2 to a test dataset $Z$ with meta-parameter $\bar{\theta} = \frac{1}{T} \sum_{t=1}^T \theta_t$, the average of the meta-parameters returned by our online meta-algorithm in Alg. 3 applied to the training datasets $(Z_t)_{t=1}^T$. We wish to study the performance of such an estimator in expectation w.r.t. the tasks sampled from the environment $\rho$.

Formally, for any $\mu \sim \rho$, we require that the corresponding true risk $\mathcal{R}_\mu(w) = \mathbb{E}_{(x,y)\sim\mu}\ell(\langle x, w\rangle, y)$ admits minimizers over the entire space $\mathbb{R}^d$ and we denote by $w_\mu$ the minimum norm one. With these ingredients, we introduce the oracle $\mathcal{E}_\rho = \mathbb{E}_{\mu\sim\rho}\mathcal{R}_\mu(w_\mu)$, representing the expected minimum error over the environment of tasks, and, introducing the *transfer risk* of the estimator $\bar{w}_{\bar\theta}$:

$$\mathcal{E}_{\text{stat}}(\bar{w}_{\bar\theta}) = \mathbb{E}_{\mu\sim\rho}\,\mathbb{E}_{Z\sim\mu^n}\,\mathcal{R}_\mu(\bar{w}_{\bar\theta}(Z)), \tag{11}$$

we give a bound on it w.r.t. the oracle $\mathcal{E}_\rho$. This is described in the following theorem.

**Theorem 5** (Transfer risk bound). *Let the same assumptions in Thm. 4 hold in the i.i.d. statistical setting. Then, introducing the regularized transfer risk of the average $\bar{w}_{\bar\theta}$ of the iterates resulting from the combination of Alg. 2 and Alg. 3,*

$$\mathcal{E}_{\text{stat}}^{\text{reg}}(\bar{w}_{\bar\theta}) = \mathbb{E}_{\mu\sim\rho}\,\mathbb{E}_{Z\sim\mu^n}\left[\mathcal{R}_\mu(\bar{w}_{\bar\theta}(Z)) + \lambda f(\bar{w}_{\bar\theta}(Z), \bar\theta)\right],$$

*for any $\theta \in \Theta$ such that $\mathbb{E}_{\mu\sim\rho}f(w_\mu, \theta) < +\infty$, the following upper bound holds in expectation w.r.t. the sampling of the datasets $(Z_t)_{t=1}^T$*

$$\mathbb{E}\,\mathcal{E}_{\text{stat}}^{\text{reg}}(\bar{w}_{\bar\theta}) \leq \mathcal{E}_\rho + \lambda\,\mathbb{E}_{\mu\sim\rho}f(w_\mu, \theta) + \frac{1}{2\lambda nT}\,\mathbb{E}\sum_{t=1}^T\sum_{i=1}^n\frac{1}{i}\big\|x_{t,i}s'_{\theta_t,i}\big\|^2_{\theta_t,*}$$

$$+ \frac{\eta F(\theta)}{T} + \frac{1}{2\eta T}\,\mathbb{E}\sum_{t=1}^T\big\|\big|\nabla'_{\theta_t}\big|\big\|^2_* + \mathbb{E}\,\mathbb{E}_{\mu\sim\rho}\,\mathbb{E}_{Z\sim\mu^n}\frac{1}{2\lambda n}\sum_{i=1}^n\frac{1}{i}\big\|x_is'_{\bar\theta,i}\big\|^2_{\bar\theta,*}.$$

The proof of the statement above is reported in App. E. It exploits the *regularized* cumulative error bound given in Thm. 4 for our Meta-Learning procedure and two nested online-to-batch conversion steps [9, 22], one within-task and one across-tasks. The bound above is composed by the expectation of the terms comparing in Thm. 4 plus an additional term. Such a term comes out from the online-to-batch conversion and, as we will see in the sequel, it does not affect the general behavior of the bound. Finally, we observe that, differently from [1, Thm. 6.1] and [3, Thm. 3.3], the theorem above holds for the average of the meta-parameters $(\theta_t)_{t=1}^T$ returned by our meta-algorithm (not for a meta-parameter randomly sampled from the pool) and, consequently, it does not require their memorization or the introduction of additional randomization to the process. In the following section we will show that specializing Thm. 4 and Thm. 5 to Ex. 1 and Ex. 2, we will get meaningful bounds.

## 6 Examples

In this section we specify our framework to Ex. 1 and Ex. 2 outlined at the end of Sec. 2. In order to do this, we require the following assumption, which is for instance satisfied by the absolute loss $\ell(\hat{y}, y) = |\hat{y} - y|$ and the hinge loss $\ell(\hat{y}, y) = \max\{0, 1 - y\hat{y}\}$, where $y, \hat{y} \in \mathcal{Y}$.

**Assumption 3** (Lipschitz Loss). *Let $\ell(\cdot, y)$ be $L$-Lipschitz for any $y \in \mathcal{Y}$.*

Below, for any task $t \in \{1, \dots, T\}$, we let the input covariance matrices $C_t = \frac{1}{n}\sum_{i=1}^n x_{t,i}x_{t,i}^\top$, $\hat{C}_t = \sum_{i=1}^n\frac{1}{i}x_{t,i}x_{t,i}^\top$, $C^{\text{tot}} = \frac{1}{T}\sum_{t=1}^T C_t$ and $\hat{C}^{\text{tot}} = \frac{1}{T}\sum_{t=1}^T \hat{C}_t$. We also use the notation $\|C^{\text{tot}}\|_{\infty,a} = \frac{1}{T}\sum_{t=1}^T\|C_t\|_\infty^a$ with $a = 1, 2$ and, in the statistical setting, we let $C_\rho = \mathbb{E}_{\mu\sim\rho}\,\mathbb{E}_{(x,y)\sim\mu}xx^\top$.

**Bias.** In App. G we report the adaptation of our method in Alg. 2 and Alg. 3 (cf. Alg. 5 and Alg. 6) and we specify Thm. 4 and Thm. 5 (cf. Cor. 40 and Cor. 42) to Ex. 1. In such a case, the resulting inner algorithm coincides with online Subgradient Descent on the regularized empirical risk and, similarly, the resulting meta-algorithm coincides with online Subgradient Descent (with approximated subgradients) on the meta-objectives $(\mathcal{L}_t)_{t=1}^T$. We thus recover the method in [11] with a slightly different choice of the inner algorithm step size. Our results (see App. G.4.2) are in line with [11], where we present the same bound in Cor. 42 with slightly worse constants.

**Feature map.** In App. H.1 we report the adaptation of our method in Alg. 2 and Alg. 3 (cf. Alg. 7 and Alg. 8) to Ex. 2. In this case, the resulting inner algorithm coincides with a pre-conditioned variant of online Subgradient Descent on the regularized empirical risk and the resulting meta-algorithm coincides with a lazy variant of online Subgradient Descent (with approximate subgradients) on the meta-objectives $(\mathcal{L}_t)_{t=1}^T$, projected on the set $\mathcal{S}$. The meta-algorithm we retrieve is a slightly different version of the algorithm we propose in [12] for an OWB statistical framework.

Our next result specifies the cumulative error bound in Thm. 4 to Ex. 2. The proof is in App. H.2.

**Corollary 6** (Cumulative error bound, feature map). *Let Asm. 3 hold, consider the setting in Thm. 4 applied to Ex. 2 and let $\hat{C}^{\mathrm{tot}}_{\theta_{1:T}} = \frac{1}{T}\sum_{t=1}^{T}\theta_t\hat{C}_t$. Then, for any sequence of vectors $(\hat{w}_t)_{t=1}^{T}$ in $\mathbb{R}^d$, introducing $\hat{B} = \frac{1}{T}\sum_{t=1}^{T}\hat{w}_t\hat{w}_t^{\top}$, for any $\theta \in \mathcal{S}$ such that $\mathrm{Ran}(\hat{B}) \subseteq \mathrm{Ran}(\theta)$, the following bound holds for our method with an appropriate choice of hyper-parameters*

$$\mathcal{E}^{\mathrm{reg}}_{\mathrm{meta}}\big((Z_t)_{t=1}^{T}\big) \leq nT\left(\frac{1}{T}\sum_{t=1}^{T}\mathcal{R}_{Z_t}(\hat{w}_t) + L\sqrt{\mathrm{Tr}(\theta^{\dagger}\hat{B})\left(\frac{\mathrm{Tr}(\hat{C}^{\mathrm{tot}}_{\theta_{1:T}})}{n} + \|\theta - \theta_0\|_F\sqrt{\frac{\|C^{\mathrm{tot}}\|_{\infty,2}}{T}}\right)}\right).$$

The next result specifies the transfer risk bound in Thm. 5 to Ex. 2. The proof is in App. H.3.

**Corollary 7** (Transfer risk bound, feature map). *Let Asm. 3 hold and consider the setting in Thm. 5 applied to Ex. 2. Then, in expectation w.r.t. the sampling of the datasets $(Z_t)_{t=1}^{T}$, introducing $B_\rho = \mathbb{E}_{\mu\sim\rho}w_\mu w_\mu^{\top}$, for any $\theta \in \mathcal{S}$ such that $\mathrm{Ran}(B_\rho) \subseteq \mathrm{Ran}(\theta)$, the following bound holds for our method with an appropriate choice of hyper-parameters*

$$\mathbb{E}\,\mathcal{E}^{\mathrm{reg}}_{\mathrm{stat}}(\bar{w}_{\bar{\theta}}) \leq \mathcal{E}_\rho + L\sqrt{\mathrm{Tr}(\theta^{\dagger}B_\rho)\left(\frac{2\big(\log(n)+1\big)\,\mathrm{Tr}(\mathbb{E}\,\bar{\theta}C_\rho)}{n} + \|\theta - \theta_0\|_F\sqrt{\frac{\mathbb{E}\,\|C^{\mathrm{tot}}\|_{\infty,2}}{T}}\right)}.$$

We now analyze the statistical setting. Following [12, 26, 25] we study whether, as the number of tasks grows, our method mimics the performance of the inner algorithm with the best feature map in hindsight (*oracle*, see App. H.4.1) for any task. We note that, once fixed in an appropriate way the meta-parameter $\theta$ in the statement (hence, the hyper-parameters), the above bound in Cor. 7 becomes comparable to the bound for the best feature map in hindsight, see the discussion in App. H.4.2. Hence, we recover the same conclusion: there is an advantage in using the feature map found by our Meta-Learning method w.r.t. solving each task independently when $\|C_\rho\|_\infty$ is small (the inputs are high-dimensional, for instance) and $B_\rho$ is low-rank (the tasks share a low dimensional representation). In addition, note that the bound in Cor. 7 converges, as the number of tasks grow, to the oracle at a rate of $\mathcal{O}(T^{-1/4})$, whereas the corresponding bounds for the bias example (cf. Cor. 40 and Cor. 42 in App. G) yield the faster $\mathcal{O}(T^{-1/2})$ rate, suggesting that feature learning is a more difficult problem than bias learning. Regarding the non-statistical setting, the bound in Cor. 6 is less clear to interpret because of the presence of the modified version of the inputs' covariance matrix $\hat{C}^{\mathrm{tot}}_{\theta_{1:T}}$. Future work may be devoted to investigate this point, which could be either an artifact of our analysis or due to some intrinsic characteristics of the problem we are considering.

# 7 Experiments

We present preliminary experiments with our OWO Meta-Learning method (ONL-ONL)[2] in the statistical setting of Ex. 2. In all experiments, the hyper-parameters $\lambda$ and $\eta$ were chosen by a meta-validation procedure (see App. I for more details) and we fixed $\theta_0 = I/d$ for the meta-algorithm in Alg. 8. We compared ONL-ONL to the modified batch-online (BAT-ONL) variant, where the meta-subgradients in the meta-training phase are computed with higher accuracy by a convex solver (such as CVX), to Independent-Task Learning (ITL), i.e. running the inner Alg. 7 with the feature map $\theta = I/d$ for each task, and, in the synthetic data experiment, to the Oracle, i.e. running the inner Alg. 7 with the best feature map in hindsight for each task, see App. H.4.1.

**Synthetic data.** We considered the regression setting with the absolute loss function. We generated $T_{\mathrm{tot}} = 3600$ tasks. For each task, the corresponding dataset $(x_i, y_i)_{i=1}^{n_{\mathrm{tot}}}$ of $n_{\mathrm{tot}} = 80$ points was generated according to the linear equation $y = \langle x, w_\mu\rangle + \epsilon$, with $x$ sampled uniformly on the unit sphere in $\mathbb{R}^d$ with $d = 20$ and $\epsilon$ sampled from a Gaussian distribution, $\epsilon \sim \mathcal{G}(0, 0.2)$. The tasks' predictors $w_\mu$ were generated as $w_\mu = P\tilde{w}_\mu$ with the components of $\tilde{w}_\mu \in \mathbb{R}^{d/5}$ sampled from $\mathcal{G}(0, 1)$ and then $\tilde{w}_\mu$ normalized to have unit norm, with $P \in \mathbb{R}^{d\times d/5}$ a matrix with orthonormal columns. In this setting, the operator norm of the inputs' covariance matrix $C_\rho$ is small (equal to $1/d$) and the weight vectors' covariance matrix $B_\rho$ is low-rank, a favorable setting for our method, according to Cor. 7. Looking at the results in Fig. 1 (Left), we can state that, in this setting, our method outperforms ITL and it tends to the Oracle as the number of training tasks increases. Moreover, the

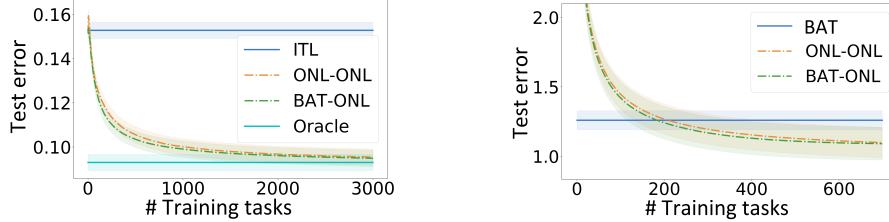

Figure 1: Synthetic data (Left) and Movielens-100k dataset (Right). Performance of different methods as the number of training tasks increases. The results are averaged over 10 runs/splits of the data.

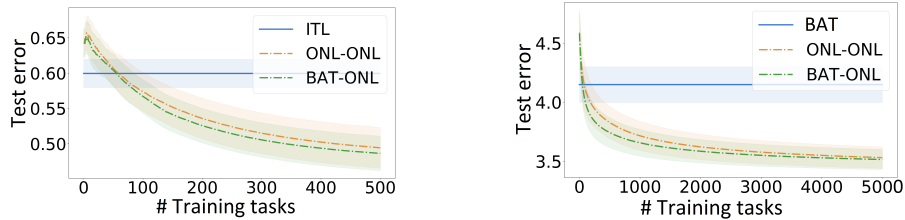

Figure 2: Mini-Wiki dataset (Left) and Jester-1 dataset (Right). Performance of different methods as the number of training tasks increases. The results are averaged over 10 splits of the data.

performance of ONL-ONL and BAT-ONL are comparable, suggesting that our approximation of the meta-subgradients is an effective way to keep the entire process fully online.

**Real data.** We further validated the proposed method on three real datasets: 1) the Movielens-100k dataset[3], containing the ratings of different users to different movies 2) the Mini-Wiki dataset from [3], containing sentences from Wikipedia pages and 3) the Jester-1 dataset[4], containing the ratings of different users to different jokes. For the Movielens-100k and the Jester-1 datasets we considered each user as a task and each movie/joke as a point. Specifically, we casted each task as a regression problem where the labels are the ratings of the users and the raw features are simply the index of the movie/joke (i.e. a matrix completion setting where the input dimension $d$ coincides with the number of points). For the Mini-Wiki dataset we casted each task as a multi-class classification problem where the labels are the Wikipedia pages and the features are vectors with dimension $d = 50$. After processing the data, we ended with a total number of $T_{\mathrm{tot}} = 939, 813, 5700$ tasks and $n_{\mathrm{tot}} = 939, 128, 100$ points per task for the Movielens-100k, the Mini-Wiki and the Jester-1 datasets, respectively. In the above formulation of the problem for the Movielens-100k and the Jester-1 datasets, it is possible to show that, the ITL algorithm is not able to predict any rate for the films/jokes without observed rates. For this reason, in order to evaluate the performance of the Meta-Learning methods ONL-ONL and BAT-ONL, we decided to introduce a more challenging method for this particular formulation of the problem in which, for the films/jokes without any observed rate, we predicted the rate coinciding with the average of the rates of all the observed users, at the end of the entire sequence of tasks. We denoted this method as BAT. In Fig. 1 (Right) and Fig. 2 we report the performance of the methods by using the absolute loss for the Movielens-100k and the Jester-1 datastes and the multi-class hinge loss for the Mini-Wiki dataset. The results we got are consistent with the synthetic experiments above, showing the effectiveness of our method also in real-life scenarios. We note also that the online Meta-Learning methods outperform the BAT method when the number of training tasks increases.

## 8   Conclusion

We presented a fully online Meta-Learning method stemming from primal-dual Online Learning. Our method can be adapted to a wide class of learning algorithms and it covers various types of tasks' relatedness. By means of a new analysis technique we derived a cumulative error bound for our method based on which it is also possible to obtain guarantees in the statistical setting. We illustrated our framework with two important examples, the bias and the feature learning, improving upon state-of-the-art results. To conclude, we believe that the generality of our framework and our method of proof could be a valuable starting point for future theoretical investigations of Meta-Learning.

**Acknowledgments**

This work was supported in part by EPSRC Grant N. EP/P009069/1.

## Footnotes

[1]Throughout the paper we use the double subscript notation "$_{t,i}$", to denote the {outer, inner} task index.

[2]The code is available at https://github.com/dstamos/Adversarial-LTL

[3] https://grouplens.org/datasets/movielens/

[4] http://goldberg.berkeley.edu/jester-data/

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
