[Supplementary Material · camera_ready_OWO_metalearning_January_SUPPLEMENTARY.pdf]

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

 \big\vert\big\vert\big\vert\nabla'_{\theta_t}\big\vert\big\vert\big\vert^2_*\Bigg).$$

**Proof.** The meta-algorithm in Alg. 3 coincides with Alg. 1 applied to the outer-tasks problem in Eq. (3), once one makes the identifications $\alpha'_m \rightsquigarrow \nabla'_{\theta_t}$ for the (approximated) subgradients and realizes that the outer-tasks problem in Eq. (3) is of the form in Eq. (4) with

$$m \rightsquigarrow M, \ j \rightsquigarrow t, \ M \rightsquigarrow T, \ v \rightsquigarrow \theta, \ \mathcal{V} \rightsquigarrow \Theta, \ g_j \rightsquigarrow \mathcal{L}_t, \ A_j \rightsquigarrow I, \ c_m \rightsquigarrow \eta, \ r \rightsquigarrow F.$$

As a consequence, denoting by $\Delta_{\text{Dual}}$ the associated dual optimality gap introduced in Thm. 1, specializing the first point of Thm. 1 to this setting and exploiting the fact $\Delta_{\text{Dual}} \geq 0$, we get

$$0 \leq -\sum_{t=1}^T \mathcal{L}_t(\theta_t) + \min_{\theta \in \Theta}\Big\{\sum_{t=1}^T \mathcal{L}_t(\theta) + \eta F(\theta)\Big\} + \frac{1}{2\eta}\sum_{t=1}^T \big\vert\big\vert\big\vert\nabla'_{\theta_t}\big\vert\big\vert\big\vert^2_* + \

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

# Appendix

The appendix is organized as follows. We start from giving a detailed discussion of previous work in App. A. In App. B we give some necessary preliminaries from convex analysis that are used throughout this work. In App. C we recall the general primal-dual Online Learning framework, which is used in App. C.2 to give the proof of Thm. 1 stated in the main body. In App. D we report the proof of Prop. 3, describing how to compute an approximated meta-subgradient for our meta-objectives. In App. E we report the proofs of the statements given in the main body in Sec. 5 for the statistical setting. In App. F we report the results regarding the application of the within-task Alg. 1 with a meta-parameter fixed in hindsight for any task. These results will be used as benchmark to evaluate our meta-procedure aiming at estimating from the data a good meta-parameter for the inner algorithm. Then, in App. G and App. H we report the results and the computation needed for specializing the general method described in the paper and the corresponding analysis to the settings outlined in Ex. 1 and Ex. 2, respectively. Finally, in App. I, we provide some experimental details we skipped in the main body.

## A  Previous work

We now discuss more in detail some of the papers mentioned above in the main body.

One of the first OWO Meta-Learning framework was presented in [1]. In that case, the proposed setting can cover a quite broad family of inner algorithms and, as observed before, it can be adapted by online-to-batch arguments to the statistical framework. However, the main drawback of that work is the fact that the proposed meta-algorithm is not efficient, since it requires memorizing the entire data sequence.

In [8, 12] the authors focus on the statistical OWB setting and they study the family of regularized empirical risk minimizers with the same regularizer introduced in Ex. 2. In [8], the authors consider a Lipschitz loss function and, in order to estimate from the data the feature map parametrizing the family, they propose to apply Frank-Wolfe or Exponentiated-Weighted as meta-algorithm to the functions given by the minimum of the regularized empirical risks associated to the observed tasks (the same meta-objectives used in this work). In [12], the feature map is estimated by projected Gradient Descent applied to the empirical risk of the inner algorithms, without regularizer. As we will see in the following App. H, the meta-algorithm we will use for this setting will be different.

In the more recent work [16], the authors consider under the Meta-Learning perspective the problem of the so-called *fine tuning*, in which the goal is to estimate a good starting point for a prescribed iterative inner algorithm. Specifically, they consider as inner algorithm one step of gradient descent from the point $\theta$, namely, for an appropriate step size $\gamma > 0$, $w_\theta = \theta - \gamma \nabla \hat{f}(\theta)$, where $\hat{f}$ is some function, for instance an approximation of the (true) risk. Then, in order to estimate the initial point $\theta$, they consider a meta-objective of the form $\mathcal{L}(\theta) = f(\theta - \gamma \nabla \hat{f}(\theta))$, where $f$ is another function with the same intuition of $\hat{f}$. The main result in [16] is to show that, under strong assumptions on the functions $f$ and $\hat{f}$, such meta-objective is (strongly) convex in the meta-parameter $\theta$. Once proven this, they propose to estimate the starting point applying as meta-algorithm Follow-The-Leader on the sequence of these functions and, relying on the well-known analysis for this algorithm, they state a cumulative error bound for it.

Perhaps closer in spirit to our work is [3]. In that work, the authors consider as inner algorithm online Mirror Descent with constant step size and a penalty term given by a Bregman divergence parametrized by a meta-parameter. On the contrary, our inner algorithm corresponds to fixing the step size as $1/(\lambda(i + 1))$ at each iteration and this allows us to derive a *regularized* cumulative error bound. This, as we will see in the following, brings benefits in the statistical setting. Furthermore, the proposed meta-algorithm here is different from the one in [3], in that it works on different objective functions. In their case, as meta-objectives, they consider the sequence of Bregman divergences evaluated at the empirical risk minimizer of the corresponding task, while in our case, we consider the minimum of the entire regularized empirical risk. Such a choice, combined with the primal-dual interpretation of online Mirror Descent and the concept of approximated subgradients, allows us to develop an OWO method without the need of adding further assumptions. On the other hand, in [3], in order to extend their work to the fully online setting, the authors need additional assumptions (specifically a growth condition on the empirical error). We also mention the very

recent (contemporary to our work) sequel [21] where the authors, considering a setting similar to the one described in [3], propose a Meta-Learning approach to estimate also the step-size of the inner algorithm. However, also in this case, the basic version of their method requires to compute a batch within-task empirical risk minimizer and, in order to extend their framework to the fully online setting, they need to introduce additional assumptions on the loss functions.

At last, we briefly discuss our work [11], which is the closest one. As already discussed in Sec. 6 and as we will see in App. G, the method and the analysis proposed there can be recovered from the OWO framework described here for the specific case of Ex. 1 in the statistical setting. In this work, we develop a different analysis which allows us to extend the study to more general family of learning algorithms, also in the non-statistical setting.

# B  Preliminaries on convex analysis

In this appendix we recall some basic concepts of convex analysis. We refer to [7, 19, 4, 31] for a complete and detailed overview.

Let $\mathcal{V}$ be an Euclidean space, i.e a finite dimensional real vector space endowed with an inner product $\langle \cdot, \cdot \rangle$. Moreover, for a generic norm $\| \cdot \|$ over $\mathcal{V}$, we recall that its dual norm $\| \cdot \|_*$ at the point $\alpha \in \mathcal{V}$ is defined as

$$\|\alpha\|_* = \sup_{v \in \mathcal{V}: \|v\| \leq 1} \langle \alpha, v \rangle. \tag{12}$$

As direct consequence of the definition above, we have the following standard fact.

**Lemma 8** (Generalized Holder's inequality)**.** *For any* $\alpha, w \in \mathcal{V}$,

$$\langle \alpha, w \rangle \leq \|\alpha\|_* \|w\|. \tag{13}$$

**Proof.**  We start from observing that $\|w\| = 0$ if, and only if, $w = 0$. If $w = 0$, the statement above is obvious. Thus, we consider the case $w \neq 0$. In such a case, by definition of the dual norm, we can write the following

$$\langle \alpha, w \rangle = \|w\| \left\langle \alpha, \frac{w}{\|w\|} \right\rangle \leq \|w\| \|\alpha\|_*. \tag{14}$$

This coincides with the desired statement. ∎

In the following, we consider extended real-valued functions. We start from giving the following basic definitions, which are frequently used in this work.

**Definition 9** ($\epsilon$-minimizer)**.** *A point* $\hat{v}_\epsilon \in \mathcal{V}$ *is an* $\epsilon$-*minimizer (with* $\epsilon \geq 0$*) of a function* $f : \mathcal{V} \to \mathbb{R} \cup \{+\infty\}$ *if, for any* $v \in \mathcal{V}$,

$$f(\hat{v}_\epsilon) \leq f(v) + \epsilon. \tag{15}$$

The concept of exact minimizer is retrieved from the definition above by setting $\epsilon = 0$. Moreover, an $\epsilon$-maximizer of a function $f$ must be intended as an $\epsilon$-minimizer of the opposite function $-f$.

**Definition 10** (Domain of a function, see e.g. [31, Sec. 2.1])**.** *For a given function* $f : \mathcal{V} \to \mathbb{R} \cup \{+\infty\}$, *define its domain as*

$$\mathrm{Dom} f = \Big\{ v \in \mathcal{V} : f(v) < +\infty \Big\} \subseteq \mathcal{V}. \tag{16}$$

**Definition 11** (Epigraph of a function, see e.g. [31, Sec. 2.1])**.** *For a given function* $f : \mathcal{V} \to \mathbb{R} \cup \{+\infty\}$, *define its epigraph as*

$$\mathrm{Epi} f = \Big\{ (v, t) \in \mathcal{V} \times \mathbb{R} : f(v) \leq t \Big\} \subseteq \mathcal{V} \times \mathbb{R}. \tag{17}$$

The above quantities are now exploited to introduce the following basic definitions.

**Definition 12** (Proper function, see e.g. [31, Sec. 2.1])**.** *A function* $f : \mathcal{V} \to \mathbb{R} \cup \{+\infty\}$ *is proper if* $\mathrm{Dom} f \neq \emptyset$.

**Definition 13** (Closed or lower semi-continuous function, see e.g. [31, Sec. 2.2])**.** *A function* $f : \mathcal{V} \to \mathbb{R} \cup \{+\infty\}$ *is closed or lower semi-continuous if* $\mathrm{Epi} f$ *is a closed set of* $\mathcal{V} \times \mathbb{R}$.

**Definition 14** (Convex function, see e.g. [31, Sec. 2.3]). *A function $f : \mathcal{V} \to \mathbb{R} \cup \{+\infty\}$ is convex if, for any $t \in [0, 1]$ and any $v, v' \in \mathrm{Dom} f$,*

$$f(tv + (1 - t)v') \leq tf(v) + (1 - t)f(v'). \tag{18}$$

The above inequality is known as Jensen's inequality and it can be extended to combinations of more points or expectations of random variables in the following way.

**Lemma 15** (Convex functions and generalized Jensen's inequality, see e.g. [7, Sec. 3.1.8]). *Let $f : \mathcal{V} \to \mathbb{R} \cup \{+\infty\}$ be a convex function and consider a random variable $X$ taking values in $\mathrm{Dom} f$ with probability $1$. Then, provided that the following expectations exist,*

$$f(\mathbb{E} \, X) \leq \mathbb{E} \, f(X). \tag{19}$$

*In particular, in the discrete case, for any sequence of vectors $(v_j)_{j=1}^m \in \mathcal{V}^m$ and weights $(a_j)_{j=1}^m \in \mathbb{R}^m$ such that $a_j \geq 0$ for any $j \in \{1, \ldots, m\}$ and $\sum_{j=1}^m a_j = 1$, we have*

$$f\Big(\sum_{j=1}^m a_j v_j\Big) \leq \sum_{j=1}^m a_j f(v_j). \tag{20}$$

One key property of convex functions is the following.

**Lemma 16** (Convex functions and continuity, see e.g. [31, Prop. 3.5]). *Let $f : \mathcal{V} \to \mathbb{R} \cup \{+\infty\}$ be a convex function. Then, $f$ is continuous on the interior of its domain. In particular, a (real-valued) convex function $f : \mathcal{V} \to \mathbb{R}$ is continuous on the entire space $\mathcal{V}$.*

We now have all the ingredients necessary to introduce the set of functions

$$\Gamma_0(\mathcal{V}) = \Big\{ f : \mathcal{V} \to \mathbb{R} \cup \{+\infty\} : f \text{ is proper, closed and convex}\Big\}. \tag{21}$$

We now recall the following definition, which is frequently used in this work.

**Definition 17** ($\epsilon$-subdifferential of a function, see e.g. [31, Sec. 3.4]). *Let $\epsilon \geq 0$. Then, the $\epsilon$-subdifferential of $f \in \Gamma_0(\mathcal{V})$ at the point $v \in \mathrm{Dom} f$ is the collection of the $\epsilon$-subgradients at that point, namely,*

$$\partial_\epsilon f(v) = \Big\{ \alpha \in \mathcal{V} : f(v') \geq f(v) + \langle \alpha, v' - v \rangle - \epsilon, \text{for any } v' \in \mathrm{Dom} f \Big\}. \tag{22}$$

The standard subdifferential $\partial f$ is retrieved from the above definition by setting $\epsilon = 0$. The following result is a direct consequence of the definition above and it links the concept of the $\epsilon$-subdifferential of a function to the corresponding set of $\epsilon$-minimizers.

**Lemma 18** (Fermat rule, see e.g. [19, Thm. 1.1.5]). *$\hat{v}_\epsilon \in \mathcal{V}$ is an $\epsilon$-minimizer of $f \in \Gamma_0(\mathcal{V})$ if, and only if, $0 \in \partial_\epsilon f(\hat{v}_\epsilon)$.*

Before proceeding, we recall the definition of the Fenchel conjugate of a function.

**Definition 19** (Fenchel conjugate of a function, see e.g. [31, Sec. 3.6]). *Let $f \in \Gamma_0(\mathcal{V})$. Then, its Fenchel conjugate $f^* : \mathcal{V} \to \mathbb{R} \cup \{+\infty\}$ is defined at $\alpha \in \mathcal{V}$ as*

$$f^*(\alpha) = \sup_{v \in \mathcal{V}} \langle v, \alpha \rangle - f(v). \tag{23}$$

In our proofs, we exploit the following standard properties of the conjugate function.

**Lemma 20** (Fenchel conjugate and rescaling, see e.g. [7, Sec. 3.3.2]). *Let $f \in \Gamma_0(\mathcal{V})$ and $c > 0$. Then, for any $\alpha \in \mathcal{V}$, $(cf)^*(\alpha) = cf^*(\alpha/c)$.*

**Lemma 21** (Separable functions and Fenchel conjugate, see e.g. [7, Sec. 3.3.2]). *Let $\mathcal{V}_1, \ldots, \mathcal{V}_m$ be Euclidean spaces. For any $v = (v_1, \ldots, v_m) \in \mathcal{V}_1 \times \cdots \times \mathcal{V}_m$, let*

$$f(v) = \sum_{j=1}^m f_j(v_j), \tag{24}$$

*with $f_j \in \Gamma_0(\mathcal{V}_j)$. Then, for any $\alpha = (\alpha_1, \ldots, \alpha_m) \in \mathcal{V}_1 \times \cdots \times \mathcal{V}_m$, we have*

$$f^*(\alpha) = \sum_{j=1}^m f_j^*(\alpha_j). \tag{25}$$

**Lemma 22** (Fenchel conjugate and monotonicity, see e.g. [31, Prop. 3.50]). *Let $f_1, f_2 \in \Gamma_0(\mathcal{V})$ such that $f_1 \leq f_2$. Then, $f_1^* \geq f_2^*$.*

**Lemma 23** (Young-Fenchel inequality, see e.g. [19, Prop. 1.2.1]). *Let $f \in \Gamma_0(\mathcal{V})$ and consider $v \in \mathrm{Dom}f$. Then, $\alpha \in \partial_\epsilon f(v)$ if, and only if,*

$$f^*(\alpha) - \langle \alpha, v \rangle \leq -f(v) + \epsilon. \tag{26}$$

We now introduce a further definition which is used throughout this work.

**Definition 24** (Lipschitz function, see e.g. [34, Def. 12.6]). *A function $f : \mathcal{V} \to \mathbb{R} \cup \{+\infty\}$ is L-Lipschitz (with $L > 0$) w.r.t. a norm $\|\cdot\|$ over $\mathcal{V}$ if, for any $v, v' \in \mathrm{Dom}f$,*

$$\left| f(v) - f(v') \right| \leq L \, \|v - v'\|. \tag{27}$$

The above definition implies the following bound on the dual norm of the subgradients.

**Lemma 25** (Lipschitz functions and bounded subgradients, see e.g. [34, Lemma 14.7]). *Let $\|\cdot\|$ be a norm over $\mathcal{V}$ and let $\|\cdot\|_*$ be its dual. A function $f : \mathcal{V} \to \mathbb{R} \cup \{+\infty\}$ with open domain is L-Lipschitz w.r.t. $\|\cdot\|$ if, and only if, for any $v \in \mathrm{Dom}f$ and for any $\alpha \in \partial f(v)$, $\|\alpha\|_* \leq L$.*

Another definition we need is the following.

**Definition 26** (Lipschitz smooth function). *Let $\|\cdot\|$ be a norm over $\mathcal{V}$ and let $\|\cdot\|_*$ be its dual. A (real-valued) function $f : \mathcal{V} \to \mathbb{R}$ is $\beta$-Lipschitz smooth (with $\beta > 0$) w.r.t. $\|\cdot\|$ if it is differentiable and, for any $v, v' \in \mathcal{V}$, it holds that*

$$\left\| \nabla f(v) - \nabla f(v') \right\|_* \leq \beta \, \|v - v'\|. \tag{28}$$

The following result describes a well-known property of Lipschitz smooth functions.

**Lemma 27** (Lipschitz smooth functions and descent lemma, see e.g. [31, Lemma 1.30]). *Let $f : \mathcal{V} \to \mathbb{R}$ be a $\beta$-Lipschitz smooth function w.r.t. a norm $\|\cdot\|$ over $\mathcal{V}$. Then, for any $v, v' \in \mathcal{V}$,*

$$f(v') \leq f(v) + \langle \nabla f(v), v' - v \rangle + \frac{\beta}{2} \, \|v' - v\|^2. \tag{29}$$

Before proceeding, we strengthen the notion of convexity as follows.

**Definition 28** (Strongly convex function, see e.g. [31, Sec 2.3]). *A function $f : \mathcal{V} \to \mathbb{R} \cup \{+\infty\}$ is $\sigma$-strongly convex (with $\sigma > 0$) w.r.t. a norm $\|\cdot\|$ over $\mathcal{V}$ if, for any $t \in [0, 1]$ and any $v, v' \in \mathrm{Dom}f$,*

$$f(tv + (1-t)v') \leq tf(v) + (1-t)f(v') - \frac{\sigma}{2} \, t(1-t) \, \|v - v'\|^2. \tag{30}$$

The following result describes a key property of strongly convex functions.

**Lemma 29** (Strongly convex functions and minimizers, see e.g. [31, Prop. 3.23]). *Let $f : \mathcal{V} \to \mathbb{R} \cup \{+\infty\}$ be a proper, closed and $\sigma$-strongly convex function w.r.t. a norm $\|\cdot\|$ over $\mathcal{V}$. Then, $f$ admits a minimizer over $\mathcal{V}$ and such a minimizer is unique.*

We now give two key results for our proofs. The first one describes the duality between strong convexity and Lipschitz smoothness, the second one allows us to study the scaling effect on the Fenchel conjugate function.

**Lemma 30** (Duality between strong convexity and Lipschitz smoothness, see e.g. [20, Thm. 6], [36, Lemma 3]). *Let $\|\cdot\|$ be a norm over $\mathcal{V}$ and let $\|\cdot\|_*$ be its dual. Let $f : \mathcal{V} \to \mathbb{R} \cup \{+\infty\}$ be a proper, closed and $\sigma$-strongly convex function w.r.t. $\|\cdot\|$. Then, $f^*$ is $(1/\sigma)$-Lipschitz smooth w.r.t. $\|\cdot\|_*$. Moreover, for any $\alpha \in \mathcal{V}$,*

$$\nabla f^*(\alpha) = \operatorname*{argmax}_{v \in \mathcal{V}} \langle \alpha, v \rangle - f(v) \in \mathrm{Dom}f. \tag{31}$$

**Lemma 31** (Fenchel conjugate and scaling effect, see e.g. [36, Lemma 4]). *Let $\|\cdot\|$ be a norm over $\mathcal{V}$ and let $\|\cdot\|_*$ be its dual. Let $f \in \Gamma_0(\mathcal{V})$ be a strongly convex function w.r.t. $\|\cdot\|$ and consider $c_1, c_2 > 0$. Then, for any $\alpha \in \mathcal{V}$, introducing the vector $v_{c_2} = \nabla f^*(\alpha/c_2)$, we have*

$$(c_2 f)^*(\alpha) - (c_1 f)^*(\alpha) = c_2 f^*(\alpha/c_2) - c_1 f^*(\alpha/c_1) \leq (c_1 - c_2)f(v_{c_2}). \tag{32}$$

In the following section we briefly recall the main results we need from Fenchel Duality.

### B.1 Fenchel Duality

For the content in this section, the reader can refer to [31, Sec. 3.6.2]. Given two Euclidean spaces $\mathcal{V}$ and $\mathcal{U}$, a linear operator $\mathcal{A} : \mathcal{V} \to \mathcal{U}$ and two functions $J \in \Gamma_0(\mathcal{V})$ and $G \in \Gamma_0(\mathcal{U})$, consider the primal problem

$$\hat{P} = \inf_{v \in \mathcal{V}} P(v) \qquad P(v) = G(\mathcal{A}v) + J(v). \tag{33}$$

The associated dual problem reads as follows

$$\hat{D} = \inf_{\alpha \in \mathcal{U}} D(\alpha) \qquad D(\alpha) = G^*(\alpha) + J^*(-\mathcal{A}^*\alpha), \tag{34}$$

where $\mathcal{A}^* : \mathcal{U} \to \mathcal{V}$ is the adjoint operator of $\mathcal{A}$ and $G^*$ and $J^*$ are the Fenchel conjugates of $G$ and $J$, respectively. We recall also that the *duality gap* associated to two generic points $v \in \mathcal{V}$ and $\alpha \in \mathcal{U}$ is defined as

$$P(v) + D(\alpha). \tag{35}$$

It is well know that, for any $v \in \mathcal{V}$ and $\alpha \in \mathcal{U}$, the above quantity is always non-negative, i.e.

$$-D(\alpha) \le P(v). \tag{36}$$

As a consequence, we have

$$\sup_{\alpha \in \mathcal{U}} \{-D(\alpha)\} = - \inf_{\alpha \in \mathcal{U}} D(\alpha) = -\hat{D} \le \inf_{v \in \mathcal{V}} P(v) = \hat{P}. \tag{37}$$

The following proposition studies when the above inequality is in fact an equality.

**Proposition 32** (Strong duality, see e.g. [31, Thm. 3.51])**.** *Consider the primal and the dual problems in Eq. (33) and Eq. (34). Assume that there exist a point $v \in \mathrm{Dom}J$ such that $G$ is continuous at $\mathcal{A}v$ and assume that the primal problem in Eq. (33) admits a solution*

$$\hat{v} \in \operatorname*{argmin}_{v \in \mathcal{V}} P(v). \tag{38}$$

*Then, the dual problem in Eq. (34) admits a solution*

$$\hat{\alpha} \in \operatorname*{argmin}_{\alpha \in \mathcal{U}} D(\alpha). \tag{39}$$

*Moreover, the following statements hold.*

1. *Strong duality holds, namely,*

$$- \min_{\alpha \in \mathcal{U}} D(\alpha) = -D(\hat{\alpha}) = -\hat{D} = \min_{v \in \mathcal{V}} P(v) = \hat{P}(\hat{v}) = \hat{P}. \tag{40}$$

2. *The optimality conditions, also known as the Karush–Kuhn–Tucker (KKT) conditions, read as follows*

$$\hat{v} \in \partial J^*(-\mathcal{A}^*\hat{\alpha}) \qquad \hat{\alpha} \in \partial G(\mathcal{A}\hat{v}). \tag{41}$$

## C  Primal-dual Online Learning

In this appendix we recall the primal-dual Online Learning framework. Specifically, in App. C.1 we report some background material which is then used in the following App. C.2 for the proof of Thm. 1 in Sec. 3 in the main body. The material in this appendix is based on [36, 35, 37, 38].

Many online algorithms on a (primal) problem can be derived from the following primal-dual framework. At each iteration $m \in \{1, \dots, M\}$, $a)$ we define a pair of *instantaneous* primal-dual problems, $b)$ we update the dual variable according to an appropriate greedy coordinate descent procedure on the dual, $c)$ we update the new primal variable by evaluating the KKT conditions at the current dual variable. We now describe the above steps in detail. Throughout this appendix, we let $\mathcal{V}$ be an Euclidean space endowed with a scalar product $\langle \cdot, \cdot \rangle$ and a generic norm $\| \cdot \|$ with dual $\| \cdot \|_*$.

**a) The primal and the dual problems.** Regarding the first step, for any iteration $m \in \{1, \dots, M\}$, consider the primal problem of the following form as in Eq. (4)

$$\hat{P}_{m+1} = \inf_{v \in \mathcal{V}} P_{m+1}(v) \qquad P_{m+1}(v) = \sum_{j=1}^{m} g_j(A_j v) + c_m r(v), \tag{42}$$

where $c_m > 0$, $r \in \Gamma_0(\mathcal{V})$ is a $\sigma_r$-strongly convex function (with $\sigma_r > 0$) w.r.t. a norm $\|\cdot\|$ such that $\inf_{v \in \mathcal{V}} r(v) = 0$, for any $j \in \{1, \ldots, M\}$, letting $\mathcal{V}_j$ an Euclidean space, $g_j \in \Gamma_0(\mathcal{V}_j)$ and $A_j : \mathcal{V} \to \mathcal{V}_j$ is a linear operator with adjoint $A_j^*$. Even though it is not necessary, to simplify the presentation, we set $P_1 \equiv 0$. Introducing the following linear operator

$$\mathcal{A}_m : \mathcal{V} \to \mathcal{V}_1 \times \cdots \times \mathcal{V}_m \qquad v \in \mathcal{V} \mapsto (A_1 v, \ldots, A_m v) \in \mathcal{V}_1 \times \cdots \times \mathcal{V}_m \qquad (43)$$

and the function $G_m \in \Gamma_0(\mathcal{V}_1 \times \ldots \mathcal{V}_m)$ defined, for any $\alpha = (\alpha_1, \ldots, \alpha_m) \in \mathcal{V}_1 \times \cdots \times \mathcal{V}_m$, as

$$G_m(\alpha) = \sum_{j=1}^{m} g_j(\alpha_j), \qquad (44)$$

we can rewrite the problem in Eq. (42) as

$$\hat{P}_{m+1} = \inf_{v \in \mathcal{V}} P_{m+1}(v) \qquad P_{m+1}(v) = G_m(\mathcal{A}_m v) + c_m r(v). \qquad (45)$$

Hence, according to what observed in App. B.1, exploiting the separability of $G_m$ (see Lemma 21 in App. B), using the scaling properties of the conjugate (see Lemma 20 in App. B) and observing that the adjoint operator of $\mathcal{A}_m$ is give by

$$\mathcal{A}_m^* : \mathcal{V}_1 \times \cdots \times \mathcal{V}_m \to \mathcal{V} \qquad \alpha = (\alpha_1, \ldots, \alpha_m) \in \mathcal{V}_1 \times \cdots \times \mathcal{V}_m \mapsto \sum_{j=1}^{m} A_j^* \alpha_j \in \mathcal{V}, \qquad (46)$$

the dual of the problem in Eq. (42) is given by

$$\hat{D}_{m+1} = \inf_{\alpha \in \mathcal{V}_1 \times \cdots \times \mathcal{V}_m} D_{m+1}(\alpha) \qquad D_{m+1}(\alpha) = \underbrace{\sum_{j=1}^{m} g_j^*(\alpha_j)}_{G_m^*(\alpha)} + \underbrace{c_m r^* \left( -\frac{1}{c_m} \sum_{j=1}^{m} A_j^* \alpha_j \right)}_{(c_m r)^*(-\mathcal{A}_m^* \alpha)}, \qquad (47)$$

where $g_j^*$ and $r^*$ represent the conjugate function of $g_j$ and $r$, respectively. To simplify, we set also in this case $D_1 \equiv 0$. We observe that, when the above problems satisfy the assumptions in Prop. 32 in App. B, since the strong convexity of $r$ is equivalent to the Lipschitz-smoothness of $r^*$ (see Lemma 30 in App. B), denoting by $\hat{v}_{m+1}$ and $\hat{\alpha}_{m+1}$ a solution of the primal and the dual problem above, the corresponding KKT conditions read as follows

$$\hat{v}_{m+1} = \nabla r^* \left( -\frac{1}{c_m} \mathcal{A}_m^* \hat{\alpha}_{m+1} \right) \qquad \hat{\alpha}_{m+1} \in \partial G_m(\mathcal{A}_m \hat{v}_{m+1}), \qquad (48)$$

where, more explicitly, we recall that

$$\mathcal{A}_m^* \hat{\alpha}_{m+1} = \sum_{j=1}^{m} A_j^* \hat{\alpha}_{m+1,j}. \qquad (49)$$

We observe that, under the assumptions above, the primal objective $P_{m+1}$ results to be proper, closed and strongly convex w.r.t. the norm $\|\cdot\|$. As a consequence, by Lemma 29 in App. B, we can in fact ensure the existence and the uniqueness of the primal solution $\hat{v}_{m+1}$.

We now are ready to describe the dual and the primal updating steps.

**b) c) The updating rules.** The algorithm updates the dual variable $\alpha_{m+1}$ in a such way that, for a given parameter $\epsilon_m \geq 0$, there exist $\alpha_m' \in \partial_{\epsilon_m} g_m(A_m v_m)$ such that

$$D_{m+1}(\alpha_{m+1}) \leq D_{m+1}(\underbrace{\alpha_{m,1}, \ldots, \alpha_{m,m-1}}, \alpha_m') = D_{m+1}(\underbrace{\alpha_m}, \alpha_m'). \qquad (50)$$

The primal variable is then updated by the KKT conditions from the dual one. More precisely, following [38], in this last step we use a slightly different version of the KKT conditions in which we divide by $c_{m+1}$ instead of $c_m$ as in Eq. (48). For more details we refer to Alg. 4, which is a more general version of Alg. 1 given in the main body in Sec. 3. We also observe that, by definition, thanks to Lemma 30 in App. B, the primal variables $(v_m)_{m=1}^{M}$ generated by the algorithm are guaranteed to belong to Dom $r$.

Note that the requirement above about the dual update in Eq. (50) is satisfied (with the equality) by the update described in the main body $\alpha_{m+1} = (\alpha_m, \alpha_m')$. The resulting primal algorithm

---

**Algorithm 4** Primal-dual online algorithm (more general version of Alg. 1)

---

**Input** $(g_m)_{m=1}^M$, $(A_m)_{m=1}^M$, $(c_m)_{m=1}^M$, $(\epsilon_m)_{m=1}^M$, $r$ as described in the text

**Initialization** $\alpha_1 = ()$, $v_1 = \nabla r^*(0) \in \mathrm{Dom}\, r$

**For** $m = 1$ to $M$

    Receive $g_m, A_m, c_{m+1}, \epsilon_m$

    Suffer $g_m(A_m v_m)$ and compute $\alpha'_m \in \partial_{\epsilon_m} g_m(A_m v_m)$

    Update $\alpha_{m+1}$ according to Eq. (50) by using $\alpha'_m$

    Define $v_{m+1} = \nabla r^*\big(-1/c_{m+1}\mathcal{A}_m^* \alpha_{m+1}\big) = \nabla r^*\big(-1/c_{m+1}\sum_{j=1}^m A_j^* \alpha_{m+1,j}\big) \in \mathrm{Dom}\, r$

**Return** $(\alpha_m)_{m=1}^{M+1}$, $(v_m)_{m=1}^{M+1}$

---

coincides in this case with a lazy variant of online Mirror Descent. However, we stress that Eq. (50) is satisfied also by other more aggressive dual steps, including for example the one generating the primal Follow-The-Regularized-Leader updating scheme. We refer to [36, 35, 37, 38] for more details about this.

We finally conclude by observing that the framework above is a slightly different version of the standard primal-dual Online Learning setting described in the papers mentioned above. The differences in our presentation are the introduction of the linear operators $(A_m)_{m=1}^M$ inside the functions $(g_m)_{m=1}^M$ and the possibility to deal with an approximation of the subdifferential $\partial g_m(A_m v_m)$. These two modifications will allow us to adapt the theory above to the Meta-Learning setting described in the main body.

### C.1 Main inequality on the dual gap

In the next proposition we study the behavior of the gap between two consecutive iterations on the dual objective for Alg. 4 (or Alg. 1). This statement will be the main tool used in App. C.2 in order to prove Thm. 1 in Sec. 3.

**Proposition 33** (Dual Gap, see [33, Lemma 1]). *Let $(\alpha_m)_{m=1}^{M+1}$ and $(v_m)_{m=1}^{M+1}$ be the iterates returned by Alg. 4 (or Alg. 1). Then,*

$$\Delta_1 = D_2(\alpha_2) - D_1(\alpha_1) \leq -g_1(A_1 v_1) + \frac{1}{2\sigma_r c_1} \left\| A_1^* \alpha'_1 \right\|_*^2 + \epsilon_1. \tag{51}$$

*Furthermore, for any $m \in \{2, \dots, M\}$, we have*

$$
\begin{aligned}
\Delta_m &= D_{m+1}(\alpha_{m+1}) - D_m(\alpha_m) \\
&\leq - g_m(A_m v_m) + \frac{1}{2\sigma_r c_m} \left\| A_m^* \alpha'_m \right\|_*^2 + \epsilon_m \\
&\quad + c_m r^*\Big(-\frac{1}{c_m}\mathcal{A}_{m-1}^* \alpha_m\Big) - c_{m-1} r^*\Big(-\frac{1}{c_{m-1}}\mathcal{A}_{m-1}^* \alpha_m\Big).
\end{aligned}
\tag{52}
$$

**Proof.** We first prove Eq. (51). Thanks to the updating rule in Eq. (50), the closed form of the dual objective in Eq. (47) and the definition $D_1 \equiv 0$, we can write

$$\Delta_1 = D_2(\alpha_2) - D_1(\alpha_1) = D_2(\alpha_2) \leq D_2(\alpha'_1) = g_1^*(\alpha'_1) + c_1 r^*\Big(-\frac{1}{c_1}A_1^* \alpha'_1\Big), \tag{53}$$

where $\alpha'_1 \in \partial_{\epsilon_1} g_1(A_1 v_1)$ is the approximated subgradient used by Alg. 4 (or Alg. 1). But, thanks to Lemma 30 in App. B, the $\sigma_r$-strong convexity of $r$ w.r.t. $\|\cdot\|$ is equivalent to the $(1/\sigma_r)$-Lipschitz smoothness of $r^*$ w.r.t. $\|\cdot\|_*$, hence, applying Lemma 27 in App. B, exploiting the definition of $v_1$ in Alg. 4 (or Alg. 1) and the assumption $r^*(0) = \inf_{v \in \mathcal{V}} r(v) = 0$, we have

$$
\begin{aligned}
r^*\Big(-\frac{1}{c_1}A_1^* \alpha'_1\Big) &\leq r^*(0) - \frac{1}{c_1}\big\langle \nabla r^*(0), A_1^* \alpha'_1\big\rangle + \frac{1}{2\sigma_r c_1^2}\left\| A_1^* \alpha'_1 \right\|_*^2 \\
&= -\frac{1}{c_1}\big\langle v_1, A_1^* \alpha_1^*\big\rangle + \frac{1}{2\sigma_r c_1^2}\left\| A_1^* \alpha'_1 \right\|_*^2.
\end{aligned}
\tag{54}
$$

Substituting in Eq. (53), we get the statement

$$\Delta_1 \le g_1^*(\alpha_1') + c_1 r^*\left(-\frac{1}{c_1} A_1^* \alpha_1'\right) \le g_1^*(\alpha_1') - \langle v_1, A_1^* \alpha_1^* \rangle + \frac{1}{2\sigma_r c_1} \left\| A_1^* \alpha_1' \right\|_*^2$$
$$\le -g_1(A_1 v_1) + \epsilon_1 + \frac{1}{2\sigma_r c_1} \left\| A_1^* \alpha_1' \right\|_*^2, \tag{55}$$

where, in the last inequality, we have exploited the fact that $\alpha_1' \in \partial_{\epsilon_1} g_1(A_1 v_1)$ and Lemma 23 in App. B. We now prove the statement for $m \in \{2, \ldots M\}$. By Eq. (50), the closed form of the dual objective in Eq. (47) and the rewriting

$$\mathcal{A}_m^* \alpha_{m+1} = \mathcal{A}_{m-1}^* \alpha_m + A_m^* \alpha_m', \tag{56}$$

with $\alpha_m' \in \partial_{\epsilon_m} g_m(A_m v_m)$ the approximated subgradient used by Alg. 4 (or Alg. 1), we have

$$\Delta_m = D_{m+1}(\alpha_{m+1}) - D_m(\alpha_m) \le D_{m+1}(\alpha_m, \ \alpha_m') - D_m(\alpha_m)$$
$$= g_m^*(\alpha_m') + c_m r^*\left(-\frac{1}{c_m}\mathcal{A}_{m-1}^*\alpha_m - \frac{1}{c_m}A_m^*\alpha_m'\right) - c_{m-1}r^*\left(-\frac{1}{c_{m-1}}\mathcal{A}_{m-1}^*\alpha_m\right). \tag{57}$$

Again, thanks to Lemma 30 in App. B, the $\sigma_r$-strong convexity of $r$ w.r.t. $\|\cdot\|$ is equivalent to the $(1/\sigma_r)$-Lipschitz smoothness of $r^*$ w.r.t. $\|\cdot\|_*$, hence, applying Lemma 27 in App. B and exploiting the definition of $v_m$ in Alg. 4 (or Alg. 1), we have

$$r^*\left(-\frac{1}{c_m}\mathcal{A}_{m-1}^*\alpha_m - \frac{1}{c_m}A_m^*\alpha_m'\right)$$
$$\le r^*\left(-\frac{1}{c_m}\mathcal{A}_{m-1}^*\alpha_m\right) - \frac{1}{c_m}\left\langle \nabla r^*\left(-\frac{1}{c_m}\mathcal{A}_{m-1}^*\alpha_m\right), A_m^*\alpha_m'\right\rangle + \frac{1}{2\sigma_r c_m^2}\left\|A_m^*\alpha_m'\right\|_*^2$$
$$= r^*\left(-\frac{1}{c_m}\mathcal{A}_{m-1}^*\alpha_m\right) - \frac{1}{c_m}\langle v_m, A_m^*\alpha_m'\rangle + \frac{1}{2\sigma_r c_m^2}\left\|A_m^*\alpha_m'\right\|_*^2.$$

Substituting into Eq. (57), we can write the following

$$\Delta_m \le g_m^*(\alpha_m') + c_m r^*\left(-\frac{1}{c_m}\mathcal{A}_{m-1}^*\alpha_m - \frac{1}{c_m}A_m^*\alpha_m'\right) - c_{m-1}r^*\left(-\frac{1}{c_{m-1}}\mathcal{A}_{m-1}^*\alpha_m\right)$$
$$\le g_m^*(\alpha_m') - \langle v_m, A_m^*\alpha_m'\rangle + \frac{1}{2\sigma_r c_m}\left\|A_m^*\alpha_m'\right\|_*^2$$
$$\qquad\qquad\qquad + c_m r^*\left(-\frac{1}{c_m}\mathcal{A}_{m-1}^*\alpha_m\right) - c_{m-1}r^*\left(-\frac{1}{c_{m-1}}\mathcal{A}_{m-1}^*\alpha_m\right)$$
$$\le -g_m(A_m v_m) + \epsilon_m + \frac{1}{2\sigma_r c_m}\left\|A_m^*\alpha_m'\right\|_*^2$$
$$\qquad\qquad\qquad + c_m r^*\left(-\frac{1}{c_m}\mathcal{A}_{m-1}^*\alpha_m\right) - c_{m-1}r^*\left(-\frac{1}{c_{m-1}}\mathcal{A}_{m-1}^*\alpha_m\right),$$

where, in the last inequality, we have exploited the fact that $\alpha_m' \in \partial_{\epsilon_m} g_m(A_m v_m)$ and Lemma 23 in App. B. The last inequality above coincides with the desired statement. ∎

## C.2 Proof of Thm. 1

In this section, starting from the result described above in Prop. 33, we present the proof of Thm. 1 reported in the main body. More precisely, we provide the proof of a more general statement with a generic strong convexity parameter $\sigma_r > 0$ for the function $r$. For convenience of the reader, we restate Thm. 1 here. The first point of the statement below is an adaptation of [33, Lemma 1], while, for the second point, we refer to [36, Lemma 5].

**Theorem 1** (Dual optimality gap for Alg. 1). *Let $(v_m)_{m=1}^M$ be the primal iterates returned by Alg. 1 when applied to the generic problem in Eq. (4) and let $\Delta_{\text{Dual}} = D_{M+1}(\alpha_{M+1}) - \hat{D}_{M+1}$ be the corresponding (non-negative) dual optimality gap at the last dual iterate $\alpha_{M+1}$ of the algorithm.*

*1. If, for any $m \in \{1, \ldots, M\}$, $c_{m+1} \ge c_m$, then,*

$$\Delta_{\text{Dual}} \le -\sum_{m=1}^M g_m(A_m v_m) + \hat{P}_{M+1} + \frac{1}{2}\sum_{m=1}^M \frac{1}{c_m}\left\|A_m^*\alpha_m'\right\|_*^2 + \sum_{m=1}^M \epsilon_m.$$

2. *If, for any $m \in \{1, \ldots, M\}$, $c_m = \sum_{j=1}^{m} \lambda_j$ for some $\lambda_j > 0$, then,*

$$\Delta_{\text{Dual}} \leq - \sum_{m=1}^{M} \left\{ g_m(A_m v_m) + \lambda_m r(v_m) \right\} + \hat{P}_{M+1} + \frac{1}{2} \sum_{m=1}^{M} \frac{1}{c_m} \left\| A_m^* \alpha_m' \right\|_*^2 + \sum_{m=1}^{M} \epsilon_m.$$

## C.3 Proof of Thm. 1 point 1.

In this subsection we prove the first point of Thm. 1, namely, the bound linking the optimality reached by the last dual iteration of Alg. 4 (or Alg. 1) to the cumulative error of the corresponding primal iterates.

**Proof of Thm. 1 point 1.** We first show that, for any $m \in \{1, \ldots, M\}$,

$$\Delta_m \leq -g_m(A_m v_m) + \frac{1}{2\sigma_r c_m} \left\| A_m^* \alpha_m' \right\|_*^2 + \epsilon_m. \tag{58}$$

As described in Prop. 33, the statement above in Eq. (58) holds for the case $m = 1$. For $m \in \{2, \ldots, M\}$, we observe the following. Thanks to the choice of the increasing parameters $c_{m+1} \geq c_m$ and the non-negativity of $r$, according to Lemma 22 in App. B, we have

$$\begin{aligned}
&c_m r^* \left( -\frac{1}{c_m} \mathcal{A}_{m-1}^* \alpha_m \right) - c_{m-1} r^* \left( -\frac{1}{c_{m-1}} \mathcal{A}_{m-1}^* \alpha_m \right) \\
&= (c_m r)^* (-\mathcal{A}_{m-1}^* \alpha_m) - (c_{m-1} r)^* (-\mathcal{A}_{m-1}^* \alpha_m) \leq 0.
\end{aligned} \tag{59}$$

Substituting this last inequality in Prop. 33, we get the statement in Eq. (58) for $m \in \{2, \ldots, M\}$. Now, we observe that, thanks to the definition $D_1 \equiv 0$, we can write

$$D_{M+1}(\alpha_{M+1}) = \sum_{m=1}^{M} \Delta_m + D_1(\alpha_1) = \sum_{m=1}^{M} \Delta_m. \tag{60}$$

Thus, summing Eq. (58) over $m \in \{1, \ldots, M\}$, we obtain that

$$D_{M+1}(\alpha_{M+1}) \leq - \sum_{m=1}^{M} g_m(A_m v_m) + \frac{1}{2\sigma_r} \sum_{m=1}^{M} \frac{1}{c_m} \left\| A_m^* \alpha_m' \right\|_*^2 + \sum_{m=1}^{M} \epsilon_m. \tag{61}$$

The desired statement now follows by summing to this last inequality the following relation

$$-\hat{D}_{M+1} \leq \hat{P}_{M+1}, \tag{62}$$

coinciding with the non-negativity of the duality gap in Eq. (37). ∎

## C.4 Proof of Thm. 1 point 2.

In this subsection we prove the second point of Thm. 1, namely, the bound linking the optimality reached by the last dual iteration of Alg. 4 (or Alg. 1) to the *regularized* cumulative error of the corresponding primal iterates.

**Proof of Thm. 1 point 2.** We first show that, for any $m \in \{1, \ldots, M\}$,

$$\Delta_m \leq -\big(g_m(A_m v_m) + \lambda_m r(v_m)\big) + \frac{1}{2\sigma_r c_m} \left\| A_m^* \alpha_m' \right\|_*^2 + \epsilon_m. \tag{63}$$

Thanks to the definition $v_1 = \nabla r^*(0)$ in Alg. 4 (or Alg. 1), Lemma 30 in App. B and the assumption $\inf_{v \in \mathcal{V}} r(v) = 0$, we can write $r(v_1) = r(\nabla r^*(0)) = \inf_{v \in \mathcal{V}} r(v) = 0$. As a consequence, by Prop. 33, the above statement in Eq. (63) holds for the case $m = 1$. For any $m \in \{2, \ldots, M\}$, introducing the notation $\lambda_{1:m} = \sum_{j=1}^{m} \lambda_j$, we can write

$$\begin{aligned}
c_m r^* &\left( -\frac{1}{c_m} \mathcal{A}_{m-1}^* \alpha_m \right) - c_{m-1} r^* \left( -\frac{1}{c_{m-1}} \mathcal{A}_{m-1}^* \alpha_m \right) \\
&\leq (c_{m-1} - c_m) \, r \left( \nabla r^* \left( -\frac{1}{c_m} \mathcal{A}_{m-1}^* \alpha_m \right) \right) \\
&= (\lambda_{1:m-1} - \lambda_{1:m}) \, r \left( \nabla r^* \left( -\frac{1}{\lambda_{1:m}} \mathcal{A}_{m-1}^* \alpha_m \right) \right) \\
&= (\lambda_{1:m-1} - \lambda_{1:m}) \, r(v_m) = -\lambda_m r(v_m),
\end{aligned} \tag{64}$$

where, in the inequality we have applied Lemma 31 in App. B to $c_1 \rightsquigarrow c_{m-1}$, $c_2 \rightsquigarrow c_m$, $f \rightsquigarrow r$, $\alpha \rightsquigarrow -\mathcal{A}_{m-1}^* \alpha_m$, in the first equality we have introduced the definition of the parameter $c_m = \lambda_{1:m}$ and in the second equality we have exploited the definition of $v_m$ in Alg. 4 (or Alg. 1). Substituting this last inequality in Prop. 33, we get the statement in Eq. (63) for $m \in \{2, \ldots, M\}$. Now, we observe again that, thanks to the definition $D_1 \equiv 0$, we have

$$D_{M+1}(\alpha_{M+1}) = \sum_{m=1}^{M} \Delta_m + D_1(\alpha_1) = \sum_{m=1}^{M} \Delta_m. \tag{65}$$

Thus, summing Eq. (63) over $m \in \{1, \ldots, M\}$, we obtain

$$D_{M+1}(\alpha_{M+1}) \leq -\Big(\sum_{m=1}^{M} g_m(A_m v_m) + \lambda_m r(v_m)\Big) + \frac{1}{2\sigma_r} \sum_{m=1}^{M} \frac{1}{\lambda_{1:m}} \left\|A_m^* \alpha_m'\right\|_*^2 + \sum_{m=1}^{M} \epsilon_m.$$

Also in this case, the desired statement follows by summing to this last inequality the following relation

$$-\hat{D}_{M+1} \leq \hat{P}_{M+1}, \tag{66}$$

coinciding the non-negativity of the duality gap in Eq. (37). ∎

# D  Computation of the approximated meta-subgardients, proof of Prop. 3

In this section we report the proof of Prop. 3, describing how to compute an approximate subgradient for our meta-objectives. In order to do this, we need the following technical lemma.

**Lemma 34** (Strong duality for the within-task problem). *Let Asm. 1 hold. For any dataset $Z$ and any meta-parameter $\theta \in \Theta$, consider the non-normalized primal within-task problem in Eq. (2). Then, the corresponding dual problem with objective in Eq. (8) admits a solution*

$$\hat{s}_\theta \in \operatorname*{argmin}_{s \in \mathbb{R}^n} \ D_{n+1}(s, \theta). \tag{67}$$

*Moreover, the following statements hold.*

1. *Strong duality holds, namely, we have*

$$n\,\mathcal{L}_Z(\theta) = -\min_{s \in \mathbb{R}^n} \ D_{n+1}(s, \theta). \tag{68}$$

2. *The KKT conditions read as follows*

$$\hat{w}_\theta = \nabla f(\cdot, \theta)^* \Big(-\frac{1}{\lambda n} \sum_{i=1}^{n} x_i \hat{s}_{\theta,i}\Big) \qquad \hat{s}_\theta \in \partial\Big(\sum_{i=1}^{n} \ell_i\Big)\big(\langle x_1, \hat{w}_\theta\rangle, \ldots, \langle x_n, \hat{w}_\theta\rangle\big), \tag{69}$$

   *where, we recall that $\hat{w}_\theta$ is the minimizer of the primal problem in Eq. (2).*

**Proof.** We rely on the standard result reported in Prop. 32 in App. B.1 according to which, the desired statements hold for the couples of within-task primal-dual problems above if, for any $\theta \in \Theta$, 1) the primal problem admits a solution and 2) there exist a point in $\mathrm{Dom} f(\cdot, \theta)$ where the function $\sum_{i=1}^{n} \ell_i(\langle x_i, \cdot\rangle)$ is continuous. Regarding the point 1), as already observed in the main body, the existence of the primal solution $\hat{w}_\theta$ is ensured by Asm. 1. Regarding the point 2), we observe that, thanks to Asm. 1, the function $\sum_{i=1}^{n} \ell_i(\langle x_i, \cdot\rangle)$ is real-valued. As a consequence, since a convex real-valued function is continuous over the entire space (see Lemma 16 in App. B), also the continuity requirement above is satisfied. Hence, the desired statement directly derives from specializing Prop. 32 in App. B to our context, observing that the strong convexity of $f(\cdot, \theta)$ is equivalent to the Lipschitz-smoothness of $f(\cdot, \theta)^*$ (see Lemma 30 in App. B). ∎

We now are ready to prove Prop. 3.

**Proposition 3** (Computation of an $\epsilon$-subgradient of $\mathcal{L}_Z$). *Let Asm. 1 hold and let $s_{\theta,n+1}$ be the output of Alg. 2 with $\theta \in \Theta$ over the dataset $Z$. Let $\nabla_\theta \in \partial\{-D_{n+1}(s_{\theta,n+1}, \cdot)\}(\theta)$, where*

$$D_{n+1}(s, \theta) = \sum_{i=1}^{n} \ell_i^*(s_i) + \lambda n f^*(\cdot, \theta)\Big(-\frac{1}{\lambda n} \sum_{i=1}^{n} x_i s_i\Big) \qquad s \in \mathbb{R}^n \tag{8}$$

*is the dual of the non-normalized Eq. (2). Then, $\nabla_\theta' = \nabla_\theta/n \in \partial_{\epsilon_\theta/n}\mathcal{L}_Z(\theta)$, with $\epsilon_\theta$ as in Prop. 2.*

**Proof.** We start from recalling that, for any $\theta \in \Theta$, the function $D_{n+1}(\cdot, \theta)$ reported in Eq. (8) is the objective of the dual problem associated to the non-normalized within-task problem in Eq. (2). Since in our assumptions, strong duality holds for this couple of problems (see Lemma 34 above), we can rewrite

$$\mathcal{L}_Z(\theta) = \max_{s \in \mathbb{R}^n} \tilde{D}_{n+1}(s, \theta) \qquad \tilde{D}_{n+1}(s, \theta) = -\frac{1}{n} D_{n+1}(s, \theta). \tag{70}$$

Thanks to Prop. 2, we know that the dual vector $s_{\theta,n+1}$ returned by Alg. 2 is an $\epsilon_\theta$-minimizer of the dual objective $D_{n+1}(\cdot, \theta)$, where $\epsilon_\theta$ is given in Prop. 2. Consequently, $s_{\theta,n+1}$ is an $(\epsilon_\theta/n)$-maximizer of the function $\tilde{D}_{n+1}(\cdot, \theta)$ defined above. We now observe that, for any $\theta' \in \Theta$, we have

$$\mathcal{L}_Z(\theta') = \max_{s \in \mathbb{R}^n} \tilde{D}_{n+1}(s, \theta') \geq \tilde{D}_{n+1}(s_{\theta,n+1}, \theta')$$

$$\geq \tilde{D}_{n+1}(s_{\theta,n+1}, \theta) + \left\langle \frac{\nabla_\theta}{n}, \theta' - \theta \right\rangle \geq \mathcal{L}_Z(\theta) - \frac{\epsilon_\theta}{n} + \left\langle \frac{\nabla_\theta}{n}, \theta' - \theta \right\rangle,$$

where, in the second inequality we have exploited the assumption $\nabla_\theta \in \partial\{-D_{n+1}(s_{\theta,n+1}, \cdot)\}(\theta)$, implying $\nabla_\theta/n \in \partial \tilde{D}_{n+1}(s_{\theta,n+1}, \cdot)(\theta)$, and in the last inequality we have used the fact that $s_{\theta,n+1}$ is an $(\epsilon_\theta/n)$-maximizer of the function $\tilde{D}_{n+1}(\cdot, \theta)$ as explained above and strong duality again. By definition of $\epsilon$-subgradients, the above inequality proves the desired statement. ∎

# E   Proofs of the statements in the statistical setting

In this section we report the proof of the transfer risk bound in Thm. 5 in Sec. 5. In order to do this, we require the following intermediate result.

**Proposition 35** (Online-to-batch conversion). *Under the same assumptions of Thm. 4, in expectation w.r.t. the sampling of the datasets $(Z_t)_{t=1}^T$, it holds that*

$$\mathbb{E}\, \mathcal{E}_{\text{stat}}^{\text{reg}}(\bar{w}_{\bar{\theta}}) \leq \mathbb{E}\, \frac{1}{nT}\, \mathcal{E}_{\text{meta}}^{\text{reg}}\big((Z_t)_{t=1}^T\big) + \mathbb{E}\, \mathbb{E}_{\mu \sim \rho}\, \mathbb{E}_{Z \sim \mu^n}\, \frac{1}{2\lambda n} \sum_{i=1}^n \frac{1}{i} \big\| x_i s'_{\bar{\theta}, i} \big\|_{\bar{\theta}, *}^2.$$

**Proof.** Throughout this proof, for any $\theta \in \Theta$, we will need to make explicit the dependency w.r.t. the dataset in the iterations $(w_{\theta,i})_{i=1}^n$ generated by Alg. 2, in their average $\bar{w}_\theta$ and in the regularized empirical risk minimizer

$$\hat{w}_\theta = \operatorname*{argmin}_{w \in \mathbb{R}^d} \mathcal{R}_{\theta,Z}(w) \qquad \mathcal{R}_{\theta,Z}(w) = \mathcal{R}_Z(w) + \lambda f(w, \theta). \tag{71}$$

Moreover, for any $\theta \in \Theta$ and any $\mu \sim \rho$, by arguments similar to the ones made for the existence of $\hat{w}_\theta$, exploiting Asm. 1, we manage to ensure the existence and the uniqueness of the regularized (true) risk minimizer

$$w_{\theta,\mu} = \operatorname*{argmin}_{w \in \mathbb{R}^d} \mathcal{R}_{\theta,\mu}(w) \qquad \mathcal{R}_{\theta,\mu}(w) = \mathcal{R}_\mu(w) + \lambda f(w, \theta). \tag{72}$$

In the sequel, we will also use also the short-hand notation

$$C = \mathbb{E}\, \mathbb{E}_{\mu \sim \rho}\, \mathbb{E}_{Z \sim \mu^n}\, \frac{1}{2\lambda n} \sum_{i=1}^n \frac{1}{i} \big\| x_i s'_{\bar{\theta}, i} \big\|_{\bar{\theta}, *}^2. \tag{73}$$

The desired statement can be written more explicitly as follows

$$\mathbb{E}\, \mathbb{E}_{\mu \sim \rho}\, \mathbb{E}_{Z \sim \mu^n}\, \mathcal{R}_{\bar{\theta}, \mu}(\bar{w}_{\bar{\theta}}(Z)) \leq \mathbb{E}\, \frac{1}{T} \sum_{t=1}^T \frac{1}{n} \sum_{i=1}^n \Big\{ \ell_{t,i}(\langle x_{t,i}, w_{\theta_t,i}(Z_t) \rangle) + \lambda f(w_{\theta_t,i}(Z_t), \theta_t) \Big\} + C. \tag{74}$$

In the following, we will explicitly write the expectation $\mathbb{E}$ in the statement above as

$$\mathbb{E}_{\mu_1, \ldots, \mu_T \sim \rho^T}\, \mathbb{E}_{Z_1 \sim \mu_1^n, \ldots, Z_T \sim \mu_T^n}. \tag{75}$$

We now prove Eq. (74). We start from observing that, for any dataset $Z \sim \mu^n$ and for any $\theta \in \Theta$ not depending on $Z$, recalling the subgradients $(s'_{\theta,i})_{i=1}^n$, $s'_{\theta,i} \in \partial \ell_i(\langle x_i, w_{\theta,i}(Z) \rangle)$, used by Alg. 2 over $Z$, we can write

$$
\begin{aligned}
\mathbb{E}_{Z \sim \mu^n} \left[ \mathcal{R}_{\theta,\mu}(\bar{w}_\theta(Z)) \right] &\leq \mathbb{E}_{Z \sim \mu^n} \frac{1}{n} \sum_{i=1}^n \mathcal{R}_{\theta,\mu}(w_{\theta,i}(Z)) \\
&= \mathbb{E}_{Z \sim \mu^n} \frac{1}{n} \sum_{i=1}^n \left\{ \ell_i(\langle x_i, w_{\theta,i}(Z) \rangle) + \lambda f(w_{\theta,i}(Z), \theta) \right\} \\
&\leq \mathbb{E}_{Z \sim \mu^n} \left[ \mathcal{L}_Z(\theta) + \frac{1}{2\lambda n} \sum_{i=1}^n \frac{1}{i} \left\| x_i s'_{\theta,i} \right\|_{\theta,*}^2 \right] \\
&= \mathbb{E}_{Z \sim \mu^n} \mathcal{L}_Z(\theta) + C.
\end{aligned}
\tag{76}
$$

In the first inequality above we have applied Jensen's inequality (see Lemma 15 in App. B) to the convex function $\mathcal{R}_{\theta,\mu}$, the first equality holds by standard online-to-batch arguments, more precisely, since $w_{\theta,i}(Z)$ depends only on the points $(z_j)_{j=1}^{i-1}$, thanks to the fact $Z \sim \mu^n$, we have

$$
\mathbb{E}_{Z \sim \mu^n} \mathcal{R}_{\theta,\mu}(w_{\theta,i}(Z)) = \mathbb{E}_{Z \sim \mu^n} \left[ \ell_i(\langle x_i, w_{\theta,i}(Z) \rangle) + \lambda f(w_{\theta,i}(Z), \theta) \right],
\tag{77}
$$

and, finally, the last inequality derives from exploiting the non-negativity of $\Delta_{\mathrm{Dual}}$ and moving the terms in Eq. (7) in Prop. 2. Hence, rewriting $\mathcal{L}_Z(\theta) = \mathcal{R}_{\theta,Z}(\hat{w}_\theta(Z))$, we can write the following

$$
\begin{aligned}
\mathbb{E}_{\mu_1,\dots,\mu_T \sim \rho^T} &\mathbb{E}_{Z_1 \sim \mu_1^n, \dots, Z_T \sim \mu_T^n} \mathbb{E}_{\mu \sim \rho} \mathbb{E}_{Z \sim \mu^n} \mathcal{R}_{\bar{\theta},\mu}(\bar{w}_{\bar{\theta}}(Z)) \\
&\leq \mathbb{E}_{\mu_1,\dots,\mu_T \sim \rho^T} \mathbb{E}_{Z_1 \sim \mu_1^n, \dots, Z_T \sim \mu_T^n} \mathbb{E}_{\mu \sim \rho} \mathbb{E}_{Z \sim \mu^n} \mathcal{R}_{\bar{\theta},Z}(\hat{w}_{\bar{\theta}}(Z)) + C \\
&\leq \mathbb{E}_{\mu_1,\dots,\mu_T \sim \rho^T} \mathbb{E}_{Z_1 \sim \mu_1^n, \dots, Z_T \sim \mu_T^n} \frac{1}{T} \sum_{t=1}^T \mathbb{E}_{\mu \sim \rho} \underbrace{\mathbb{E}_{Z \sim \mu^n} \mathcal{R}_{\theta_t,Z}(\hat{w}_{\theta_t}(Z))}_{} + C,
\end{aligned}
\tag{78}
$$

where, in the first inequality we have applied Eq. (76) with $\theta = \bar{\theta}$ and in the second inequality we have applied Jensen's inequality (see Lemma 15 in App. B) to the convex function $\mathbb{E}_{\mu \sim \rho} \mathbb{E}_{Z \sim \mu^n} \mathcal{L}_Z$. We now observe that, by definition of $\hat{w}_{\theta_t}(Z)$ and $w_{\theta_t,\mu}$, we can write the following

$$
\begin{aligned}
\underbrace{\mathbb{E}_{Z \sim \mu^n} \mathcal{R}_{\theta_t,Z}(\hat{w}_{\theta_t}(Z))}_{} &\leq \mathbb{E}_{Z \sim \mu^n} \mathcal{R}_{\theta_t,Z}(w_{\theta_t,\mu}) = \mathbb{E}_{Z \sim \mu^n} \mathcal{R}_{\theta_t,\mu}(w_{\theta_t,\mu}) \\
&\leq \underbrace{\mathbb{E}_{Z \sim \mu^n} \mathcal{R}_{\theta_t,\mu}(\bar{w}_{\theta_t}(Z))}_{}.
\end{aligned}
\tag{79}
$$

Substituting in Eq. (78), we can write the following

$$
\mathbb{E}_{\mu_1,\dots,\mu_T \sim \rho^T} \mathbb{E}_{Z_1 \sim \mu_1^n, \dots, Z_T \sim \mu_T^n} \mathbb{E}_{\mu \sim \rho} \mathbb{E}_{Z \sim \mu^n} \mathcal{R}_{\bar{\theta},\mu}(\bar{w}_{\bar{\theta}}(Z))
$$

$$
\leq \mathbb{E}_{\mu_1,\dots,\mu_T \sim \rho^T} \mathbb{E}_{Z_1 \sim \mu_1^n, \dots, Z_T \sim \mu_T^n} \frac{1}{T} \sum_{t=1}^T \mathbb{E}_{\mu \sim \rho} \underbrace{\mathbb{E}_{Z \sim \mu^n} \mathcal{R}_{\theta_t,Z}(\hat{w}_{\theta_t}(Z))}_{} + C
$$

$$
\leq \mathbb{E}_{\mu_1,\dots,\mu_T \sim \rho^T} \mathbb{E}_{Z_1 \sim \mu_1^n, \dots, Z_T \sim \mu_T^n} \frac{1}{T} \sum_{t=1}^T \mathbb{E}_{\mu \sim \rho} \underbrace{\mathbb{E}_{Z \sim \mu^n} \mathcal{R}_{\theta_t,\mu}(\bar{w}_{\theta_t}(Z))}_{} + C
$$

$$
= \frac{1}{T} \sum_{t=1}^T \mathbb{E}_{\mu_1,\dots,\mu_{t-1} \sim \rho^{t-1}} \mathbb{E}_{Z_1 \sim \mu_1^n, \dots, Z_{t-1} \sim \mu_{t-1}^n} \mathbb{E}_{\mu_t \sim \rho} \mathbb{E}_{Z_t \sim \mu_t^n} \mathcal{R}_{\theta_t,\mu_t}(\bar{w}_{\theta_t}(Z_t)) + C
$$

$$
\leq \frac{1}{T} \sum_{t=1}^T \mathbb{E}_{\mu_1,\dots,\mu_{t-1} \sim \rho^{t-1}} \mathbb{E}_{Z_1 \sim \mu_1^n, \dots, Z_{t-1} \sim \mu_{t-1}^n} \mathbb{E}_{\mu_t \sim \rho} \mathbb{E}_{Z_t \sim \mu_t^n} \frac{1}{n} \sum_{i=1}^n \mathcal{R}_{\theta_t,\mu_t}(w_{\theta_t,i}(Z_t)) + C
$$

$$
= \mathbb{E}_{\mu_1,\dots,\mu_T \sim \rho^T} \mathbb{E}_{Z_1 \sim \mu_1^n, \dots, Z_T \sim \mu_T^n} \frac{1}{T} \sum_{t=1}^T \frac{1}{n} \sum_{i=1}^n \left\{ \ell_{t,i}(\langle x_{t,i}, w_{\theta_t,i}(Z_t) \rangle) + \lambda f(w_{\theta_t,i}(Z_t), \theta_t) \right\} + C
$$

where, in the first equality we have exploited the fact that $\theta_t$ depends only on $(Z_j)_{j=1}^{t-1}$ and the i.i.d. sampling of the datasets, in the third inequality we have applied Jensen's inequality (see Lemma 15 in App. B) to the convex function $\mathcal{R}_{\theta_t,\mu_t}$ and, finally, in the second equality we have exploited the fact that $w_{\theta_t,i}(Z_t)$ depends only on the points $(z_{t,j})_{j=1}^{i-1}$ and, consequently, thanks to the fact $Z_t \sim \mu_t^n$, as already observed in Eq. (77),

$$\mathbb{E}_{Z_t \sim \mu_t^n} \, \mathcal{R}_{\theta_t,\mu_t}(w_{\theta_t,i}(Z_t)) = \mathbb{E}_{Z_t \sim \mu_t^n} \left[ \ell_{t,i}(\langle x_{t,i}, w_{\theta_t,i}(Z_t)\rangle) + \lambda f(w_{\theta_t,i}(Z_t),\theta_t) \right]. \qquad (80)$$

This coincides with the desired statement in Eq. (74). ∎

The above result in Prop. 35 is a different version of [1, Thm. 6.1] and [3, Thm. 3.3], where the authors give statistical guarantees for the meta-parameter defined by sampling uniformly from the whole pool of the meta-parameters $(\theta_t)_{t=1}^T$ returned by their method. Their result is consequently in expectation w.r.t. the data and w.r.t. this uniform sampling. On the contrary, in our case, leveraging on the convexity of our meta-objectives and the fact that we derived a *regularized* cumulative error bound for the inner algorithm (see Prop. 2), we have been able to obtain statistical guarantees for the average of the meta-parameters, without adding randomness and without the need of memorizing the previous meta-parameters. We now are ready to prove Thm. 5.

**Theorem 5** (Transfer risk bound). *Let the same assumptions in Thm. 4 hold in the i.i.d. statistical setting. Then, introducing the regularized transfer risk of the average $\bar{w}_{\bar{\theta}}$ of the iterates resulting from the combination of Alg. 2 and Alg. 3,*

$$\mathcal{E}_{\text{stat}}^{\text{reg}}(\bar{w}_{\bar{\theta}}) = \mathbb{E}_{\mu \sim \rho} \, \mathbb{E}_{Z \sim \mu^n} \left[ \mathcal{R}_\mu(\bar{w}_{\bar{\theta}}(Z)) + \lambda f(\bar{w}_{\bar{\theta}}(Z), \bar{\theta}) \right],$$

*for any $\theta \in \Theta$ such that $\mathbb{E}_{\mu \sim \rho} f(w_\mu, \theta) < +\infty$, the following upper bound holds in expectation w.r.t. the sampling of the datasets $(Z_t)_{t=1}^T$*

$$\mathbb{E} \, \mathcal{E}_{\text{stat}}^{\text{reg}}(\bar{w}_{\bar{\theta}}) \leq \mathcal{E}_\rho + \lambda \, \mathbb{E}_{\mu \sim \rho} f(w_\mu, \theta) + \frac{1}{2\lambda n T} \, \mathbb{E} \sum_{t=1}^T \sum_{i=1}^n \frac{1}{i} \big\| x_{t,i} s'_{\theta_t,i} \big\|_{\theta_t,*}^2$$

$$+ \frac{\eta F(\theta)}{T} + \frac{1}{2\eta T} \, \mathbb{E} \sum_{t=1}^T \big\| \big\| \nabla'_{\theta_t} \big\| \big\|_*^2 + \mathbb{E} \, \mathbb{E}_{\mu \sim \rho} \, \mathbb{E}_{Z \sim \mu^n} \frac{1}{2\lambda n} \sum_{i=1}^n \frac{1}{i} \big\| x_i s'_{\bar{\theta},i} \big\|_{\bar{\theta},*}^2.$$

**Proof.** The desired statement derives from combining Prop. 35 with the regularized cumulative error bound in Thm. 4 with $\hat{w}_t = w_{\mu_t}$ for any $t \in \{1, \dots, T\}$ and observing that, thanks to the definition of the vectors $\hat{w}_t$ and the i.i.d. sampling of the datasets, we can write

$$\mathbb{E}_{\mu_1,\dots,\mu_T \sim \rho^T} \, \mathbb{E}_{Z_1 \sim \mu_1^n, \dots, Z_T \sim \mu_T^n} \frac{1}{T} \sum_{t=1}^T f(\hat{w}_t, \theta) = \mathbb{E}_{\mu_1,\dots,\mu_T \sim \rho^T} \frac{1}{T} \sum_{t=1}^T f(w_{\mu_t}, \theta) \qquad (81)$$

$$= \mathbb{E}_{\mu \sim \rho} \, f(w_\mu, \theta)$$

$$\mathbb{E}_{\mu_1,\dots,\mu_T \sim \rho^T} \, \mathbb{E}_{Z_1 \sim \mu_1^n, \dots, Z_T \sim \mu_T^n} \frac{1}{T} \sum_{t=1}^T \mathcal{R}_{Z_t}(\hat{w}_t) = \frac{1}{T} \sum_{t=1}^T \mathbb{E}_{\mu_t \sim \rho} \, \mathbb{E}_{Z_t \sim \mu_t^n} \, \mathcal{R}_{Z_t}(w_{\mu_t}) \qquad (82)$$

$$= \mathbb{E}_{\mu \sim \rho} \, \mathbb{E}_{Z \sim \mu^n} \, \mathcal{R}_Z(w_\mu)$$

$$= \mathbb{E}_{\mu \sim \rho} \, \mathcal{R}_\mu(w_\mu)$$

where, in the last equality we have used the fact that $Z \sim \mu$ and the independence of $w_\mu$ on the data $Z$. ∎

# F  Fixed parameter in hindsight

In this section we report the results regarding the application of Alg. 2 with an appropriate meta-parameter fixed in hindsight for any task. In App. F.1 we will focus on the non-statistical setting, while in App. F.2 we will consider the statistical setting. These results will be used as benchmark to evaluate the quality of the meta-parameters estimated by our OWO Meta-Learning procedure.

## F.1 Non-statistical setting

In the next result we give a (regularized) cumulative error bound for the iterates generated by the application of Alg. 2 with an appropriate meta-parameter fixed in hindsight for any tasks. Such a bound will be compared to the corresponding bound we have obtained in Thm. 4 for our Meta-Learning procedure.

**Theorem 36** (Cumulative error bound with fixed meta-parameter in hindsight). *Let Asm. 1 hold. Consider a sequence of vectors $(\hat{w}_t)_{t=1}^T$ in $\mathbb{R}^d$ and any $\theta \in \Theta$ such that $f(\hat{w}_t, \theta) < +\infty$ for any $t \in \{1, \dots, T\}$. Let $(w_{t,i})_{i=1}^n$ be the iterates generated by Alg. 2 with a meta-parameter $\theta$ as above over the dataset $Z_t$, by means of the subgradients $(s'_{t,i})_{i=1}^n$, with $s'_{t,i} \in \partial \ell_{t,i}(\langle x_{t,i}, w_{t,i}\rangle)$. Then, the following upper bound holds*

$$\sum_{t=1}^T \mathcal{E}_{\text{inner}}^{\text{reg}}(Z_t, \theta) \leq nT\left(\frac{1}{T}\sum_{t=1}^T \mathcal{R}_{Z_t}(\hat{w}_t) + \frac{\lambda}{T}\sum_{t=1}^T f(\hat{w}_t, \theta) + \frac{1}{2\lambda nT}\sum_{t=1}^T \sum_{i=1}^n \frac{1}{i}\left\|x_{t,i}s'_{t,i}\right\|_{\theta,*}^2\right). \quad (83)$$

**Proof.** We start from observing that, according to Prop. 2, since $\Delta_{\text{Dual}} \geq 0$, by definition of $\mathcal{L}_t$ as minimum, we can write

$$\mathcal{E}_{\text{inner}}^{\text{reg}}(Z_t, \theta) \leq n\left(\mathcal{R}_{Z_t}(\hat{w}_t) + \lambda f(\hat{w}_t, \theta) + \frac{1}{2\lambda n}\sum_{i=1}^n \frac{1}{i}\left\|x_{t,i}s'_{t,i}\right\|_{\theta,*}^2\right). \quad (84)$$

The statement directly derives from summing over the datasets the above bound. ∎

We observe that the bound for our method in Thm. 4 is composed by two main parts: one part (the first row) is similar to the benchmark bound in Eq. (83), the other part (the second row) can be considered as the additional effort due to the estimation of the meta-parameter from the data. As we will see in the following, for the settings in Ex. 1 and Ex. 2, these additional terms can be made vanishing in the number of tasks $T$, by choosing in an appropriate way the hyper-parameter $\eta$.

## F.2 Statistical setting

In the next result we give a (regularized) transfer risk bound for the average of the iterates generated by the application of Alg. 2 with an appropriate meta-parameter fixed in hindsight for any tasks. Such a bound will be compared to the corresponding bound we have obtained in Thm. 5 for our Meta-Learning procedure.

**Theorem 37** (Transfer risk bound with fixed meta-parameter in hindsight). *Let Asm. 1 hold. Consider in the i.i.d. statistical setting any $\theta \in \Theta$ such that $\mathbb{E}_{\mu\sim\rho} f(w_\mu, \theta) < +\infty$. Let $\bar{w}_\theta$ be the average of the iterates $(w_{\theta,i})_{i=1}^n$ generated by Alg. 2 with a meta-parameter $\theta$ as above over the dataset $Z$, by means of the subgradients $(s'_{\theta,i})_{i=1}^n$, with $s'_{\theta,i} \in \partial \ell_{t,i}(\langle x_{t,i}, w_{\theta,i}\rangle)$. Then, the following upper bound holds*

$$\mathcal{E}_{\text{stat}}^{\text{reg}}(\bar{w}_\theta) \leq \mathcal{E}_\rho + \lambda\,\mathbb{E}_{\mu\sim\rho} f(w_\mu, \theta) + \frac{1}{2\lambda n}\,\mathbb{E}_{\mu\sim\rho}\,\mathbb{E}_{Z\sim\mu^n}\sum_{i=1}^n \frac{1}{i}\left\|x_i s'_{\theta,i}\right\|_{\theta,*}^2. \quad (85)$$

**Proof.** We start from observing that, according to Prop. 2, since $\Delta_{\text{Dual}} \geq 0$, by definition of $\mathcal{L}_Z$ as minimum, we can write

$$\frac{1}{n}\mathcal{E}_{\text{inner}}^{\text{reg}}(Z, \theta) = \frac{1}{n}\sum_{i=1}^n \left\{\ell_i\big(\langle x_i, w_{\theta,i}\rangle\big) + \lambda f\big(w_{\theta,i}, \theta\big)\right\}$$

$$\leq \mathcal{R}_Z(w_\mu) + \lambda f\big(w_\mu, \theta\big) + \frac{1}{2\lambda n}\sum_{i=1}^n \frac{1}{i}\left\|x_i s'_{\theta,i}\right\|_{\theta,*}^2. \quad (86)$$

Taking the expectation of the above bound w.r.t. $\mu \sim \rho$ and $Z \sim \mu^n$, recalling that, as already observed in Eq. (76),

$$\mathbb{E}_{Z\sim\mu^n}\left[\mathcal{R}_{\theta,\mu}(\bar{w}_\theta)\right] \leq \mathbb{E}_{Z\sim\mu^n} \frac{1}{n}\sum_{i=1}^n \left\{\ell_i(\langle x_i, w_{\theta,i}\rangle) + \lambda f(w_{\theta,i}, \theta)\right\} \quad (87)$$

and recalling that $\mathbb{E}_{Z\sim\mu^n} \mathcal{R}_Z(w_\mu) = \mathcal{R}_\mu(w_\mu)$, we obtain the desired statement. ∎

| **Algorithm 5** Within-task algorithm for Ex. 1 | **Algorithm 6** Meta-algorithm for Ex. 1 |
|---|---|
| **Input** $\lambda > 0, \theta \in \mathbb{R}^d, Z = (z_i)_{i=1}^n$ | **Input** $\eta > 0, (Z_t)_{t=1}^T$ |
| **Initialization** $s_{\theta,1} = (), w_{\theta,1} = \theta$ | **Initialization** $\theta_1 = 0$ |
| **For** $i = 1$ to $n$ | **For** $t = 1$ to $T$ |
| $\quad$ Receive the datapoint $z_i = (x_i, y_i)$ | $\quad$ Receive incrementally the dataset $Z_t$ |
| $\quad$ Compute $s'_{\theta,i} \in \partial \ell_i(\langle x_i, w_{\theta,i}\rangle) \subseteq \mathbb{R}$ | $\quad$ Run Alg. 5 with $\theta_t$ over $Z_t$ |
| $\quad$ Define $(s_{\theta,i+1})_i = s'_{\theta,i}, \gamma_i = \lambda(i+1)$ | $\quad$ Compute $s_{\theta_t, n+1}$ |
| $\quad$ Define $p_{\theta,i} = x_i s'_{\theta,i} + \lambda(w_{\theta,i} - \theta)$ | $\quad$ Define $\nabla'_{\theta_t} = X_t^\top s_{\theta_t, n+1}/n$ |
| $\quad$ Update $w_{\theta,i+1} = w_{\theta,i} - 1/\gamma_i\, p_{\theta,i}$ | $\quad$ Update $\theta_{t+1} = \theta_t - \nabla'_{\theta_t}/\eta$ |
| **Return** $(w_{\theta,i})_{i=1}^{n+1}, \bar{w}_\theta = \dfrac{1}{n} \sum_{i=1}^n w_{\theta,i}, s_{\theta,n+1}$ | **Return** $(\theta_t)_{t=1}^{T+1}, \bar{\theta} = \dfrac{1}{T} \sum_{t=1}^T \theta_t$ |

Looking at the bound in Thm. 5 for our method and the benchmark performance in Eq. (85), the conclusions and the comments we can derive are an adaptation to the statistical setting of the comments we have given above for the performance of our method in the non-statistical setting.

# G   Specializing to the bias in Ex. 1

In this chapter we specify our Meta-Learning framework to the setting in Ex. 1. We recall that, in such a case, the meta-parameter coincides with a bias vector $\theta \in \mathbb{R}^d$ and, as we will see in the following, the tasks' similarity translates into the existence of a bias vector closed to the tasks' target vectors. We start this chapter by specializing in App. G.1 our general OWO method to Ex. 1, deriving the corresponding inner and meta-algorithm. The method is then analyzed in App. G.2 and App. G.3, where we consider the non-statistical setting and the statistical setting, respectively. Finally, in App. G.4, we discuss the results.

## G.1   Deriving the method for Ex. 1

We start from specializing the generic inner algorithm in Alg. 2 and the generic meta-algorithm in Alg. 3 to the setting outlined in Ex. 1. The algorithms we obtain are reported in Alg. 5 and Alg. 6, respectively, where, $X_t \in \mathbb{R}^{n \times d}$ denotes the input vectors' matrix of the task $t$, having as $i$–th row the input vector $x_{t,i}$. The deduction is reported in Lemma 38 and Lemma 39 below, respectively.

We start from the deduction of the inner algorithm in Alg. 5.

**Lemma 38** (Derivation of the inner Alg. 5, bias)**.** *For any $i \in \{0, \dots, n\}$, let $w_{\theta,i+1}$ be the update of the (primal) variable deriving from applying Alg. 2 to the dataset $Z = (x_i, y_i)_{i=1}^n$ in the setting outlined in Ex. 1 with bias $\theta \in \mathbb{R}^d$. Let $s'_{\theta,i} \in \partial \ell_i(\langle x_i, w_{\theta,i}\rangle)$ be the subgradient used by such an algorithm to compute $w_{\theta,i+1}$. Then, $w_{\theta,1} = \theta$ and, for any $i \in \{1, \dots, n\}$, introducing the subgradient of the regularized loss*

$$p_{\theta,i} = x_i s'_{\theta,i} + \lambda(w_{\theta,i} - \theta) \in \partial\Big(\ell_i(\langle x_i, \cdot\rangle) + \frac{\lambda}{2}\,\|\cdot - \theta\|_2^2\Big)(w_{\theta,i}), \tag{88}$$

*we have*

$$w_{\theta,i+1} = w_{\theta,i} - \frac{1}{\lambda(i+1)}\,p_{\theta,i}. \tag{89}$$

**Proof.** We start from observing that, according to the choices made in Ex. 1, for any $\theta, w, u \in \mathbb{R}^d$, we have

$$f(w, \theta) = \frac{1}{2}\,\|w - \theta\|_2^2 \qquad f(\cdot, \theta)^*(u) = \frac{1}{2}\,\|u\|_2^2 + \langle u, \theta\rangle \qquad \nabla f(\cdot, \theta)^*(u) = u + \theta.$$

Consequently, according to the definition of $w_{\theta,1}$ in Alg. 2, we have

$$w_{\theta,1} = \nabla f(\cdot, \theta)^*(0) = \theta. \tag{90}$$

We now show the desired closed form of $w_{\theta,i+1}$ for any $i \in \{1, \ldots, n\}$. In such a case, denoting by $X_{1:i} \in \mathbb{R}^{i \times d}$ the matrix containing the first $i$ input vectors as rows, by definition of $w_{\theta,i+1}$ in Alg. 2, we can write

$$w_{\theta,i+1} = \nabla f(\cdot, \theta)^* \left( -\frac{1}{\lambda(i+1)} X_{1:i}^\top s_{\theta,i+1} \right) = -\frac{1}{\lambda(i+1)} X_{1:i}^\top s_{\theta,i+1} + \theta. \tag{91}$$

For $i = 1$ the statement holds, as a matter of fact, since $w_{\theta,1} = \theta$, exploiting Eq. (91) and introducing the subgradient $p_{\theta,1} = x_1 s'_{\theta,1} + \lambda(w_{\theta,1} - \theta) = x_1 s'_{\theta,1}$, we can write

$$w_{\theta,2} = -\frac{1}{2\lambda} x_1 s'_{\theta,1} + \theta = w_{\theta,1} - \frac{1}{2\lambda} p_{\theta,1}. \tag{92}$$

Now, we show that the statement holds also for $i \in \{2, \ldots, n\}$. Since $X_{1:i}^\top s_{\theta,i+1} = X_{1:i-1}^\top s_{\theta,i} + x_i s'_{\theta,i}$, we can write the following

$$
\begin{aligned}
w_{\theta,i+1} &= -\frac{1}{\lambda(i+1)} X_{1:i}^\top s_{\theta,i+1} + \theta = -\frac{1}{\lambda(i+1)} \left( X_{1:i-1}^\top s_{\theta,i} + x_i s'_{\theta,i} \right) + \theta \\
&= \frac{\lambda i}{\lambda(i+1)} \left( -\frac{1}{\lambda i} X_{1:i-1}^\top s_{\theta,i} \right) - \frac{x_i s'_{\theta,i}}{\lambda(i+1)} + \theta \\
&= \frac{\lambda(i+1)(w_{\theta,i} - \theta) - x_i s'_{\theta,i} - \lambda(w_{\theta,i} - \theta)}{\lambda(i+1)} + \theta \\
&= \frac{\lambda(i+1) w_{\theta,i} - p_{\theta,i}}{\lambda(i+1)} = w_{\theta,i} - \frac{1}{\lambda(i+1)} p_{\theta,i},
\end{aligned}
\tag{93}
$$

where, in the first and the fourth equality, we have exploited Eq. (91) and in the fifth equality we have exploited the form of the subgradient $p_{\theta,i} = x_i s'_{\theta,i} + \lambda(w_{\theta,i} - \theta)$. ∎

We now proceed with the deduction of the meta-algorithm in Alg. 6.

**Lemma 39** (Derivation of the meta-algorithm in Alg. 6, bias)**.** *For any $t \in \{0, \ldots, T\}$, let $\theta_{t+1}$ be the update of the variable deriving from applying Alg. 3 to the data $(Z_t)_{t=1}^T$ in the setting outlined in Ex. 1. Let $\nabla'_{\theta_t}$ be the approximated meta-subgradient computed as described in Prop. 3 and used by the algorithm to compute $\theta_{t+1}$. Then, $\theta_1 = 0 \in \mathbb{R}^d$ and, for any $t \in \{1, \ldots, T\}$, we have*

$$\theta_{t+1} = \theta_t - \frac{1}{\eta} \nabla'_{\theta_t}. \tag{94}$$

*Moreover, for any $t \in \{1, \ldots, T\}$, we have*

$$\nabla'_{\theta_t} = \frac{1}{n} X_t^\top s_{\theta_t, n+1}, \tag{95}$$

*where $s_{\theta_t, n+1} \in \mathbb{R}^n$ is the output of Alg. 6 with bias vector $\theta_t$ over the dataset $Z_t$ and, under Asm. 3,*

$$\left\| \nabla'_{\theta_t} \right\|_2^2 \le L^2 \|C_t\|_\infty. \tag{96}$$

**Proof.** We start from observing that, according to the choices made in Ex. 1, for any $k, \theta, u \in \mathbb{R}^d$, we have

$$F(\theta) = \frac{1}{2} \|\theta\|_2^2 \qquad F^*(k) = \frac{1}{2} \|k\|_2^2 \qquad \nabla F^*(k) = k \qquad f(\cdot, \theta)^*(u) = \frac{1}{2} \|u\|_2^2 + \langle u, \theta \rangle.$$

Consequently, according to the definition of $\theta_1$ in Alg. 3, we have

$$\theta_1 = \nabla F^*(0) = 0. \tag{97}$$

We now show the desired closed form of $\theta_{t+1}$, for any $t \in \{1, \ldots, T\}$. In such a case, by the definition of $\theta_{t+1}$ in Alg. 3, we can write

$$\theta_{t+1} = \nabla F^* \left( -\frac{1}{\eta} \sum_{j=1}^t \nabla'_{\theta_j} \right) = -\frac{1}{\eta} \sum_{j=1}^t \nabla'_{\theta_j}. \tag{98}$$

For $t = 1$ the statement holds, as a matter of fact, since $\theta_1 = 0$, exploiting Eq. (98), we can write

$$\theta_2 = -\frac{1}{\eta} \nabla'_{\theta_1} = \theta_1 - \frac{1}{\eta} \nabla'_{\theta_1}. \tag{99}$$

For $t \in \{2, \ldots, T\}$, we observe that, according to Eq. (98), we have

$$\theta_{t+1} = -\frac{1}{\eta} \sum_{j=1}^{t} \nabla'_{\theta_j} = -\frac{1}{\eta} \sum_{j=1}^{t-1} \nabla'_{\theta_j} - \frac{1}{\eta} \nabla'_{\theta_t} = \theta_t - \frac{1}{\eta} \nabla'_{\theta_t}. \tag{100}$$

We now specify the closed form of the approximated meta-subgradients, computed as described in Prop. 3 for Ex. 1. We start from observing that adding to the notation in Prop. 3 the further task index $t$, by strong duality (see Lemma 34), we can rewrite

$$\mathcal{L}_t(\theta) = \max_{s \in \mathbb{R}^n} \tilde{D}_{t,n+1}(s, \theta) \qquad \tilde{D}_{t,n+1}(s, \theta) = -\frac{1}{n} D_{t,n+1}(s, \theta) \tag{101}$$

where, according to Eq. (8), in the setting outlined in Ex. 1,

$$
\begin{aligned}
-D_{t,n+1}(s, \theta) &= -\sum_{i=1}^{n} \ell^*_{t,i}(s_i) - \lambda n f(\cdot, \theta)^* \Big( -\frac{1}{\lambda n} \sum_{i=1}^{n} x_{t,i} s_i \Big) \\
&= -\sum_{i=1}^{n} \ell^*_{t,i}(s_i) - \lambda n f(\cdot, \theta)^* \Big( -\frac{1}{\lambda n} X_t^\top s \Big) \\
&= -\sum_{i=1}^{n} \ell^*_{t,i}(s_i) - \frac{1}{2\lambda n} \big\| X_t^\top s \big\|_2^2 + \big\langle X_t^\top s, \theta \big\rangle.
\end{aligned} \tag{102}
$$

Consequently, recalling that the output $s_{\theta_t, n+1}$ of the inner algorithm coincides with the last iterate of the corresponding dual inner iteration, according to Prop. 3, we have

$$\nabla_{\theta_t} = X_t^\top s_{\theta_t, n+1} \tag{103}$$

and, consequently,

$$\nabla'_{\theta_t} = \nabla_{\theta_t}/n \in \partial_{\epsilon_{\theta_t}/n} \mathcal{L}_t(\theta_t), \tag{104}$$

where $\epsilon_{\theta_t}$ is outlined in Prop. 3 and it must be specified to Ex. 1. In order to prove Eq. (96), we start from observing that $s_{\theta_t, n+1}$ is the vector in $\mathbb{R}^n$ having as component $i$ the subgradient $s'_{\theta_t, i} \in \partial \ell_{t,i}(\langle x_{t,i}, w_{\theta_t, i} \rangle)$. Hence, under Asm. 3, exploiting Lemma 25 in App. B, any component of $s_{\theta_t, n+1}$ is absolutely bounded by $L$, and, consequently, $\| s_{\theta_t, n+1} \|_2 \leq L\sqrt{n}$. This allows us to get the desired bound by applying Holder's inequality (see Lemma 8 in App. B) to the matrices' scalar product as follows

$$
\begin{aligned}
\big\| \nabla'_{\theta_t} \big\|_2^2 &= \frac{1}{n} \operatorname{Tr}\Big( \frac{1}{n} \sum_{i=1}^{n} x_{t,i} x_{t,i}^\top s_{\theta_t, n+1} s_{\theta_t, n+1}^\top \Big) \leq \frac{1}{n} \Big\| \frac{1}{n} \sum_{i=1}^{n} x_{t,i} x_{t,i}^\top \Big\|_\infty \| s_{\theta_t, n+1} \|_2^2 \\
&\leq L^2 \Big\| \frac{1}{n} \sum_{i=1}^{n} x_{t,i} x_{t,i}^\top \Big\|_\infty = L^2 \| C_t \|_\infty,
\end{aligned}
$$

where in the last equality we have introduced the definition of $C_t$. ∎

We observe that the inner Alg. 5 we have deduced is a slightly different version of the inner algorithm used in [11] in the statistical setting, where the step size decreases as $1/(\lambda i)$ instead of $1/(\lambda(i+1))$. Instead, the meta-algorithm in Alg. 6 we have retrieved is exactly the same analyzed in that work. We refer to the discussion in App. A for more details about that work.

We also observe that for the setting in Ex. 1, our method in Alg. 5 and Alg. 6 scales linearly with the dimension of the input space. Thus, it will be appropriate also for datasets in more rich observation spaces, such as [41].

In the next section, we analyze the performance of our OWO Meta-Learning method applied to Ex. 1, in the non-statistical setting.

## G.2 Analysis of the method in the non-statistical setting for Ex. 1

In the next result we specify Thm. 4 to Ex. 1, that is, we provide a (regularized) cumulative error bound for the procedure deriving from combining Alg. 5 with Alg. 6.

**Corollary 40** (Cumulative error bound, bias). *Let Asm. 3 hold and consider the setting in Thm. 4 applied to Ex. 1. Then, introducing the empirical variance of the vectors $(\hat{w}_t)_{t=1}^T$ w.r.t. a bias vector $\theta \in \mathbb{R}^d$*

$$\hat{V}(\theta) = \frac{1}{2T} \sum_{t=1}^{T} \|\hat{w}_t - \theta\|_2^2, \tag{105}$$

*the following (regularized) cumulative error bound holds for any $\theta \in \mathbb{R}^d$*

$$\mathcal{E}_{\text{meta}}^{\text{reg}}\big((Z_t)_{t=1}^T\big) \leq nT \left( \frac{1}{T} \sum_{t=1}^{T} \mathcal{R}_{Z_t}(\hat{w}_t) + \lambda \hat{V}(\theta) + \frac{L^2 \text{Tr}(\hat{C}^{\text{tot}})}{2\lambda n} + \frac{\eta \|\theta\|_2^2}{2T} + \frac{L^2 \|C^{\text{tot}}\|_{\infty,1}}{2\eta} \right). \tag{106}$$

*Hence, optimizing w.r.t. the hyper-parameters $\lambda$ and $\eta$, for*

$$\lambda = L \sqrt{\frac{\text{Tr}(\hat{C}^{\text{tot}})}{2n\hat{V}(\theta)}} \qquad \eta = \frac{L \sqrt{T \|C^{\text{tot}}\|_{\infty,1}}}{\|\theta\|_2}, \tag{107}$$

*we get*

$$\mathcal{E}_{\text{meta}}^{\text{reg}}\big((Z_t)_{t=1}^T\big) \leq nT \left( \frac{1}{T} \sum_{t=1}^{T} \mathcal{R}_{Z_t}(\hat{w}_t) + L \left( \sqrt{\frac{2\hat{V}(\theta)\text{Tr}(\hat{C}^{\text{tot}})}{n}} + \|\theta\|_2 \sqrt{\frac{\|C^{\text{tot}}\|_{\infty,1}}{T}} \right) \right).$$

**Proof.** Specializing Thm. 4 to the quantities outlined in Ex. 1, exploiting the bound on the norm of the approximated meta-subgradients given in Eq. (96) (exploiting Asm. 3) and using the notation in Eq. (105), for any $\theta \in \mathbb{R}^d$, we get

$$\mathcal{E}_{\text{meta}}^{\text{reg}}\big((Z_t)_{t=1}^T\big) \leq nT \left( \frac{1}{T} \sum_{t=1}^{T} \mathcal{R}_{Z_t}(\hat{w}_t) + \lambda \hat{V}(\theta) + \frac{1}{2\lambda nT} \sum_{t=1}^{T} \sum_{i=1}^{n} \frac{1}{i} \left\| x_{t,i} s'_{\theta_t,i} \right\|_2^2 \right.$$
$$\left. + \frac{\eta \|\theta\|_2^2}{2T} + \frac{L^2 \|C^{\text{tot}}\|_{\infty,1}}{2\eta} \right). \tag{108}$$

The statement derives from the above inequality observing that, under Asm. 3, using the definition of $\hat{C}^{\text{tot}}$, we can write

$$\frac{1}{T} \sum_{t=1}^{T} \sum_{i=1}^{n} \frac{1}{i} \left\| x_{t,i} s'_{\theta_t,i} \right\|_2^2 \leq L^2 \text{Tr}\Big( \frac{1}{T} \sum_{t=1}^{T} \sum_{i=1}^{n} \frac{1}{i} x_{t,i} x_{t,i}^\top \Big) = L^2 \text{Tr}(\hat{C}^{\text{tot}}). \tag{109}$$

■

In order to evaluate the quality of the bound above, we specify Thm. 36 to Ex. 1, that is, we provide a (regularized) cumulative error bound for the procedure deriving from running the within-task Alg. 5 with a bias vector fixed in hindsight for any task.

**Corollary 41** (Cumulative error bound with fixed meta-parameter in hindsight, bias). *Let Asm. 3 hold and consider the setting in Thm. 36 applied to Ex. 1. Then, according to the notation in Eq. (105), the following (regularized) cumulative error bound holds for any $\theta \in \mathbb{R}^d$*

$$\sum_{t=1}^{T} \mathcal{E}_{\text{inner}}^{\text{reg}}(Z_t, \theta) \leq nT \left( \frac{1}{T} \sum_{t=1}^{T} \mathcal{R}_{Z_t}(\hat{w}_t) + \lambda \hat{V}(\theta) + \frac{L^2 \text{Tr}(\hat{C}^{\text{tot}})}{2\lambda n} \right). \tag{110}$$

*Hence, optimizing w.r.t. the hyper-parameter $\lambda$, for*

$$\lambda = L \sqrt{\frac{\text{Tr}(\hat{C}^{\text{tot}})}{2n\hat{V}(\theta)}}, \tag{111}$$

*we get*

$$\sum_{t=1}^{T} \mathcal{E}_{\text{inner}}^{\text{reg}}(Z_t, \theta) \leq nT \left( \frac{1}{T} \sum_{t=1}^{T} \mathcal{R}_{Z_t}(\hat{w}_t) + L \sqrt{\frac{2\hat{V}(\theta)\text{Tr}(\hat{C}^{\text{tot}})}{n}} \right). \tag{112}$$

**Proof.** Specializing Thm. 36 to the quantities outlined in Ex. 1, using the notation in Eq. (105), for any $\theta \in \mathbb{R}^d$, we get

$$\sum_{t=1}^{T} \mathcal{E}_{\text{inner}}^{\text{reg}}(Z_t, \theta) \leq nT \left( \frac{1}{T} \sum_{t=1}^{T} \mathcal{R}_{Z_t}(\hat{w}_t) + \lambda \hat{V}(\theta) + \frac{1}{2\lambda nT} \sum_{t=1}^{T} \sum_{i=1}^{n} \frac{1}{i} \left\| x_{t,i} s'_{t,i} \right\|_2^2 \right). \quad (113)$$

The statement derives from the above inequality observing that, under Asm. 3, using the definition of $\hat{C}^{\text{tot}}$, we can write

$$\frac{1}{T} \sum_{t=1}^{T} \sum_{i=1}^{n} \frac{1}{i} \left\| x_{t,i} s'_{t,i} \right\|_2^2 \leq L^2 \text{Tr}\left( \frac{1}{T} \sum_{t=1}^{T} \sum_{i=1}^{n} \frac{1}{i} x_{t,i} x_{t,i}^\top \right) = L^2 \text{Tr}(\hat{C}^{\text{tot}}). \quad (114)$$

∎

We postpone to App. G.4 a discussion about the results we reported above. In the next section, we analyze the performance of our OWO Meta-Learning method applied to Ex. 1, in the statistical setting.

### G.3 Analysis of the method in the statistical setting for Ex. 1

In the result below we specify Thm. 5 to Ex. 1, that is, we provide a (regularized) transfer risk bound for the average $\bar{w}_{\bar{\theta}}$ of the estimators returned by the combination of Alg. 5 with Alg. 6.

**Corollary 42** (Transfer risk bound, bias)**.** *Let Asm. 3 hold and consider the statistical setting in Thm. 5 applied to Ex. 1. Then, introducing the exact variance of the vectors $w_\mu$ w.r.t. a bias vector $\theta \in \mathbb{R}^d$*

$$V_\rho(\theta) = \frac{1}{2} \, \mathbb{E}_{\mu \sim \rho} \left\| w_\mu - \theta \right\|_2^2, \quad (115)$$

*the following (regularized) transfer risk bound holds for any $\theta \in \mathbb{R}^d$*

$$\mathbb{E} \, \mathcal{E}_{\text{stat}}^{\text{reg}}(\bar{w}_{\bar{\theta}}) \leq \mathcal{E}_\rho + \lambda V_\rho(\theta) + \frac{(\log(n)+1)L^2 \text{Tr}(C_\rho)}{\lambda n} + \frac{\eta \|\theta\|_2^2}{2T} + \frac{L^2 \mathbb{E} \left\| C^{\text{tot}} \right\|_{\infty,1}}{2\eta}. \quad (116)$$

*Hence, optimizing w.r.t. the hyper-parameters $\lambda$ and $\eta$, for*

$$\lambda = L \sqrt{\frac{(\log(n)+1)\text{Tr}(C_\rho)}{n V_\rho(\theta)}} \qquad \eta = \frac{L \sqrt{T \, \mathbb{E} \left\| C^{\text{tot}} \right\|_{\infty,1}}}{\|\theta\|_2}, \quad (117)$$

*we get*

$$\mathbb{E} \, \mathcal{E}_{\text{stat}}^{\text{reg}}(\bar{w}_{\bar{\theta}}) \leq \mathcal{E}_\rho + L \left( 2 \sqrt{\frac{(\log(n)+1) V_\rho(\theta) \text{Tr}(C_\rho)}{n}} + \|\theta\|_2 \sqrt{\frac{\mathbb{E} \left\| C^{\text{tot}} \right\|_{\infty,1}}{T}} \right). \quad (118)$$

**Proof.** Specializing Thm. 5 to the quantities outlined in Ex. 1, exploiting the bound on the norm of the approximated meta-subgradients given in Eq. (96) (exploiting Asm. 3) and using the notation in Eq. (115), the following bound holds for any $\theta \in \mathbb{R}^d$

$$\mathbb{E} \, \mathcal{E}_{\text{stat}}^{\text{reg}}(\bar{w}_{\bar{\theta}}) \leq \mathcal{E}_\rho + \lambda V_\rho(\theta) + \frac{1}{2\lambda nT} \, \mathbb{E} \sum_{t=1}^{T} \sum_{i=1}^{n} \frac{1}{i} \left\| x_{t,i} s'_{\theta_t,i} \right\|_2^2$$

$$+ \frac{\eta \|\theta\|_2^2}{2T} + \frac{L^2 \mathbb{E} \left\| C^{\text{tot}} \right\|_{\infty,1}}{2\eta} + \frac{1}{2\lambda n} \mathbb{E} \, \mathbb{E}_{\mu \sim \rho} \, \mathbb{E}_{Z \sim \mu^n} \sum_{i=1}^{n} \frac{1}{i} \left\| x_i s'_{\bar{\theta},i} \right\|_2^2.$$

The desired statement derives from the above inequality and from observing that, thanks to Asm. 3 and the i.i.d. sampling of the data, using the inequality $\sum_{i=1}^{n} 1/i \leq \log(n) + 1$ and the definition of $C_\rho$, we have

$$\mathbb{E} \, \frac{1}{T} \sum_{t=1}^{T} \sum_{i=1}^{n} \frac{1}{i} \left\| x_{t,i} s'_{\theta_t,i} \right\|_2^2 \leq L^2 \mathbb{E} \, \text{Tr}\left( \frac{1}{T} \sum_{t=1}^{T} \sum_{i=1}^{n} \frac{1}{i} x_{t,i} x_{t,i}^\top \right) \leq L^2 (\log(n)+1)\text{Tr}(C_\rho)$$

$$\mathbb{E} \, \mathbb{E}_{\mu \sim \rho} \, \mathbb{E}_{Z \sim \mu^n} \sum_{i=1}^{n} \frac{1}{i} \left\| x_i s'_{\bar{\theta},i} \right\|_2^2 \leq L^2 \mathbb{E}_{\mu \sim \rho} \, \mathbb{E}_{Z \sim \mu^n} \, \text{Tr}\left( \sum_{i=1}^{n} \frac{1}{i} x_i x_i^\top \right) \leq L^2 (\log(n)+1)\text{Tr}(C_\rho).$$

∎

In order to evaluate the quality of the bound above, we specify Thm. 37 to Ex. 1, that is, we provide a (regularized) transfer risk bound for $\bar{w}_\theta$, the average of the iterations returned by running the within-task Alg. 5 with bias vector $\theta$ fixed in hindsight for any task.

**Corollary 43** (Transfer risk bound with fixed meta-parameter in hindsight, bias). *Let Asm. 3 hold and consider the statistical setting in Thm. 37 applied to Ex. 1. Then, according to the notation in Eq. (115), the following (regularized) transfer risk bound holds for any $\theta \in \mathbb{R}^d$*

$$\mathcal{E}_{\text{stat}}^{\text{reg}}(\bar{w}_{\bar{\theta}}) \le \mathcal{E}_\rho + \lambda V_\rho(\theta) + \frac{L^2(\log(n)+1)\text{Tr}(C_\rho)}{2\lambda n}. \tag{119}$$

*Hence, optimizing w.r.t. the hyper-parameter $\lambda$, for*

$$\lambda = L \sqrt{\frac{(\log(n)+1)\text{Tr}(C_\rho)}{2nV_\rho(\theta)}}, \tag{120}$$

*we get*

$$\mathcal{E}_{\text{stat}}^{\text{reg}}(\bar{w}_{\bar{\theta}}) \le \mathcal{E}_\rho + L \sqrt{\frac{2(\log(n)+1)V_\rho(\theta)\text{Tr}(C_\rho)}{n}}. \tag{121}$$

**Proof.** Specializing Thm. 37 to the quantities outlined in Ex. 1, using the notation in Eq. (115), for any $\theta \in \mathbb{R}^d$, we get

$$\mathcal{E}_{\text{stat}}^{\text{reg}}(\bar{w}_{\bar{\theta}}) \le \mathcal{E}_\rho + \lambda V_\rho(\theta) + \frac{1}{2\lambda n} \, \mathbb{E}_{\mu\sim\rho} \, \mathbb{E}_{Z\sim\mu^n} \, \sum_{i=1}^n \frac{1}{i} \, \left\| x_i s_{\theta,i}' \right\|_2^2.$$

The statement derives from the above inequality observing that, under Asm. 3, exploiting the i.i.d. sampling of the data and the inequality $\sum_{i=1}^n 1/i \le \log(n) + 1$, introducing the definition of $C_\rho$, we can write

$$\mathbb{E}_{\mu\sim\rho} \, \mathbb{E}_{Z\sim\mu^n} \, \sum_{i=1}^n \frac{1}{i} \, \left\| x_i s_{\theta,i}' \right\|_2^2 \le L^2 \mathbb{E}_{\mu\sim\rho} \, \mathbb{E}_{Z\sim\mu^n} \, \text{Tr}\Big(\sum_{i=1}^n \frac{1}{i} \, x_i x_i^\top\Big) \le L^2(\log(n)+1) \, \text{Tr}(C_\rho). \quad \blacksquare$$

Also in this case, the comments to the bounds above are postponed in the following App. G.4.

### G.4 Discussion of the results for Ex. 1

We start from discussing the results in Cor. 41 and Cor. 43, where the bias vector used by the inner algorithm is fixed in hindsight for any task.

#### G.4.1 Advantage of selecting the right bias

Looking at the bounds in Cor. 41 and Cor. 43, we can state that the advantage in using one bias vector $\theta \in \mathbb{R}^d$ in comparison to the others is determined by the associated empirical variance $\hat{V}(\theta)$ in Cor. 41 or by the corresponding exact variance $V_\rho(\theta)$ in Cor. 43. This inspires us to consider as the best algorithm in our class (*oracle*) the algorithm associated to the bias vector minimizing the above quantities:

$$\hat{\theta} = \operatorname*{argmin}_{\theta\in\mathbb{R}^d} \hat{V}(\theta) = \frac{1}{T} \sum_{t=1}^T \hat{w}_t, \tag{122}$$

for the non-statistical setting in Cor. 41, and

$$\theta_\rho = \operatorname*{argmin}_{\theta\in\mathbb{R}^d} V_\rho(\theta) = \mathbb{E}_{\mu\sim\rho} \, w_\mu, \tag{123}$$

for the statistical setting in Cor. 43. In the following, we will consider these two reasonable vectors as benchmark in order to evaluate the quality of the bias returned by our Meta-Learning procedure.

On the other hand, solving the tasks independently (ITL), in this case, corresponds to the unbiased case, i.e. to the application of the inner Alg. 5 with bias $\theta_{\text{ITL}} = 0 \in \mathbb{R}^d$ for any task. In particular,

from the above bounds, we can say that there is an advantage in using the optimal bias w.r.t. solving each task independently, when the tasks are *similar* in the sense that the variance of the associated target vectors is much smaller than their second moment, i.e. when

$$\hat{V}(\hat{\theta}) = \min_{\theta \in \mathbb{R}^d} \frac{1}{2T} \sum_{t=1}^{T} \|\hat{w}_t - \theta\|_2^2 = \frac{1}{2T} \sum_{t=1}^{T} \|\hat{w}_t - \hat{\theta}\|_2^2 \ll \frac{1}{2T} \sum_{t=1}^{T} \|\hat{w}_t\|_2^2 = \hat{V}(0) \qquad (124)$$

for the non-statistical setting and

$$V_\rho(\theta_\rho) = \min_{\theta \in \mathbb{R}^d} \mathbb{E}_{\mu \sim \rho} \frac{1}{2} \|w_\mu - \theta\|_2^2 = \mathbb{E}_{\mu \sim \rho} \frac{1}{2} \|w_\mu - \theta_\rho\|_2^2 \ll \mathbb{E}_{\mu \sim \rho} \frac{1}{2} \|w_\mu\|_2^2 = V_\rho(0) \qquad (125)$$

for the statistical setting.

We now can make the following observations about the bounds we have obtained in Cor. 40 and Cor. 42 for our Meta-Learning procedures.

### G.4.2 Bias resulting from our Meta-Learning method

Looking at the bounds in Cor. 40 and Cor. 42, we can state that our Meta-Learning methods are effective, because, when the number of training tasks is sufficiently large w.r.t. the number of points $n$ (hence the term $T^{-1/2}$ is negligible), with an appropriate tuning of the hyper-parameters $\lambda$ and $\eta$, the bias vector estimated by our methods can provide comparable guarantees as those for the corresponding best bias vector in hindsight in Cor. 41 and Cor. 43. As a consequence, when the tasks are similar as explained above, our methods can provide a significant advantage w.r.t. ITL. These observations are in line with [11], where we only consider the statistical setting and we present the same bound in Cor. 42 with slightly worse constants.

## H  Specializing to the feature map in Ex. 2

In this chapter we specify our Meta-Learning framework to the setting in Ex. 2. We recall that, in such a case, the meta-parameter coincides with a linear feature map $\theta \in \mathbb{S}_+^d$ and, as we will see in the following, the tasks' similarity translates into the existence of a low-rank linear feature map containing in its range the tasks' target vectors. We start this chapter by specializing in App. H.1 our general OWO method to Ex. 2, deriving the corresponding inner and meta-algorithm. The method is then analyzed in App. H.2 and App. H.3, where we consider the non-statistical setting and the statistical setting, respectively. Finally in App. H.4, we discuss the results.

### H.1  Deriving the method for Ex. 2

We start from specializing the generic inner algorithm in Alg. 2 and the generic meta-algorithm in Alg. 3 to the setting outlined in Ex. 2. The algorithms we obtain are reported in Alg. 7 and Alg. 8, respectively, where, $\text{proj}_{\mathcal{S}}$ is the Euclidean projection over the set $\mathcal{S}$ and we recall that $X_t \in \mathbb{R}^{n \times d}$ denotes the input vectors' matrix of the task $t$, having as $i$–th row the input vector $x_{t,i}$. The deduction is reported in Lemma 44 and Lemma 45 below, respectively.

We start from the deduction of the inner-algorithm in Alg. 7.

**Lemma 44** (Derivation of the inner Alg. 7, feature map)**.** *For any $i \in \{0, \dots, n\}$, let $w_{\theta,i+1}$ be the update of the (primal) variable deriving from applying Alg. 2 to the dataset $Z = (x_i, y_i)_{i=1}^{n}$ in the setting outlined in Ex. 2 with feature map $\theta \in \mathcal{S}$. Let $s'_{\theta,i} \in \partial \ell_i(\langle x_i, w_{\theta,i} \rangle)$ be the subgradient used by such an algorithm to compute $w_{\theta,i+1}$. Then, $w_{\theta,i+1} \in \text{Ran}(\theta)$. Moreover, $w_{\theta,1} = 0 \in \mathbb{R}^d$ and, for any $i \in \{1, \dots, n\}$, introducing the subgradient of the regularized loss*

$$p_{\theta,i} = x_i s'_{\theta,i} + \lambda \theta^\dagger w_{\theta,i} \in \partial \Big( \ell_i(\langle x_i, \cdot \rangle) + \frac{\lambda}{2} \langle \cdot, \theta^\dagger \cdot \rangle \Big)(w_{\theta,i}), \qquad (126)$$

*we have*

$$w_{\theta,i+1} = w_{\theta,i} - \frac{1}{\lambda(i+1)} \big( \theta x_i s'_{\theta,i} + \lambda w_{\theta,i} \big) = w_{\theta,i} - \frac{1}{\lambda(i+1)} \theta p_{\theta,i}. \qquad (127)$$

| **Algorithm 7** Within-task algorithm for Ex. 2 | **Algorithm 8** Meta-algorithm for Ex. 2 |
|---|---|

**Input** $\lambda > 0, \theta \in \mathcal{S}, Z = (z_i)_{i=1}^n$

**Initialization** $s_{\theta,1} = (), w_{\theta,1} = 0$

**For** $i = 1$ to $n$

    Receive the datapoint $z_i = (x_i, y_i)$

    Compute $s'_{\theta,i} \in \partial\ell_i(\langle x_i, w_{\theta,i}\rangle) \subseteq \mathbb{R}$

    Define $(s_{\theta,i+1})_i = s'_{\theta,i}, \gamma_i = \lambda(i+1)$

    Define $p_{\theta,i} = x_i s'_{\theta,i} + \lambda\theta^\dagger w_{\theta,i}$

    Update $w_{\theta,i+1} = w_{\theta,i} - 1/\gamma_i\, \theta p_{\theta,i}$

**Return** $(w_{\theta,i})_{i=1}^{n+1}, \bar{w}_\theta = \dfrac{1}{n}\displaystyle\sum_{i=1}^n w_{\theta,i}, \ s_{\theta,n+1}$

---

**Input** $\eta > 0, (Z_t)_{t=1}^T, \theta_0 \in \mathcal{S}$

**Initialization** $\theta_1 = \theta_0, P_1 = 0 \in \mathbb{S}^d$

**For** $t = 1$ to $T$

    Receive incrementally the dataset $Z_t$

    Run Alg. 7 with $\theta_t$ over $Z_t$

    Compute $s_{\theta_t,n+1}$

    Define $\nabla'_{\theta_t} = -\dfrac{q_t q_t^\top}{2\lambda n^2}\quad q_t = X_t^\top s_{\theta_t,n+1}$

    Update $P_{t+1} = P_t + \nabla'_{\theta_t}$

    Update $\theta_{t+1} = \mathrm{proj}_{\mathcal{S}}\left(-P_{t+1}/\eta + \theta_0\right)$

**Return** $(\theta_t)_{t=1}^{T+1}, \bar{\theta} = \dfrac{1}{T}\displaystyle\sum_{t=1}^T \theta_t$

---

**Proof.** We start from observing that, according to the choices made in Ex. 2, for any $\theta \in \mathcal{S}$ and for any $w, u \in \mathbb{R}^d$, we have

$$f(w, \theta) = \frac{1}{2}\langle w, \theta^\dagger w\rangle + \iota_{\mathrm{Ran}(\theta)}(w) \quad f(\cdot, \theta)^*(u) = \frac{1}{2}\|\theta^{1/2}u\|_2^2 \quad \nabla f(\cdot, \theta)^*(u) = \theta u. \tag{128}$$

As a consequence, as observed in Prop. 2, for any $\theta \in \Theta$, we get that $w_{\theta,i+1} \in \mathrm{Dom} f(\cdot, \theta) = \mathrm{Ran}(\theta)$, for any $i \in \{0, \dots, n\}$. Moreover, according to the definition of $w_{\theta,1}$ in Alg. 2, we have

$$w_{\theta,1} = \nabla f(\cdot, \theta)^*(0) = 0. \tag{129}$$

We now show the closed form of $w_{\theta,i+1}$ for any $i \in \{1, \dots, n\}$. In such a case, denoting by $X_{1:i} \in \mathbb{R}^{i \times d}$ the matrix containing the first $i$ input vectors as rows, by definition of $w_{\theta,i+1}$ in Alg. 2, we can write

$$w_{\theta,i+1} = \nabla f(\cdot, \theta)^*\left(-\frac{1}{\lambda(i+1)} X_{1:i}^\top s_{\theta,i+1}\right) = -\frac{1}{\lambda(i+1)} \theta X_{1:i}^\top s_{\theta,i+1}. \tag{130}$$

For $i = 1$ the statement holds, as a matter of fact, since $w_{\theta,1} = 0$, exploiting Eq. (130) and introducing the subgradient $p_{\theta,1} = x_1 s'_{\theta,1} + \lambda\theta^\dagger w_{\theta,1} = x_1 s'_{\theta,1}$, we can write

$$w_{\theta,2} = -\frac{1}{2\lambda} \theta x_1 s'_{\theta,1} = w_{\theta,1} - \frac{1}{2\lambda} \theta p_{\theta,1}. \tag{131}$$

Now, we show that the statement holds also for $i \in \{2, \dots, n\}$. Since $X_{1:i}^\top s_{\theta,i+1} = X_{1:i-1}^\top s_{\theta,i} + x_i s'_{\theta,i}$, we can write the following

$$
\begin{aligned}
w_{\theta,i+1} &= -\frac{1}{\lambda(i+1)} \theta X_{1:i}^\top s_{\theta,i+1} = -\frac{1}{\lambda(i+1)}\left(\theta X_{1:i-1}^\top s_{\theta,i} + \theta x_i s'_{\theta,i}\right) \\
&= \frac{\lambda i}{\lambda(i+1)}\left(-\frac{1}{\lambda i} \theta X_{1:i-1}^\top s_{\theta,i}\right) - \frac{\theta x_i s'_{\theta,i}}{\lambda(i+1)} \\
&= \frac{\lambda(i+1)w_{\theta,i} - \theta x_i s'_{\theta,i} - \lambda w_{\theta,i}}{\lambda(i+1)} \\
&= w_{\theta,i} - \frac{1}{\lambda(i+1)}\left(\theta x_i s'_{\theta,i} + \lambda w_{\theta,i}\right) = w_{\theta,i} - \frac{1}{\lambda(i+1)} \theta p_{\theta,i},
\end{aligned} \tag{132}
$$

where, in the first and the fourth equality, we have exploited Eq. (130) and in the sixth equality we have exploited the form of the subgradient $p_{\theta,i} = x_i s'_{\theta,i} + \lambda\theta^\dagger w_{\theta,i}$ and the fact that $w_{\theta,i} \in \mathrm{Ran}(\theta)$. $\blacksquare$

We now proceed with the deduction of the meta-algorithm in Alg. 8.

**Lemma 45** (Derivation of the meta-algorithm in Alg. 8, feature map). *For any $t \in \{0, \ldots, T\}$, let $\theta_{t+1}$ be the update of the variable deriving from applying Alg. 3 to the data $(Z_t)_{t=1}^T$ in the setting outlined in Ex. 2. Let $\nabla'_{\theta_t}$ be the approximated meta-subgradient computed as described in Prop. 3 and used by the algorithm to compute $\theta_{t+1}$. Then, $\theta_{t+1} \in \mathcal{S}$. Specifically, we have $\theta_1 = \theta_0$ and, for any $t \in \{1, \ldots, T\}$,*

$$\theta_{t+1} = \text{proj}_{\mathcal{S}}\Big(-\frac{1}{\eta}\sum_{j=1}^{t} \nabla'_{\theta_j} + \theta_0\Big). \tag{133}$$

*Moreover, for any $t \in \{1, \ldots, T\}$,*

$$\nabla'_{\theta_t} = -\frac{1}{2\lambda n^2}\, X_t^{\top} s_{\theta_t,n+1} s_{\theta_t,n+1}^{\top} X_t, \tag{134}$$

*where $s_{\theta_t,n+1} \in \mathbb{R}^n$ is the output of Alg. 8 with feature map $\theta_t$ over the dataset $Z_t$ and, under Asm. 3,*

$$\big\|\nabla'_{\theta_t}\big\|_F^2 \le \frac{L^4\|C_t\|_{\infty}^2}{4\lambda^2}. \tag{135}$$

**Proof.** We start from observing that, according to the choices made in Ex. 2, according to Lemma 30 in App. B, for any $K \in \mathbb{S}^d$, $\theta \in \mathcal{S}$ and $u \in \mathbb{R}^d$, we have

$$
\begin{aligned}
F(\theta) &= \frac{1}{2}\|\theta - \theta_0\|_F^2 + \iota_{\mathcal{S}}(\theta) \\
F^*(K) &= \max_{\theta \in \mathcal{S}} \langle \theta, K \rangle - \frac{1}{2}\|\theta - \theta_0\|_F^2 \\
\nabla F^*(K) &= \underset{\theta \in \mathcal{S}}{\text{argmax}}\ \langle \theta, K \rangle - \frac{1}{2}\|\theta - \theta_0\|_F^2 = \underset{\theta \in \mathcal{S}}{\text{argmin}}\ \frac{1}{2}\|\theta - \theta_0\|_F^2 - \langle \theta, K \rangle \\
&= \underset{\theta \in \mathcal{S}}{\text{argmin}}\ \frac{1}{2}\|\theta - (K + \theta_0)\|_F^2 - \frac{1}{2}\|K\|_F^2 - \langle \theta_0, K \rangle \\
&= \text{proj}_{\mathcal{S}}(K + \theta_0) \\
f(\cdot, \theta)^*(u) &= \frac{1}{2}\|\theta^{1/2} u\|_2^2.
\end{aligned}
\tag{136}
$$

Consequently, according to the definition of $\theta_1$ in Alg. 3, we have

$$\theta_1 = \nabla F^*(0) = \theta_0. \tag{137}$$

The desired closed form of $\theta_{t+1}$ for any $t \in \{1, \ldots, T\}$ directly derives from the definition of $\theta_{t+1}$ in Alg. 3, according to which

$$\theta_{t+1} = \nabla F^*\Big(-\frac{1}{\eta}\sum_{j=1}^{t} \nabla'_{\theta_j}\Big) = \text{proj}_{\mathcal{S}}\Big(-\frac{1}{\eta}\sum_{j=1}^{t} \nabla'_{\theta_j} + \theta_0\Big). \tag{138}$$

We now specify the closed form of the approximated meta-subgradients, computed as described in Prop. 3 for Ex. 2. We start from observing that adding to the notation in Prop. 3 the further task index $t$, by strong duality (see Lemma 34), we can rewrite

$$\mathcal{L}_t(\theta) = \max_{s \in \mathbb{R}^n} \tilde{D}_{t,n+1}(s, \theta) \qquad \tilde{D}_{t,n+1}(s,\theta) = -\frac{1}{n}D_{t,n+1}(s,\theta) \tag{139}$$

where, according to Eq. (8), in the setting outlined in Ex. 2,

$$
\begin{aligned}
-D_{t,n+1}(s,\theta) &= -\sum_{i=1}^{n} \ell_{t,i}^*(s_i) - \lambda n f(\cdot, \theta)^*\Big(-\frac{1}{\lambda n}\sum_{i=1}^{n} x_{t,i} s_i\Big) \\
&= -\sum_{i=1}^{n} \ell_{t,i}^*(s_i) - \lambda n f(\cdot, \theta)^*\Big(-\frac{1}{\lambda n} X_t^{\top} s\Big) \\
&= -\sum_{i=1}^{n} \ell_{t,i}^*(s_i) - \frac{1}{2\lambda n} s^{\top} X_t \theta X_t^{\top} s.
\end{aligned}
\tag{140}
$$

Consequently, recalling that the output $s_{\theta_t,n+1}$ of the inner algorithm coincides with the last iterate of the corresponding dual inner iteration, according to Prop. 3, we have

$$\nabla_{\theta_t} = -\frac{1}{2\lambda n}\, X_t{}^\top s_{\theta_t,n+1} s_{\theta_t,n+1}^\top X_t \tag{141}$$

and, consequently,

$$\nabla'_{\theta_t} = \nabla_{\theta_t}/n \in \partial_{\epsilon_{\theta_t}/n}\mathcal{L}_t(\theta_t), \tag{142}$$

where $\epsilon_{\theta_t}$ is outlined in Prop. 3 and it must be specified to Ex. 2. In order to prove Eq. (135), we start from observing that $s_{\theta_t,n+1}$ is the vector in $\mathbb{R}^n$ having as component $i$ the subgradient $s'_{\theta_t,i} \in \partial \ell_{t,i}(\langle x_{t,i}, w_{\theta_t,i}\rangle)$. Hence, under Asm. 3, by Lemma 25 in App. B, any component of $s_{\theta_t,n+1}$ is absolutely bounded by $L$, and, consequently, $\|s_{\theta_t,n+1}\|_2 \leq L\sqrt{n}$. This allows us to get the desired bound by applying Holder's inequality (see Lemma 8 in App. B) to the matrices' scalar product as follows

$$\left\|\nabla'_{\theta_t}\right\|_F = \frac{1}{2\lambda n}\,\mathrm{Tr}\!\left(\frac{1}{n}\sum_{i=1}^n x_{t,i}x_{t,i}^\top s_{\theta_t,n+1}s_{\theta_t,n+1}^\top\right) \leq \frac{1}{2\lambda n}\left\|\frac{1}{n}\sum_{i=1}^n x_{t,i}x_{t,i}^\top\right\|_\infty \|s_{\theta_t,n+1}\|_2^2$$

$$\leq \frac{L^2}{2\lambda}\left\|\frac{1}{n}\sum_{i=1}^n x_{t,i}x_{t,i}^\top\right\|_\infty = \frac{L^2\|C_t\|_\infty}{2\lambda},$$

where in the last equality we have introduced the definition of $C_t$. ∎

We observe that the meta-algorithm we have retrieved in Alg. 8 is a slightly different version of that one proposed in [12], where we consider only an OWB statistical Meta-Learning framework. We refer to the discussion in App. A for more details about that work.

We also observe that for the setting in Ex. 2, our Meta-Learning method in Alg. 7 and Alg. 8 requires to compute the eigenvalue decomposition of a rank one perturbation of the current matrix. This can be performed using methods such as the ones in [39], which essentially scale quadratically w.r.t. the input dimension. As done in [8] for an OWB statistical Meta-Learning setting, a cheaper alternative here may be to use as meta-algorithm Frank-Wolfe, which requires to compute only the maximum eigenvalue. However, the better scaling property of this method comes at the price of a slower learning/convergence rate.

In the next section, we analyze the performance of our OWO Meta-Learning method applied to Ex. 2, in the non-statistical setting.

## H.2 Analysis of the method in the non-statistical setting for Ex. 2

In the next result we specify Thm. 4 to Ex. 2, that is, we provide a (regularized) cumulative error bound for the procedure deriving from combining Alg. 7 with Alg. 8.

**Corollary 6** (Cumulative error bound, feature map, long version). *Let Asm. 3 hold and consider the setting in Thm. 4 applied to Ex. 2. Then, introducing the empirical covariance matrix of the vectors $(\hat{w}_t)_{t=1}^T$*

$$\hat{B} = \frac{1}{T}\sum_{t=1}^T \hat{w}_t\hat{w}_t^\top, \tag{143}$$

*the following (regularized) cumulative error bound holds for any $\theta \in \mathcal{S}$ such that $\mathrm{Ran}(\hat{B}) \subseteq \mathrm{Ran}(\theta)$,*

$$\mathcal{E}_{\mathrm{meta}}^{\mathrm{reg}}\big((Z_t)_{t=1}^T\big) \leq nT\left(\frac{1}{T}\sum_{t=1}^T \mathcal{R}_{Z_t}(\hat{w}_t) + \frac{\lambda\,\mathrm{Tr}(\theta^\dagger \hat{B})}{2} + \frac{L^2\,\mathrm{Tr}(\hat{C}_{\theta_{1:T}}^{\mathrm{tot}})}{2\lambda n} + \frac{\eta\|\theta-\theta_0\|_F^2}{2T} + \frac{L^4\|C^{\mathrm{tot}}\|_{\infty,2}}{8\lambda^2\eta}\right)$$

*where we have defined the matrix*

$$\hat{C}_{\theta_{1:T}}^{\mathrm{tot}} = \frac{1}{T}\sum_{t=1}^T \theta_t\hat{C}_t. \tag{144}$$

*Hence, optimizing w.r.t. the hyper-parameters $\lambda$ and $\eta$, for*

$$\lambda = L\sqrt{\frac{1}{\mathrm{Tr}(\theta^\dagger\hat{B})}\left(\frac{\mathrm{Tr}(\hat{C}_{\theta_{1:T}}^{\mathrm{tot}})}{n} + \|\theta-\theta_0\|_F\sqrt{\frac{\|C^{\mathrm{tot}}\|_{\infty,2}}{T}}\right)} \qquad \eta = \frac{L^2\sqrt{T}\,\|C^{\mathrm{tot}}\|_{\infty,2}}{2\lambda\|\theta-\theta_0\|_F},$$

$$\tag{145}$$

*we get*

$$\mathcal{E}_{\text{meta}}^{\text{reg}}\big((Z_t)_{t=1}^T\big) \le nT\left(\frac{1}{T}\sum_{t=1}^T \mathcal{R}_{Z_t}(\hat{w}_t) + L\sqrt{\text{Tr}(\theta^\dagger \hat{B})\left(\frac{\text{Tr}(\hat{C}_{\theta_{1:T}}^{\text{tot}})}{n} + \|\theta - \theta_0\|_F\sqrt{\frac{\|C^{\text{tot}}\|_{\infty,2}}{T}}\right)}\right).$$

**Proof.** Specializing Thm. 4 to the quantities outlined in Ex. 2, exploiting the bound on the norm of the approximated meta-subgradients given in Eq. (135) (exploiting Asm. 3), and using the notation in Eq. (143) for any $\theta \in \mathcal{S}$ such that $\text{Ran}(\hat{B}) \subseteq \text{Ran}(\theta)$, we get

$$\mathcal{E}_{\text{meta}}^{\text{reg}}\big((Z_t)_{t=1}^T\big) \le nT\left(\frac{1}{T}\sum_{t=1}^T \mathcal{R}_{Z_t}(\hat{w}_t) + \frac{\lambda\text{Tr}(\theta^\dagger\hat{B})}{2} + \frac{1}{2\lambda nT}\sum_{t=1}^T\sum_{i=1}^n \frac{1}{i}\left\|\theta_t^{1/2}x_{t,i}s'_{\theta_t,i}\right\|_2^2\right.$$
$$\left. + \frac{\eta\|\theta - \theta_0\|_F^2}{2T} + \frac{L^4\|C^{\text{tot}}\|_{\infty,2}}{8\lambda^2\eta}\right).$$
(146)

The statement derives from the above inequality observing that, under Asm. 3 using the definition of $\hat{C}_{\theta_{1:T}}^{\text{tot}}$ in Eq. (144), we can write

$$\frac{1}{T}\sum_{t=1}^T\sum_{i=1}^n\frac{1}{i}\left\|\theta_t^{1/2}x_{t,i}s'_{t,i}\right\|_2^2 \le L^2\text{Tr}\Big(\frac{1}{T}\sum_{t=1}^T\theta_t\sum_{i=1}^n\frac{1}{i}x_{t,i}x_{t,i}^\top\Big) = L^2\text{Tr}(\hat{C}_{\theta_{1:T}}^{\text{tot}}).$$
(147)

■

Also in this case, in order to evaluate the quality of the bound above, we specify Thm. 36 to Ex. 2, that is, we provide a (regularized) cumulative error bound for the procedure deriving from running the within-task Alg. 7 with an appropriate feature map fixed in hindsight for any task.

**Corollary 46** (Cumulative error bound with fixed meta-parameter in hindsight, feature map). *Let Asm. 3 hold and consider the setting in Thm. 36 applied to Ex. 2. Then, according to the notation in Eq. (143), the following (regularized) cumulative error bound holds for any $\theta \in \mathcal{S}$ such that* $\text{Ran}(\hat{B}) \subseteq \text{Ran}(\theta)$

$$\sum_{t=1}^T \mathcal{E}_{\text{inner}}^{\text{reg}}(Z_t, \theta) \le nT\left(\frac{1}{T}\sum_{t=1}^T \mathcal{R}_{Z_t}(\hat{w}_t) + \frac{\lambda\text{Tr}(\theta^\dagger\hat{B})}{2} + \frac{L^2\text{Tr}(\theta\hat{C}^{\text{tot}})}{2\lambda n}\right).$$
(148)

*Hence, optimizing w.r.t. the hyper-parameter $\lambda$, for*

$$\lambda = L\sqrt{\frac{\text{Tr}(\theta\hat{C}^{\text{tot}})}{n\text{Tr}(\theta^\dagger\hat{B})}},$$
(149)

*we get*

$$\sum_{t=1}^T \mathcal{E}_{\text{inner}}^{\text{reg}}(Z_t, \theta) \le nT\left(\frac{1}{T}\sum_{t=1}^T \mathcal{R}_{Z_t}(\hat{w}_t) + L\sqrt{\frac{\text{Tr}(\theta^\dagger\hat{B})\,\text{Tr}(\theta\hat{C}^{\text{tot}})}{n}}\right).$$
(150)

**Proof.** Specializing Thm. 36 to the quantities outlined in Ex. 2, using the notation in Eq. (143), for any $\theta \in \mathcal{S}$ such that $\text{Ran}(\hat{B}) \subseteq \text{Ran}(\theta)$, we get

$$\sum_{t=1}^T \mathcal{E}_{\text{inner}}^{\text{reg}}(Z_t, \theta) \le nT\left(\frac{1}{T}\sum_{t=1}^T \mathcal{R}_{Z_t}(\hat{w}_t) + \frac{\lambda\text{Tr}(\theta^\dagger\hat{B})}{2} + \frac{1}{2\lambda nT}\sum_{t=1}^T\sum_{i=1}^n\frac{1}{i}\left\|\theta^{1/2}x_{t,i}s'_{t,i}\right\|_2^2\right).$$

The statement derives from the above inequality observing that, under Asm. 3 using the definition of the matrix $\hat{C}^{\text{tot}}$, we can write

$$\frac{1}{T}\sum_{t=1}^T\sum_{i=1}^n\frac{1}{i}\left\|\theta^{1/2}x_{t,i}s'_{t,i}\right\|_2^2 \le L^2\text{Tr}\Big(\theta\frac{1}{T}\sum_{t=1}^T\sum_{i=1}^n\frac{1}{i}x_{t,i}x_{t,i}^\top\Big) = L^2\text{Tr}(\theta\hat{C}^{\text{tot}}).$$
(151)

■

We postpone to App. H.4 a discussion about the results we have reported above. In the next section, we analyze the performance of our OWO Meta-Learning method applied to Ex. 2, in the statistical setting.

### H.3 Analysis of the method in the statistical setting for Ex. 2

In the result below we specify Thm. 5 to Ex. 2, that is, we provide a (regularized) transfer risk bound for the average $\bar{w}_{\bar{\theta}}$ of the estimators returned by the combination of Alg. 7 with Alg. 8.

**Corollary 7** (Transfer risk bound, bias, long version). *Let Asm. 3 hold and consider the statistical setting in Thm. 5 applied to Ex. 2. Then, introducing the exact covariance matrix of the vectors $w_{\mu}$*

$$B_{\rho} = \mathbb{E}_{\mu \sim \rho} w_{\mu} w_{\mu}^{\top}, \tag{152}$$

*the following (regularized) transfer risk bound holds for any $\theta \in \mathcal{S}$ such that $\mathrm{Ran}(B_{\rho}) \subseteq \mathrm{Ran}(\theta)$*

$$\mathbb{E}\,\mathcal{E}_{\mathrm{stat}}^{\mathrm{reg}}(\bar{w}_{\bar{\theta}}) \leq \ \mathcal{E}_{\rho} + \frac{\lambda \mathrm{Tr}(\theta^{\dagger} B_{\rho})}{2} + \frac{L^2(\log(n)+1)\mathrm{Tr}\big(\mathbb{E}\,\bar{\theta} C_{\rho}\big)}{\lambda n} \\ + \frac{\eta \|\theta - \theta_0\|_F^2}{2T} + \frac{L^4 \mathbb{E}\,\|C^{\mathrm{tot}}\|_{\infty,2}}{8\lambda^2 \eta}. \tag{153}$$

*Hence, optimizing w.r.t. the hyper-parameters $\lambda$ and $\eta$, for*

$$\lambda = L\sqrt{\frac{1}{\mathrm{Tr}(\theta^{\dagger} B_{\rho})}\left(\frac{2(\log(n)+1)\mathrm{Tr}\big(\mathbb{E}\,\bar{\theta} C_{\rho}\big)}{n} + \|\theta - \theta_0\|_F \sqrt{\frac{\mathbb{E}\,\|C^{\mathrm{tot}}\|_{\infty,2}}{T}}\right)} \tag{154}$$

$$\eta = \frac{L^2 \sqrt{T\,\mathbb{E}\,\|C^{\mathrm{tot}}\|_{\infty,2}}}{2\lambda\|\theta - \theta_0\|_F}, \tag{155}$$

*we get*

$$\mathbb{E}\,\mathcal{E}_{\mathrm{stat}}^{\mathrm{reg}}(\bar{w}_{\bar{\theta}}) \leq \ \mathcal{E}_{\rho} + L\sqrt{\mathrm{Tr}(\theta^{\dagger} B_{\rho})\left(\frac{2(\log(n)+1)\,\mathrm{Tr}\big(\mathbb{E}\,\bar{\theta} C_{\rho}\big)}{n} + \|\theta - \theta_0\|_F \sqrt{\frac{\mathbb{E}\,\|C^{\mathrm{tot}}\|_{\infty,2}}{T}}\right)}.$$

**Proof.** Specializing Thm. 5 to the quantities outlined in Ex. 2, exploiting the bound on the norm of the approximated meta-subgradients given in Eq. (135) (exploiting Asm. 3) and using the notation in Eq. (152), for any $\theta \in \mathcal{S}$ such that $\mathrm{Ran}(B_{\rho}) \subseteq \mathrm{Ran}(\theta)$, we get the following

$$\mathbb{E}\,\mathcal{E}_{\mathrm{stat}}^{\mathrm{reg}}(\bar{w}_{\bar{\theta}}) \leq \ \mathcal{E}_{\rho} + \frac{\lambda \mathrm{Tr}(\theta^{\dagger} B_{\rho})}{2} + \frac{1}{2\lambda nT}\,\mathbb{E}\,\sum_{t=1}^{T}\sum_{i=1}^{n}\frac{1}{i}\,\big\|\theta_t^{1/2} x_{t,i} s'_{\theta_t,i}\big\|_2^2 \\ + \frac{\eta \|\theta - \theta_0\|_F^2}{2T} + \frac{L^4 \mathbb{E}\,\|C^{\mathrm{tot}}\|_{\infty,2}}{8\lambda^2 \eta} \\ + \frac{1}{2\lambda n}\,\mathbb{E}\,\mathbb{E}_{\mu \sim \rho}\,\mathbb{E}_{Z \sim \mu^n}\sum_{i=1}^{n}\frac{1}{i}\,\big\|\bar{\theta}^{1/2} x_i s'_{\bar{\theta},i}\big\|_2^2. \tag{156}$$

The desired statement derives from the above inequality and from observing that, thanks to Asm. 3, the i.i.d. sampling of the data and the fact that $\theta_t$ depends only on the previous datasets $(Z_j)_{j=1}^{t-1}$, using the inequality $\sum_{i=1}^{n} 1/i \leq \log(n) + 1$ and the definition of $C_{\rho}$, we have

$$\mathbb{E}\,\frac{1}{T}\sum_{t=1}^{T}\sum_{i=1}^{n}\frac{1}{i}\,\big\|\theta_t^{1/2} x_{t,i} s'_{\theta_t,i}\big\|_2^2 \leq L^2 \mathbb{E}\,\mathrm{Tr}\Big(\frac{1}{T}\sum_{t=1}^{T}\theta_t \sum_{i=1}^{n}\frac{1}{i}\,x_{t,i} x_{t,i}^{\top}\Big) \\ = L^2(\log(n)+1)\mathrm{Tr}\big(\mathbb{E}\,\bar{\theta} C_{\rho}\big) \tag{157}$$

$$\mathbb{E}\,\mathbb{E}_{\mu \sim \rho}\,\mathbb{E}_{Z \sim \mu^n}\sum_{i=1}^{n}\frac{1}{i}\,\big\|\bar{\theta} x_i s'_{\bar{\theta},i}\big\|_2^2 \leq L^2 \mathbb{E}\,\mathbb{E}_{\mu \sim \rho}\,\mathbb{E}_{Z \sim \mu^n}\,\mathrm{Tr}\Big(\bar{\theta}\sum_{i=1}^{n}\frac{1}{i}\,x_i x_i^{\top}\Big) \\ = L^2(\log(n)+1)\mathrm{Tr}\big(\mathbb{E}\,\bar{\theta} C_{\rho}\big). \tag{158}$$

∎

In order to evaluate the quality of the bound above, we specify Thm. 37 to Ex. 2, that is, we provide a (regularized) transfer risk bound for $\bar{w}_\theta$, the average of the iterations returned by running the within-task Alg. 7 with an appropriate feature map $\theta$ fixed in hindsight for any task.

**Corollary 47** (Transfer risk bound with fixed meta-parameter in hindsight, feature map). *Let Asm. 3 hold and consider the statistical setting in Thm. 37 applied to Ex. 2. Then, according to the notation in Eq. (152), the following (regularized) transfer risk bound holds for any $\theta \in \mathcal{S}$ such that* $\mathrm{Ran}(B_\rho) \subseteq \mathrm{Ran}(\theta)$

$$\mathcal{E}_{\mathrm{stat}}^{\mathrm{reg}}(\bar{w}_{\bar\theta}) \leq \mathcal{E}_\rho + \frac{\lambda \mathrm{Tr}(\theta^\dagger B_\rho)}{2} + \frac{L^2(\log(n)+1)\mathrm{Tr}(\theta C_\rho)}{2\lambda n}. \tag{159}$$

*Hence, optimizing w.r.t. the hyper-parameter $\lambda$, for*

$$\lambda = L \sqrt{\frac{(\log(n)+1)\mathrm{Tr}(\theta C_\rho)}{n \mathrm{Tr}(\theta^\dagger B_\rho)}}, \tag{160}$$

*we get*

$$\mathcal{E}_{\mathrm{stat}}^{\mathrm{reg}}(\bar{w}_{\bar\theta}) \leq \mathcal{E}_\rho + L \sqrt{\frac{(\log(n)+1)\mathrm{Tr}(\theta^\dagger B_\rho)\mathrm{Tr}(\theta C_\rho)}{n}}.$$

**Proof.** Specializing Thm. 37 to the quantities outlined in Ex. 2, using the notation in Eq. (152), for any $\theta \in \mathcal{S}$ such that $\mathrm{Ran}(B_\rho) \subseteq \mathrm{Ran}(\theta)$, we get the following

$$\mathcal{E}_{\mathrm{stat}}^{\mathrm{reg}}(\bar{w}_{\bar\theta}) \leq \mathcal{E}_\rho + \frac{\lambda \mathrm{Tr}(\theta^\dagger B_\rho)}{2} + \frac{1}{2\lambda n} \, \mathbb{E}_{\mu \sim \rho} \, \mathbb{E}_{Z \sim \mu^n} \sum_{i=1}^n \frac{1}{i} \, \left\| \theta^{1/2} x_i s_{\theta,i}' \right\|_2^2.$$

The desired statement derives from the above inequality and from observing that, under Asm. 3, exploiting the i.i.d. sampling of the data and the inequality $\sum_{i=1}^n 1/i \leq \log(n)+1$, introducing the definition of the matrix $C_\rho$, we can write

$$\mathbb{E}_{\mu \sim \rho} \, \mathbb{E}_{z_n \sim \mu^n} \sum_{i=1}^n \frac{1}{i} \, \left\| \theta^{1/2} x_i s_{\theta,i}' \right\|_2^2 \leq L^2 \mathbb{E}_{\mu \sim \rho} \, \mathbb{E}_{Z \sim \mu^n} \, \mathrm{Tr}\left( \theta \sum_{i=1}^n \frac{1}{i} \, x_i x_i^\top \right)$$
$$\leq L^2(\log(n)+1)\mathrm{Tr}(\theta C_\rho). \tag{161}$$

$\blacksquare$

Also in this case, the comments to the bounds above are postponed in the following App. H.4.

## H.4   Discussion of the results for Ex. 2

We start from discussing the results in Cor. 46 and Cor. 47, where the feature map used by the inner algorithm is fixed in hindsight for any task.

### H.4.1   Advantage of selecting the right feature map

We first comment the bounds in the statistical setting in Cor. 47 . In this case, proceeding in the same way as described for Ex. 1, we should define the best algorithm in our class (*oracle*) the algorithm associated to the feature map minimizing the bound in Cor. 47. However, in our case, to simplify the analysis we consider as the oracle the algorithm associated to the feature map $\theta_\rho$ minimizing only a part of the above bound which is available is closed form. Specifically, appealing to the infimal formulation of the MTL trace norm regularizer in [2, Eq. (13)], we minimize only the term $\mathrm{Tr}(\theta^\dagger B_\rho)$ over the subset of the feature maps $\{\theta \in \mathcal{S} : \mathrm{Ran}(B_\rho) \subseteq \mathrm{Ran}(\theta)\}$ for which our bound holds:

$$\min_{\theta \in \mathcal{S}: \mathrm{Ran}(B_\rho) \subseteq \mathrm{Ran}(\theta)} \mathrm{Tr}(\theta^\dagger B_\rho) = \mathrm{Tr}(B_\rho^{1/2})^2 = \|W_\rho\|_{\mathrm{Tr}}^2, \tag{162}$$

where $W_\rho$ is a square root of $B_\rho$. We consider as the optimal feature map the corresponding minimizer

$$\theta_\rho = \operatorname*{argmin}_{\theta \in \mathcal{S}: \mathrm{Ran}(B_\rho) \subseteq \mathrm{Ran}(\theta)} \mathrm{Tr}(\theta^\dagger B_\rho) = \frac{W_\rho}{\mathrm{Tr}(W_\rho)}. \tag{163}$$

Similarly to what observed in App. G.4 for the setting in Ex. 1, we will consider this feature map as benchmark in order to evaluate the performance of our Meta-Learning procedure. With such a choice of feature map $\theta_\rho$, the bound in Cor. 47 (up to logarithmic factors) becomes proportional to

$$\sqrt{\frac{\text{Tr}(\theta_\rho{}^\dagger B_\rho)\,\text{Tr}(\theta_\rho C_\rho)}{n}} \leq \|W_\rho\|_{\text{Tr}} \sqrt{\frac{\|C_\rho\|_\infty}{n}}, \tag{164}$$

where, in the inequality above, we have applied Holder's inequality (see Lemma 8 in App. B) to the matrices' scalar product and we have exploited the fact $\text{Tr}\,(\theta_\rho) = 1$.

On the other hand, solving the tasks independently (ITL), in this case, corresponds to apply Alg. 7 with the feature map $\theta_{\text{ITL}} = I/d$ for any task. Substituting this value, the bound above becomes proportional to

$$\|W_\rho\|_F \sqrt{\frac{\text{Tr}(C_\rho)}{n}}. \tag{165}$$

Comparing the bounds in Eq. (164) and Eq. (165), we can conclude that there is an advantage in using the optimal feature map $\theta_\rho$ w.r.t. solving each task independently, when the tasks are *similar* in the sense that $\|C_\rho\|_\infty \ll \text{Tr}\,(C_\rho)$ (when the inputs are high-dimensional for instance) and when $\|W_\rho\|_{\text{Tr}}$ is comparable to $\|W_\rho\|_F$ (i.e. when the matrix $W_\rho$ is low-rank, meaning that the tasks' target vectors are expected to lie in a low-dimensional subspace, the range of the optimal feature map). This is inline with previous literature, such as [12, 26, 25].

Regarding the non-statistical setting, in order to comment the cumulative error bound in Cor. 46, one can proceed as above introducing the corresponding sub-optimal algorithm in the class associated to the corresponding sub-optimal feature map $\hat{\theta}$. The associated bound, in this case, becomes proportional to

$$\|\hat{W}\|_{\text{Tr}} \sqrt{\frac{\|\hat{C}^{\text{tot}}\|_\infty}{n}}, \tag{166}$$

where $\hat{W}$ is a square root of $\hat{B}$. Comparing this last bound to the corresponding bound for ITL

$$\|\hat{W}\|_F \sqrt{\frac{\text{Tr}(\hat{C}^{\text{tot}})}{n}}, \tag{167}$$

we see that there is an advantage in using the optimal feature map $\hat{\theta}$ w.r.t. solving each task independently, when $\|\hat{C}^{\text{tot}}\|_\infty \ll \text{Tr}(\hat{C}^{\text{tot}})$ and $\hat{W}$ is low-rank ($\|\hat{W}\|_{\text{Tr}}$ is comparable to $\|\hat{W}\|_F$). The first condition on the weighted input covariance matrix $\hat{C}^{\text{tot}} = \frac{1}{T}\sum_{t=1}^{T}\sum_{i=1}^{n}\frac{1}{i}\,x_{t,i}x_{t,i}^\top$ is less clear to interpret than the more natural one $\|C^{\text{tot}}\|_\infty \ll \text{Tr}(C^{\text{tot}})$ with the standard empirical input covariance matrix $C^{\text{tot}} = \frac{1}{T}\sum_{t=1}^{T}\frac{1}{n}\sum_{i=1}^{n}x_{t,i}x_{t,i}^\top$. However, in certain data configurations these two input covariance matrices, may still be closed one to each other. We think that this issue is avoidable by choosing the inner step size in different way and we will address it in future work.

We now can make the following observations about the bounds we have obtained in Cor. 6 and Cor. 7 for our Meta-Learning procedure.

### H.4.2 Feature Map resulting from our Meta-Learning method

In order to analyze the effectiveness of our Meta-Learning method, we investigate whether it mimics the performance of the best algorithm in the class, when the number of training tasks is sufficiently large w.r.t. the number of within-task points. In such a case, the term $T^{-1/4}$ is negligible and, applying Holder's inequality and exploiting the fact that, by construction, $\text{Tr}(\bar{\theta}) \leq 1$, as described above, the bound in Cor. 7 (up to logarithmic factors) can be upper bounded by

$$\sqrt{\frac{\text{Tr}(\theta^\dagger B_\rho)\text{Tr}(C_\rho)}{n}}, \tag{168}$$

where $\theta \in \mathcal{S}$ is the fixed feature map in the statement, defining the choice of the hyper-parameters for our method. In particular, choosing $\theta = \theta_\rho$ in Eq. (163), the quantity above in Eq. (168) can be upper bounded by the bound in Eq. (164) for the best algorithm in the class. As a consequence, when

the tasks are similar as explained above, our methods can provide a significant advantage w.r.t. ITL in the statistical setting. We conclude observing that the cumulative error bound in Cor. 6 for the non-statistical setting is less clear to interpret because of the presence of the modified version of the covariance matrix $\hat{C}^{\text{tot}}_{\theta_{1:T}}$. Future work may be devoted to investigate this point, which could be either an artifact of our analysis or due to some intrinsic characteristics of the feature learning problem we are considering.

# I   Experimental details

In this section, we start from describing in App. I.1 how we tuned the hyper-parameters for our OWO Meta-Learning method in the statistical setting. After that, in App. I.2 we give some closed form expressions that we used for the implementation.

## I.1   Hyper-parameters tuning for our statistical OWO Meta-Learning method

Denote by $\bar{\theta}_{T,\lambda,\eta}$ the average of the meta-parameters computed with $T$ iterations (hence $T$ datasets and tasks) of our meta-algorithm with hyper-parameters $\lambda$ and $\eta$. In all the experiments, we obtained this meta-parameter by learning it on a collection of $T_{\text{tr}}$ *training* datasets (tasks), each comprising a dataset $Z_{\text{tr}}$ of $n = n_{\text{tr}}$ input-output pairs $z = (x, y) \in \mathcal{Z} = \mathcal{X} \times \mathcal{Y}$. We performed this meta-training for different values of $\lambda \in \{\lambda_1, \ldots, \lambda_p\}$ and $\eta \in \{\eta_1, \ldots, \eta_r\}$ and we selected the best meta-parameter based on the prediction error measured on a separate set of $T_{\text{va}}$ *validation* datasets (tasks). Once such optimal $\lambda$ and $\eta$ values were selected, we reported the error of the corresponding estimator on a set of $T_{\text{te}}$ *test* datasets (tasks).

Note that the tasks in the test and validation sets were all provided with a training inner dataset $Z_{\text{tr}}$ of $n_{\text{tr}}$ points and a test inner dataset $Z_{\text{te}}$ of $n_{\text{te}}$ points, both sampled from the same distribution. Indeed, in order to evaluate the performance of a meta-parameter $\theta$, we needed first to train the corresponding algorithm on the training dataset $Z_{\text{tr}}$, and then, to test the performance of the resulting vector on the test set $Z_{\text{te}}$.

In addition to this, since we considered the online setting, the training datasets arrived one at the time, therefore model selection was performed *online*: the system kept track of all candidate values $\bar{\theta}_{T_{\text{tr}},\lambda_j,\eta_k}$, $j \in \{1, \ldots, p\}$, $k \in \{1, \ldots, r\}$, and, whenever a new training task was presented, these meta-parameters were all updated by incorporating the corresponding new observations. The best meta-parameter $\theta$ was then returned at each iteration, based on its performance on the validation set, as explained before. The previous procedure describes how to tune simultaneously both $\lambda$ and $\eta$. When the meta-parameter $\theta$ we used was fixed a priori (e.g. in ITL), we just needed to tune the hyper-parameter $\lambda$; in such a case the procedure was analogous to that one described above.

Specifically, in the experiments reported in the main body, we applied the validation procedure above as described in the following.

**Synthetic data.** We considered 14 candidates values for both $\lambda$ and $\eta$ in the range $[10^{-5}, 10^5]$ with logarithmic spacing and we evaluated the performance of the estimated feature maps by using $T = T_{\text{tr}} = 3000$, $T_{\text{va}} = 100$, $T_{\text{te}} = 500$ of the available tasks for meta-training, meta-validation and meta-testing, respectively. In order to train and to test the inner algorithm, we splitted each within-task dataset into $n = n_{\text{tr}} = 50\% \, n_{\text{tot}}$ for training and $n_{\text{te}} = 50\% \, n_{\text{tot}}$ for test.

**Movielens-100k dataset.** In this case, we removed all movies with less than 20 users' ratings. We considered 14 candidates values for both $\lambda$ and $\eta$ in the range $[10^{-5}, 10^5]$ with logarithmic spacing and we evaluated the performance of the estimated feature maps by splitting the tasks into $T = T_{\text{tr}} = 700$, $T_{\text{va}} = 100$, $T_{\text{te}} = 139$ tasks used for meta-training, meta-validation and meta-testing, respectively. In order to train and to test the inner algorithm, we splitted each within-task dataset into $n = n_{\text{tr}} = 75\% \, n_{\text{tot}}$ for training and $n_{\text{te}} = 25\% \, n_{\text{tot}}$ for test.

**Mini-Wiki dataset.** We considered 14 candidates values for both $\lambda$ and $\eta$ in the range $[10^{-5}, 10^5]$ with logarithmic spacing and we evaluated the performance of the estimated feature maps by splitting the tasks into $T = T_{\text{tr}} = 500$, $T_{\text{va}} = 100$, $T_{\text{te}} = 213$ tasks used for meta-training, meta-validation and meta-testing, respectively. In order to train and to test the inner algorithm, we splitted each within-task dataset into $n = n_{\text{tr}} = 75\% \, n_{\text{tot}}$ for training and $n_{\text{te}} = 25\% \, n_{\text{tot}}$ for test.

**Algorithm 9** Within-task algorithm for Ex. 2, multi-class setting, $\ell_i = \ell_{z_i}$ with $\ell_{z_i}$ in Eq. (170)

---

**Input** $\lambda > 0, \theta \in \mathcal{S}, Z = (z_i)_{i=1}^n$

**Initialization** $S_{\theta,1} = (), W_{\theta,1} = 0$

**For** $i = 1$ to $n$

  Receive the datapoint $z_i = (x_i, y_i)$

  Compute $S'_{\theta,i} \in \partial \ell_i(W_{\theta,i}) \subseteq \mathbb{R}^{d \times M}$

  Define $(S_{\theta,i+1})_i = S'_{\theta,i}, \gamma_i = \lambda(i+1)$

  Define $P_{\theta,i} = S'_{\theta,i} + \lambda \theta^\dagger W_{\theta,i} \in \mathbb{R}^{d \times M}$

  Update $W_{\theta,i+1} = W_{\theta,i} - 1/\gamma_i \, \theta P_{\theta,i}$

**Return** $(W_{\theta,i})_{i=1}^{n+1}, \bar{W}_\theta = \dfrac{1}{n} \sum_{i=1}^n W_{\theta,i}, \, S_{\theta,n+1}$

---

**Algorithm 10** Meta-algorithm for Ex. 2, multi-class setting

---

**Input** $\eta > 0, (Z_t)_{t=1}^T, \theta_0 \in \mathcal{S}$

**Initialization** $\theta_1 = \theta_0, P_1 = 0 \in \mathbb{S}^d$

**For** $t = 1$ to $T$

  Receive incrementally the dataset $Z_t$

  Run Alg. 7 with $\theta_t$ over $Z_t$

  Compute $(S'_{\theta_t,i})_{i=1}^n$

  Define $\nabla'_{\theta_t} = -\dfrac{Q_t Q_t^\top}{2\lambda n^2}$    $Q_t = \sum_{i=1}^n S'_{\theta_t,i}$

  Update $P_{t+1} = P_t + \nabla'_{\theta_t}$

  Update $\theta_{t+1} = \mathrm{proj}_\mathcal{S}(-P_{t+1}/\eta + \theta_0)$

**Return** $(\theta_t)_{t=1}^{T+1}, \bar{\theta} = \dfrac{1}{T} \sum_{t=1}^T \theta_t$

---

**Jester-1 dataset.** In this case, we randomly subsampled the $24983$ jokes to end up with $5700$ total number of tasks. We considered $14$ candidates values for both $\lambda$ and $\eta$ in the range $[10^{-5}, 10^5]$ with logarithmic spacing and we evaluated the performance of the estimated feature maps by splitting the tasks into $T = T_{\mathrm{tr}} = 5000, T_{\mathrm{va}} = 200, T_{\mathrm{te}} = 500$ tasks used for meta-training, meta-validation and meta-testing, respectively. In order to train and to test the inner algorithm, we splitted each within-task dataset into $n = n_{\mathrm{tr}} = 75\% \, n_{\mathrm{tot}}$ for training and $n_{\mathrm{te}} = 25\% \, n_{\mathrm{tot}}$ for test.

All the experiments were conducted on an Intel Xeon E5-2697 V3 2.60Ghz CPU with 32GB RAM.

## I.2 Closed forms for the implementation

At last, we report the closed forms we used in our experiments.

**Absolute loss for regression.** Let $\mathcal{Y} \subseteq \mathbb{R}$. For any $\hat{y}, y \in \mathcal{Y}$, let $\ell(\hat{y}, y) = |\hat{y} - y|$ and denote $\ell_y(\cdot) = \ell(\cdot, y)$. Then, we have

$$\partial \ell_y(\hat{y}) = \begin{cases} \{1\} & \text{if } \hat{y} - y > 0 \\ \{-1\} & \text{if } \hat{y} - y < 0 \\ [-1,1] & \text{if } \hat{y} - y = 0. \end{cases} \tag{169}$$

**Hinge loss for multi-class classification.** Let $\mathcal{X} \subseteq \mathbb{R}^d$ and $\mathcal{Y} = \{1, \dots, M\}$, where $M$ is the number of classes. We measure the error of the predictors' matrix $W \in \mathbb{R}^{d \times M}$ over a datapoint $z = (x, y) \in \mathcal{X} \times \mathcal{Y}$ by the loss function

$$\ell_z(W) = \max_{m \in \{1, \dots M\}} \mathbf{1}_{m \neq y} + \langle W(:, m), x \rangle - \langle W(:, y), x \rangle, \tag{170}$$

where, for any $m \in \{1, \dots, M\}$, $W(:, m)$ denotes the $m$-th column of $W$ and

$$\mathbf{1}_{m \neq y} = \begin{cases} 1 & \text{if } m \neq y \\ 0 & \text{if } m = y. \end{cases} \tag{171}$$

Introducing the class-index

$$\hat{m} = \underset{m \in \{1, \dots M\}}{\mathrm{argmax}} \, \mathbf{1}_{m \neq y} + \langle W(:, m), x \rangle - \langle W(:, y), x \rangle, \tag{172}$$

we compute a subgradient $S' \in \partial \ell_z(W) \subseteq \mathbb{R}^{d \times M}$ as the matrix with $m$-th column given by

$$S'(:, m) = \begin{cases} x & \text{if } m = \hat{m} \\ -x & \text{if } m = y \\ 0 & \text{otherwise.} \end{cases} \tag{173}$$

As described in the main body, in the Mini-Wiki dataset experiment, we considered the setting outlined in Ex. 2 with the multi-class hinge loss above. In such a case, our OWO Meta-Learning method is reported in Alg. 9 – Alg. 10 and it coincides with a matrix-variant of Alg. 7 – Alg. 8.