[Reviews · NeurIPS 2019]

Reviewer 1



This work proposes algorithms for the online-within-online meta-learning setting as oppposed to the more prevalent statistical setting. In this particular meta-learning setting tasks arrive sequentially manner (outer loop) and then the learning per task itself happens in an online fashion. The aim is to have low average regret over tasks. The inner loop optimization is done via Online Mirror Descent (OMD). The inner algorithm design is carefully chosen to provide good approximations of (sub)-gradients of the outer meta objective. Specifically two nested online primal-dual online algorithms are utilized. This algorithmic framework is then extended from the adversarial to the statistical setting by two nested online-to-batch conversions. Experiments on synthetic data and on the movielens-100k dataset which contains the rationgs of different users to different movies are presented where the proposed algorithms perform significantly better as a function of number of training tasks than treating the tasks independently (ITL). In the movielens-100k dataset each of the 943 users' ratings are considered a separate task. Comments: While I will leave the judgement of theoretical novelty to more qualified reviewers I have a number of more practical questions: - Does this framework extend to meta-reinforcement learning settings trivially? (the online feedback is episodic sparse reward) or does that require fundamental change in viewpoint? - Minor nitpick: While the experiments on movielens dataset is a good one and I understand that scope of the paper is to study OWO setting theoretically, are there fundamental challenges to scaling up to say supervised datasets in rich observation spaces (meta-datasets of images as proposed in Meta-Dataset: A Dataset of Datasets for Learning to Learn from Few Examples by Zhu et al. recently)? - Minor nitpick: The relegation of related work to supplementary is a bit odd. It actually served to better situate the paper in light of related work and especially with respect to the batch statistical setting papers and other online meta-learning settings. Update: Thanks to the authors comments on extensions to reinforcement learning. I have maintained my good scores.

Reviewer 2



In this paper, a primal-dual online learning framework is proposed to deal with the online within online meta-learning problem. This setting is important in practice since it can be used to model the life-long learning scenario naturally. Comparing to recent advances on this topic [1][2][3], this paper proposes an alternative primal-dual view, providing a more general algorithmic framework and more refined theoretical results. The contribution is somewhat significant to this point for providing novel tools for future researches. On the other hand, as the primal-dual approach is classical for online learning, I expected deeper insights for thinking meta-learning under this perspective instead of a direct application of the theoretical tools. The discussion towards this direction is limited unfortunately. It seems that the proposed approach is more a theoretically guaranteed solver for classical objectives instead of a novel view of the original problem. While overall, I still think the paper proposes a good alternative to previous works thus I lean towards acceptance. Further comments: In the current version, the discussions of the related work are put in the supplementary materials. I strongly suggest putting them back in the main paper, especially for the overview of the other works under the online-within-online scenario for better comparison. [1] G. Denevi, C. Ciliberto, R. Grazzi, and M. Pontil, Learning-to-learn stochastic gradient descent with biased regularization. ICML 2019. [2] C. Finn, A. Rajeswaran, S. Kakade, and S. Levine. Online meta-learning. ICML 2019. [3] M. Khodak, M.-F. Balcan, and A. Talwalkar. Provable guarantees for gradient-based meta-learning. ICML 2019. ------ after rebuttal The author response does not change my evaluations very much. The main contribution lies in proposing the theoretical analysis of the online-within-online meta-learning setting from the online primal-dual framework, which is a good alternative comparing to related work.

Reviewer 3



In terms of clarity, this submission is fairly good at writing since I find it's not hard to follow this submission, the notations are clear. But I'm not quite sure about the significance of this submission since I only literally checked proof in it and is not very familiar with learning theory literature.

[Author Response · NeurIPS 2019]

We thank the reviewers for their valuable comments. We reply to each outstanding point below.

**REVIEWER 1. R.** *Extension to meta-reinforcement learning.* **A.** One starting point in this direction would be to consider the simplified setting of contextual bandits with linear reward functions. However, even in this case the extension of the proposed method does not seem straightforward and it requires further investigation. **R.** *...scaling up to...rich observation spaces?* **A.** The scaling properties of our method depend on the specific setting considered. For instance, for the bias setting, our method is based on the combination of Alg. 5 & Alg. 6 and it scales linearly with the dimension of the input space. We thus expect that in this setting the method will be appropriate also for datasets in more rich observation spaces. In the feature map setting, our method in Alg. 7 & Alg. 8 requires to compute the eigenvalue decomposition of a rank one perturbation of the current matrix. This can be performed using methods such as [*On the efficient update of the singular value decomposition*, Stange, 2008], which essentially scale quadratically with respect to the input dimensionality. An interesting alternative here may be to use Frank-Wolfe as meta-algorithm, which requires to compute only the maximum eigenvalue. However the better scaling property of this method comes at the price of a slower learning/convergence rate. **R.** *Related work in supplementary.* **A.** We agree, we will move the discussion on previous work to the main body. Thanks also for the additional reference, which we will add to the paper.

**REVIEWER 2. R.** *I expected more meta-learning insights. It seems that the proposed approach is more a theoretically guaranteed solver . . .* **A.** As described in the paper, the proposed method takes inspiration from multitask learning (MTL) and, as such, it allows to translate key MTL insights to meta-learning. In the paper, we particularly stress this aspect in Sec. 6, when we specialize our analysis to the settings of the bias and the feature map. In such cases, we provide an in depth interpretation of the bounds from the meta-learning perspective. We will highlight more the novelty of the proposed method from the meta-learning point of view also in the rest of the paper. **R.** *Related work in supplementary.* **A.** We agree, we will discuss the most related work in the main body. **R.** *. . . how the current theoretical framework helps practical algorithm design.* **A.** The proposed framework provides us with a flexible meta-learning scheme, which is able to cover various kinds of tasks' relatedness and a wide family of inner/meta algorithms, by choosing in an appropriate way the complexity terms ($f$ and $F$) and the aggressiveness of the updates. For instance, the feature learning setting can be immediately extended to other structured sparsity frameworks, by considering alternative choices for the set $\Theta$, different from the matricial simplex. Furher examples in our framework are e.g. online Gradient Descent, Matrix Exponentiated, $p$-norm and Follow-The-Regularized-Leader. We will discuss this in the paper.

Figure 1: (Left) Mini-Wiki Dataset. (Right) Jester1 Dataset.

**REVIEWER 3. R.** *. . . the proposed error bound seems novel although is related to previous work. I'm not quite sure about the significance of this submission.* **A.** We care to point out that the framework proposed in this work offers key improvements with respect to previous work in the meta-learning literature. First, differently from recent related work (see references [9,14] listed in the paper), the online-within-online method we propose can be adapted to a wide class of algorithms. Second, our analysis is innovative in that it allows us to provide guarantees for our method in the adversarial setting under very standard assumptions, by leveraging the delicate interplay between the within-task problem and the outer-task problem. On the contrary, other papers, such as [14,18] require to introduce stronger assumptions like growth conditions for the loss function. Third, note that when we apply our analysis to the statistical setting, we provide guarantees for the average of the estimators returned by our method, while in other papers [1,18] the authors provides guarantees for one estimator sampled from the pool, which requires the memorization of all the estimators and adds randomness to the process. **R.** *The authors should present more experiments to validate the proposed method.* **A.** We further validated the proposed method on two additional experimental settings satisfying all the assumptions of our theory. Specifically, we considered 1) a multi-class classification problem over the Mini-Wiki dataset from [18] and 2) a matrix-completion (reformulated as regression) problem on the Jester1 dataset in [*Eigentaste: A constant time collaborative filtering algorithm*, Goldberg et al., 2001], containing user ratings of jokes. In the Mini-Wiki Dataset we have 813 tasks, 128 available points per task and 50 dimensions. In the Jester1 dataset we have 24983 tasks (users), 100 points (jokes) for each task and 100 dimensions. In fig. 1 we report the averaged test performance of our method in the feature map framework on these datasets. For the Mini-Wiki dataset we used the multi-class hinge loss, for the Jester1 dataset the absolute loss. Notice that in both cases, coherently to what already observed in the paper, our meta-learning approach (ONL-ONL) reveals to be an effective approach to transfer the knowledge among the tasks in comparison to solving each task independently (ITL).

[Meta-Review · NeurIPS 2019]

This paper presents a method for online-within-online meta-learning where each task is revealed one after another and online learning is applied for within-task. The primal-dual online learning is the main ingredient. All of reviewers agree that the paper is well written and has valuable contributions, while a few relevant work is already available. During the discussion period, a reviewer with most negative review raised his/her score, enabling us to reach a consensus.